# Limited proteolysis-coupled mass spectrometry captures proteome-wide protein structural alterations and biomolecular condensation in living cells

Franziska Elsässer [1], Roberta Florea [1], Felix Räsch[2], Mostafa Zedan [2], Nesli-Ece Sen[3], Tim Pflästerer[2], Tatjana Kleele [2], Robbie Loewith [3], Karsten Weis [2], Natalie de Souza [1] & Paola Picotti [1]✉

## Abstract

**The function of a protein is determined by its structure, which may change dynamically in response to post-translational modifications, interaction with other molecules, or environmental factors like temperature. Limited proteolysis-coupled mass spectrometry (LiP-MS) captures such structural alterations on a proteome-wide scale via the detection of altered protease susceptibility patterns of proteins. However, this technique has so far required cell lysis, which exposes proteins to non-native conditions and can disrupt labile interactions such as those occurring within biomolecular condensates. To study protein structures directly within cells, we developed in-cell LiP-MS. We optimized conditions for introduction of proteinase K into human cells using electroporation and validated that intracellular cleavage occurs. In-cell LiP-MS captured the known binding of rapamycin to FKBP1A within the cell. Moreover, it detected global protein structural alterations upon sodium arsenite treatment and captured the structural dynamics of hundreds of proteins from biomolecular condensates with peptide level resolution and within live human cells. The data allowed monitoring of structural alterations of individual sites on the involved proteins, such as known RNA-binding and intrinsically-disordered regions, and dissected the timing of the different events. We detected known (G3BP1) and novel structural alterations of proteins from stress granules as well as from nuclear speckles and validated alteration of nuclear speckles by fluorescence microscopy and of the protein SERBP1 by polysome profiling. Our dataset further provides a resource describing the structural changes of human proteins in response to a cellular stress leading to biomolecular condensation and pinpoints structurally altered regions. Comparison of LiP-based structural fingerprints before and after cell lysis revealed which human proteins are susceptible to structural change upon cell lysis, therefore guiding the design of future experiments requiring native protein structures.**

**Keywords** LiP-MS; Structural Proteomics; Stress Granules; Nuclear Speckles; Biomolecular Condensation

**Subject Categories** Proteomics; Structural Biology

## Introduction

Proteins serve many functions in cells, regulating growth, differentiation, and survival. Since protein structure is a key determinant of protein function, the study of protein structure before and after various perturbations (e.g., stress, disease, or treatment with drugs) yields insight into protein function and dysfunction in health and disease. Structural predictions based on Alphafold or RoseTTAFold are now available for over 98% of human proteins (Baek et al, 2021; Tunyasuvunakool et al, 2021), complementing the large body of experimentally solved protein structures. However, most available protein structures are a static snapshot, and information on protein structural changes between conditions is very limited.

Techniques to determine high-resolution protein structure typically require that the protein or protein complex of interest is purified. Such in vitro analyses do not recapitulate important properties of the cellular environment such as molecular crowding, presence of small-molecule or macromolecule interactors, and subcellular gradients of pH or metabolites. Also, certain features of protein structure depend on subcellular localization and might be altered upon cell lysis. Low-affinity or dynamic interactions, such as those within biomolecular condensates, can be lost or difficult to capture upon cell lysis due to dilution of the cellular content (Jain et al, 2016). It is therefore important to study protein dynamics within the native cellular environment.

In-cell nuclear magnetic resonance (NMR) and Förster resonance energy transfer (FRET) techniques provide information on the dynamics of protein structures with atomic resolution inside

[1]Institute of Molecular Systems Biology, Department of Biology, ETH Zurich, Zurich, Switzerland. [2]Institute of Biochemistry, Department of Biology, ETH Zurich, Zurich, Switzerland. [3]Department of Molecular and Cellular Biology, University of Geneva, Geneva, Switzerland. ✉E-mail: picotti@imsb.biol.ethz.ch

cells (Ha et al, 1999; Sakakibara et al, 2009), but typically one protein is studied at a time, both techniques require labeling, and FRET requires extensive optimization. Cryo-electron tomography can be used to solve protein structures in their cellular context, but studying protein dynamics remains challenging, and the approach is not applicable on a proteome-wide scale. Mass spectrometry-based methods report on structural changes of multiple proteins simultaneously. For example, protein footprinting methods like covalent protein painting (CPP) and fast photochemical oxidation of proteins (FPOP) cover many hundreds or even thousands of proteins within cells (Bamberger et al, 2021, 2022; Espino et al, 2015; Kaur et al, 2020; Shortt et al, 2023) or organisms (Espino and Jones, 2019; Son et al, 2024), while chemical crosslinking-based methods (XL-MS) also provide information on structural changes but at lower proteome coverage (Bakhtina et al, 2023; O'Reilly et al, 2023).

We previously developed limited proteolysis coupled mass spectrometry (LiP-MS), a structural proteomics approach that allows the study of structural changes, as measured by protease susceptibility changes, in thousands of proteins simultaneously with peptide-level resolution (Feng et al, 2014). In LiP-MS, native proteins are cleaved by a protease for a short period of time, followed by complete trypsin digestion under denaturing conditions, which results in structure-specific peptide patterns that can be quantitatively read out by mass spectrometry (Fig. 1A). We have shown that the protease susceptibility changes in LiP-MS can detect numerous molecular events, including allosteric regulation, changes in enzyme activity, protein-protein interactions, post-translational modification, protein-small molecule interactions, and protein aggregation events (Cappelletti et al, 2021; Piazza et al, 2018). As such, LiP-detected structural changes include not only conformational changes, unfolding or modification of individual proteins, but also report on changes in binding events. LiP-MS has been applied for drug target identification (Piazza et al, 2020) and to characterize structural changes in biofluids of individuals suffering from Parkinson's disease (Mackmull et al, 2022). Although powerful, LiP-MS has so far only been applicable to cell or tissue lysates (Malinovska et al, 2023; Schopper et al, 2017), where various protein structures may be altered due to the loss of intracellular conditions.

Here we describe our development of in-cell LiP-MS. We show delivery of proteinase K (PK), the low specificity protease also used for classical LiP, into living mammalian cells using an electroporation approach optimized to ensure rapid (millisecond) and reproducible intracellular uptake of the protease and the desired level of intracellular cleavage. We detected PK-induced LiP signals across the proteome in an electroporation-dependent manner and validated the delivery of active protease into cells using an intracellular sensor based on FRET. In-cell LiP-MS captured the well-characterized and high-affinity interaction of rapamycin with the protein FKBP1A (Heitman et al, 1991) as previously observed with standard LiP-MS in cell lysates (Piazza et al, 2020). In-cell LiP-MS also revealed structural changes within biomolecular condensates that are formed in part through low-affinity interactions. Under arsenite stress, in-cell LiP-MS captured structural alterations in known and novel stress granule-related proteins as well as in nuclear speckle proteins, which we could validate with fluorescence microscopy. Our study provides the first available resource of in vivo proteome-wide protein structural changes during arsenite stress, identifies proteins that undergo structural alterations upon cell lysis, and pinpoints structurally altered regions in proteins, including those involved in biomolecular condensation. The in-cell LiP-MS method will allow the study of protein structural changes under any conditions of interest within living cells with peptide-level resolution and on a proteome-wide scale.

## Results

### Development of in-cell LiP-MS

To enable the study of protein structural states in their native cellular environment, we identified electroporation conditions to introduce PK into mammalian cells (Fig. 1B). We subjected HEK293 cells to electroporation immediately after addition of PK, and then incubated briefly (1 min at 37 °C) to allow limited proteolysis of intracellular proteins. Extracellular PK was removed by washing, PK activity was quenched by addition of a chaotrope and heating to 98 °C, and samples were snap-frozen. Cells were subsequently lysed, trypsin treated, and analyzed with quantitative, label-free MS.

To assess whether PK had been delivered into cells, we assessed the fraction of fully tryptic peptides (FT; i.e., those with two tryptic ends) and half-tryptic peptides (HT, i.e., those with only a single tryptic end), thus using proteolytic cleavage itself as a readout of PK delivery. Peptides with one or two non-tryptic termini are likely to be generated by PK cleavage. Therefore, an increase in HT peptide intensity or a decrease in FT peptide intensity in a sample sequentially digested with PK and trypsin, reports on PK cleavage events when compared to a sample digested with trypsin alone. We observed evidence of PK cleavage in cells electroporated in the presence of PK, but not in the absence of electroporation (Fig. 1C; cleavages were detected on 2491 proteins). Simply adding PK to cells without electroporation also yielded no significant increase in proteolytic cleavage over the no-PK control (Fig. 1D,G), likely because we removed extracellular peptides by washing. Gene ontology enrichment analysis showed that proteins with peptide signatures indicative of PK cleavage were significantly enriched in cytosolic and nucleus-associated terms (Fig. 1F), as would be expected upon PK entry into cells. The protein content of the cell supernatant was only minimally increased upon electroporation (Appendix Fig. S1A; n = 4 replicates), and we could detect no significant abundance changes of specific proteins (Appendix Fig. S1B; FC > 1.5, q-value < 0.05, n = 4 replicates), indicating that there is no systematic loss of any specific proteins from cells upon electroporation. We conclude that electroporation introduces PK into mammalian cells with only a small and likely transitory loss of cellular content (Saulis, 1997).

In order to identify conditions that result in rapid and reproducible intracellular delivery of PK with minimal electroporation-induced effects on cellular physiology, we examined proteome-wide changes in peptide and protein levels upon electroporation alone. We observed no significant electroporation-induced changes with one 25-ms pulse at 1000 V or with up to two 25-ms pulses at 800 V (Fig. 1E; Appendix Fig. S1C). Moreover, FT and HT peptides were symmetrically distributed in all tested electroporation settings, suggesting that electroporation alone does not cause protein cleavage (Appendix Fig. S1C,F).

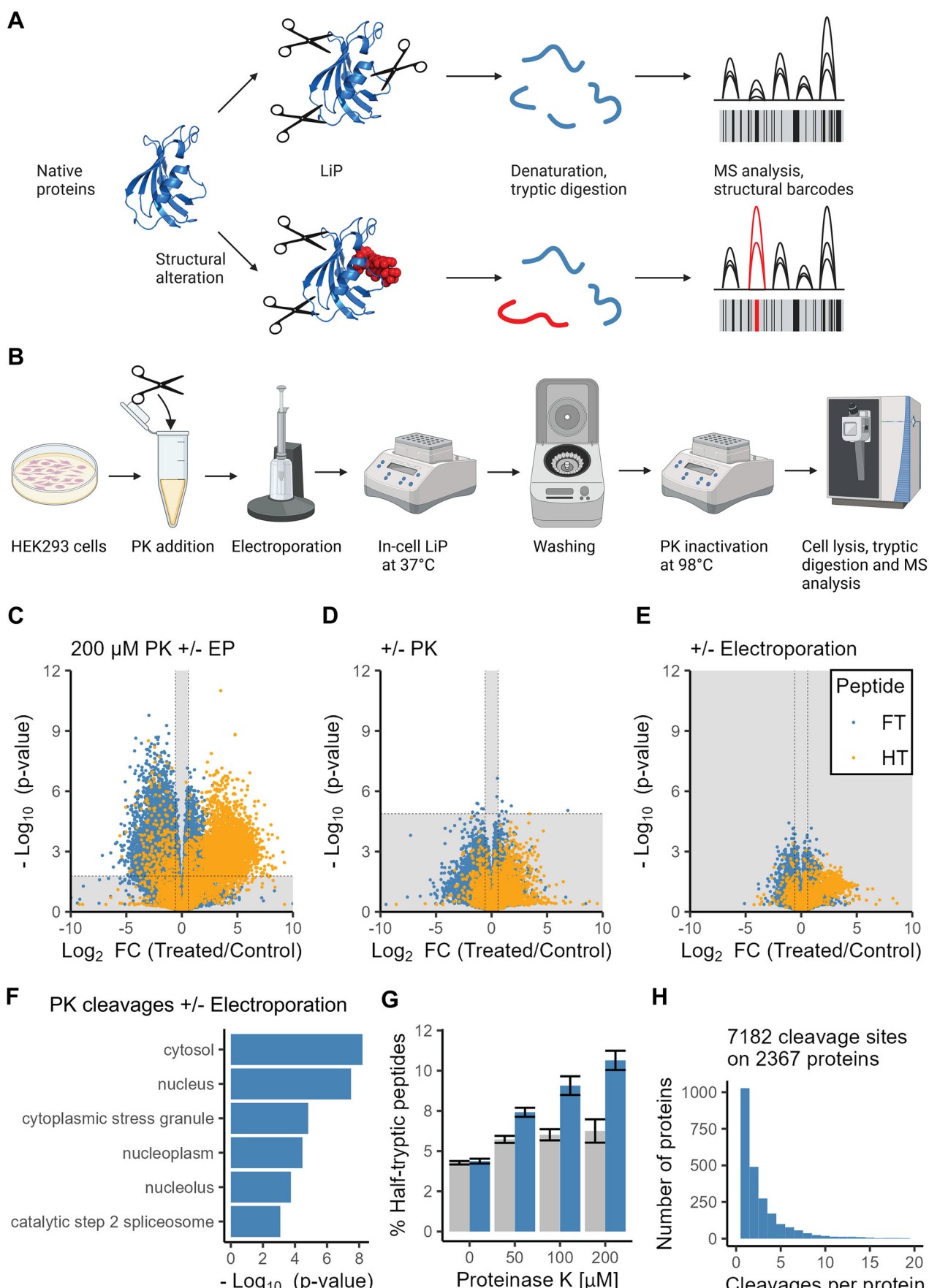

**Figure 1.  Development of in-cell limited proteolysis-coupled mass spectrometry (in-cell LiP-MS).**

(A) Illustration of LiP-MS. Native proteins are incubated with a protease (e.g. PK) for a short time to generate structure-specific peptides. The samples are then subjected to denaturing conditions, and proteins are digested with trypsin and analyzed by mass spectrometry. A structural change leads to altered proteolytic cleavage. (B) Schematic of the in-cell LiP-MS workflow. PK is introduced into cells by electroporation. During a brief incubation, structure-specific peptides are generated. After protease inactivation, cell lysis and trypsin treatment, the peptides are analyzed by mass spectrometry. (C–E) Changes in peptide intensities in treated samples compared to control samples. Shown are (C) samples treated with 200 µM PK with and without electroporation, (D) samples with and without 200 µM PK in the absence of electroporation, and (E) samples with and without electroporation (1 pulse of 25 ms at 1000 V) in the absence of PK. Each data point represents a single peptide; half-tryptic peptides are shown in orange, and fully tryptic peptides are shown in blue. The x-axes report the $\log_2$-transformed ratio of peptide intensities. The y-axes report the negative $\log_{10}$-transformed q-value, which is the p-value of the t-test that was adjusted for multiple testing. The shaded gray region marks significance levels (FC > 1.5, q-value < 0.05, $n = 5$ technical replicates). (F) Gene ontology enrichment analysis of proteins with PK cleavages after treatment with 200 µM PK with and without electroporation (p-value < 0.001). (G) Fraction of half-tryptic peptides without (control; gray) and with electroporation (blue) of indicated PK concentration ($n = 5$, error bars indicate standard deviation). (H) Cleavages per protein in optimized in-cell LiP-MS (100 µM PK, 1 pulse of 25 ms at 1000 V, 2 min incubation time, $n = 6$ technical replicates). Cleavages were determined by counting half tryptic (HT) peptides that increased in intensity and fully tryptic (FT) peptides that decreased in intensity upon electroporation of PK, requiring the counted peptides to be non-overlapping and to be found in at least 3 replicates of a given condition. To generate technical replicates, cells were split into the indicated number of aliquots before treatment.

Under higher voltage and with more pulses, we observed a drop in peptide and protein intensity relative to the non-electroporated control (Appendix Fig. S1C,D), suggesting that high voltage induces protein aggregation or cleavage. Cell viability as assayed with trypan blue staining did not significantly decrease within 5 min after electroporation with one 25-ms pulse of 1000 V (ANOVA q-value < 0.05; Appendix Fig. S1E), and we chose these conditions for subsequent experiments. There were no abundance changes for any proteins up to 4 h after electroporation (Appendix Fig. S1G) including for several stress markers with good protein coverage (HSP60 70%, Thioredoxin 72%, Cytochrome c 46%), indicating that our electroporation conditions do not markedly stress the cells. In keeping with these results, ATP rate measurements (Seahorse assay) showed no signs of oxidative stress. We observed a slight but statistically insignificant decrease in oxygen consumption rate (OCR), and therefore in mitochondrial respiration, in cells 24 h after electroporation (Appendix Fig. S1H), indicating if anything a small reduction in oxidative stress and reactive oxygen species relative to control cells.

To maximize the structural information from an in-cell LiP experiment, we then optimized the enzyme-to-substrate ratios for the efficiency and the reproducibility of cleavage. Whereas the fraction of HT peptides increased with increasing PK concentrations (Fig. 1G; Appendix Fig. S2A–C; 17% HT peptides at 1000 µM PK), the reproducibility of cleavage decreased compared to 100 µM PK (Appendix Fig. S2A–C), possibly because of over-digestion. Most peptides were quantified consistently across all replicates showing high reproducibility of the method (Appendix Fig. S2D). Our data indicate that 100 µM extracellular PK (for $3 \times 10^6$ cells) led to the highest number of reproducible intracellular cleavages, and we thus chose this condition for electroporation in subsequent experiments. After further optimizing incubation time and number of replicates (Fig. EV1A), we observed about 15% HT peptides, compared to the 20–30% HT peptides typical for classical LiP (Malinovska et al, 2023) (Fig. EV1B), and reducing it yielded lower digestion reproducibility; we therefore selected 2 min as the optimal incubation time. Under these conditions, 17050 peptides on 3616 proteins, out of 4760 detected proteins, changed significantly relative to the non-electroporated control (Fig. EV1A). Based on HT peptides that increased in intensity and FT peptides that decreased in intensity, which we further required to be non-overlapping and detected in at least 3 replicates per condition, we identified minimally 7182 distinct PK cleavage sites on 2367 proteins upon electroporation of PK (Figs. 1H and EV1A).

We note that this measure will substantially underestimate the structural coverage of in-cell LiP-MS in a perturbation experiment where the perturbation causes protein structural changes across the proteome. Such changes will generate PK accessibility changes that are much larger than those seen at steady state, and these may be detected in addition. For example, only a fraction of the changing peptides in an in-cell LiP-MS experiment after 90 min of arsenite treatment were observed to be cleaved at steady state (Fig. EV1C). In principle, in-cell LiP-MS can detect structural changes for any region that is detected by mass spectrometry, as long as that region changes proteolytic accessibility sufficiently under a given condition. In other words, the sequence coverage of proteins after differential analysis, which is on average 29% for optimized in-cell LiP-MS conditions (Fig. EV1D,E), will determine the maximum structural coverage.

We also quantified cleavage sites along the sequence of all detected proteins, considering only HT peptides because they allow exact localization of the cleavage site. For the 3203 bona-fide HT peptides, cleavages were distributed almost equally along protein sequences without a marked preference for protein termini, including for proteins for which ≤2 cleavages were found (Fig. EV1H). We note that this analysis excluded 30 peptides that represented the very N terminus of a protein. To conclude, we have identified conditions to deliver PK into HEK293 cells to generate reproducible PK-induced cleavages in more than 2300 mainly intracellular proteins without unduly compromising cell viability or the cellular proteome.

## Validation of PK uptake

To orthogonally test that PK enters cells upon electroporation and is active intracellularly, we used HEK293 cells expressing a FRET-based sensor of protease activity. The sensor consists of ECFP and YPet proteins separated by a short linker sequence (Gray et al, 2010). Upon excitation of ECFP, yellow YPet fluorescence indicates an intact sensor, whereas cleavage of the linker results in loss of FRET signal and therefore an increase in the ECFP:YPet fluorescence ratio. We first confirmed that the sensor behaved as expected upon addition of PK to a native lysate of the reporter cells. We observed cleavage within minutes of adding PK, and prior exposure of the protease to our optimized electroporation conditions did not affect its activity (Fig. 2A). Electroporation of PK into cells also resulted in an increase in the ECFP fluorescence

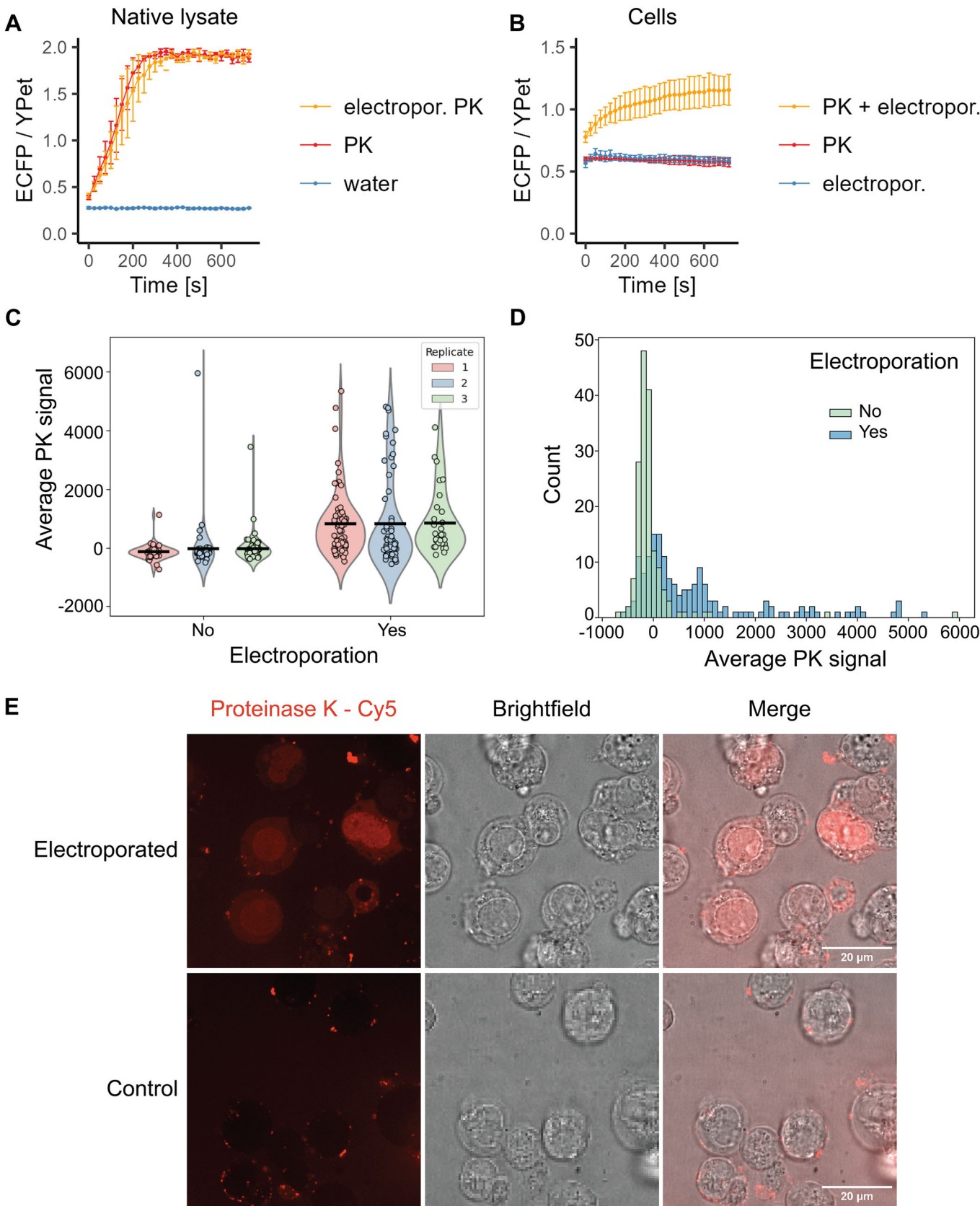

◄ **Figure 2. Intracellular PK delivery by electroporation.**

(A) FRET signal as a function of time of a native lysate of HEK293 cells containing an ECFP/YPET FRET sensor of protease activity that was incubated with PK, PK exposed to electroporation conditions, or water ($n = 4$ technical replicates, error bars indicate the standard deviation). (B) FRET signal from HEK293 cells expressing an ECFP/YPET FRET sensor that were electroporated with PK, electroporated without protease, or that were incubated with PK without electroporation ($n = 3$ technical replicates, error bars indicate the standard deviation). (C) PK-Cy5 signal after background subtraction in cells with and without electroporation from three independent technical replicates. $p$-value between mean of electroporated and control replicates <0.001, two-tailed t-test (replicate 1: $p$-value = 2.045E-07; replicate 2: $p$-value = 0.0006261; replicate 3: $p$-value = 2.032E-07). (D) Distribution of the average PK-Cy5 signal intensity from (C). (E) Example image of PK-Cy5 with and without electroporation. To generate technical replicates, cells were split into the indicated number of aliquots before treatment. Source data are available online for this figure.

relative to that of YPet, confirming that active PK entered the cells (Fig. 2B). The larger variation we observed in the in-cell context was likely to be due to cell-to-cell variability, for example in sensor expression and PK internalization, as well as to technical variability in electroporation. Neither addition of PK without electroporation nor electroporation alone affected sensor fluorescence (Fig. 2B). These data provide orthogonal evidence that electroporation allows the introduction of active PK into mammalian cells.

Addition of PK to cells without electroporation yielded a small increase in the intensity of HT peptides compared to untreated cells (Fig. EV1B), which can be attributed primarily to cleavages occurring on the cell surface, since these cleavages were enriched for proteins on the plasma membrane (Fig. EV1F,G). The percentage of HT peptides did not change if we included an endocytosis inhibitor prior to addition of PK (Appendix Fig. S3A), suggesting that endocytosis does not play a role in PK uptake in the timeframe of the in-cell LiP-MS experiment (5 min).

To more directly visualize PK uptake, we fluorescently labeled PK with Cy5 and used an inactivated version of this labeled protein (to prevent PK-induced cell clumping) for electroporation and imaging with confocal microscopy. As expected, the average Cy5-PK signal per cell was significantly increased upon electroporation ($p$-value < 0.001, $n = 3$; Fig. 2C), with 44.5% of cells showing a Cy5 signal (Fig. 2D; 73 of 164 cells; average fluorescence >500), compared to only 3.7% of non-electroporated cells (6 of 164 cells). Importantly, these data showed very stable average fluorescence intensity across replicates, which should minimize effects of uptake on the robustness of the method. Cy5-PK entered both the cytosol and the nucleus of cells (Fig. 2E).

We estimated the number of PK molecules that enter the cell upon electroporation by comparison to a calibration curve consisting of four accurately quantified heavy-labeled PK-specific peptides in cell lysates. After addition of 12 nmoles of PK to cells followed by electroporation under our optimized conditions, the electroporated sample contained 52.5 pmol PK and the control sample (no electroporation) contained 16.0 pmol PK (Appendix Fig. S3B). The PK we detected in the control sample is likely to be extracellular PK that was not removed during washing. We reasoned that the difference between the two samples (36.6 pmol) corresponds to intracellular PK.

## In-cell LiP-MS reproducibly identifies changes in proteolytic accessibility upon drug binding

Next, we asked whether in-cell LiP-MS can capture the well-characterized high-affinity binding of rapamycin to the protein FKBP1A. After treatment with 20 µM rapamycin for 10 min, cells were electroporated with PK and processed following the optimized in-cell workflow. We also compared the results to a standard LiP-MS experiment, using a similar enzyme-to-substrate ratio in both conditions.

In-cell LiP-MS identified multiple FKPB1A-derived peptides as among the strongest hits in a comparison of rapamycin-treated and control (DMSO-treated) cells (Fig. 3A; FC > 1.5, $p$-value < 0.01, $n = 6$ replicates). Standard LiP-MS also identified multiple FKBP1A-derived peptides among the most significant hits (FC > 1.5, $p$-value < 0.01, $n = 4$ replicates; Fig. 3B), as we have previously shown (Piazza et al, 2020). Three of the four changing FKBP1A peptides detected by in-cell LiP-MS overlap with those detected by standard LiP-MS, but were longer due to missed cleavages. We note that we defined significance using a $p$-value threshold of $p < 0.01$ for this experiment because no significant changes were detected after adjustment for multiple testing. This effect is frequently observed in cases where only a very small number of changes occur in a setting of tens of thousands of detected peptides, as expected for drug-target experiments with a specific drug like rapamycin (Piazza et al, 2020).

We compared several characteristics of the data generated by the two methods. The average number of detected proteins was similar in both approaches (in-cell LiP: 4188; standard LiP: 4006), and the average number of peptides was higher for in-cell LiP (73286) than standard LiP (54223; Fig. EV2A–C), corresponding to higher protein sequence coverage in the in-cell method (Fig. 3C). Most peptides were quantified consistently across replicates (Figs. 3D and EV2A). We observed slightly higher coefficients of variation for in-cell LiP-MS, but still within the range of CVs reported for standard LiP experiments (Malinovska et al, 2023), and a similar fraction of HT peptides between the in-cell and in-lysate methods (Fig. 3E,F). In standard LiP-MS, peptides without missed trypsin cleavages account for 85% of total peptide intensity, compared to 73% in in-cell LiP-MS (Fig. 3G), owing to the use of guanidinium hydrochloride as a PK inactivation agent in the in-cell method and therefore less efficient trypsin digestion (Proc et al, 2010) (Methods). Finally, we observed higher abundance of long peptides (>20 amino acids) for in-cell than standard LiP-MS (Figs. 3H and EV2D,E), likely because intracellular PK causes less secondary cleavages and again because of the lower efficiency of tryptic digestion under the experimental conditions of the in-cell approach (Methods). Note that, for both in-cell and standard LiP-MS, the direction of peptide intensity changes upon rapamycin treatment can deviate from expected behavior due to these technical effects (i.e., secondary cleavages by PK and missed cleavages by trypsin). We detected no known downstream effects of rapamycin treatment by either method, most probably because our analyses were done at too early a time point to detect them, as confirmed by Western blot (Fig. EV2F).

In summary, despite a small increase in missed trypsin cleavages, in-cell LiP-MS generated data of similar quality to the

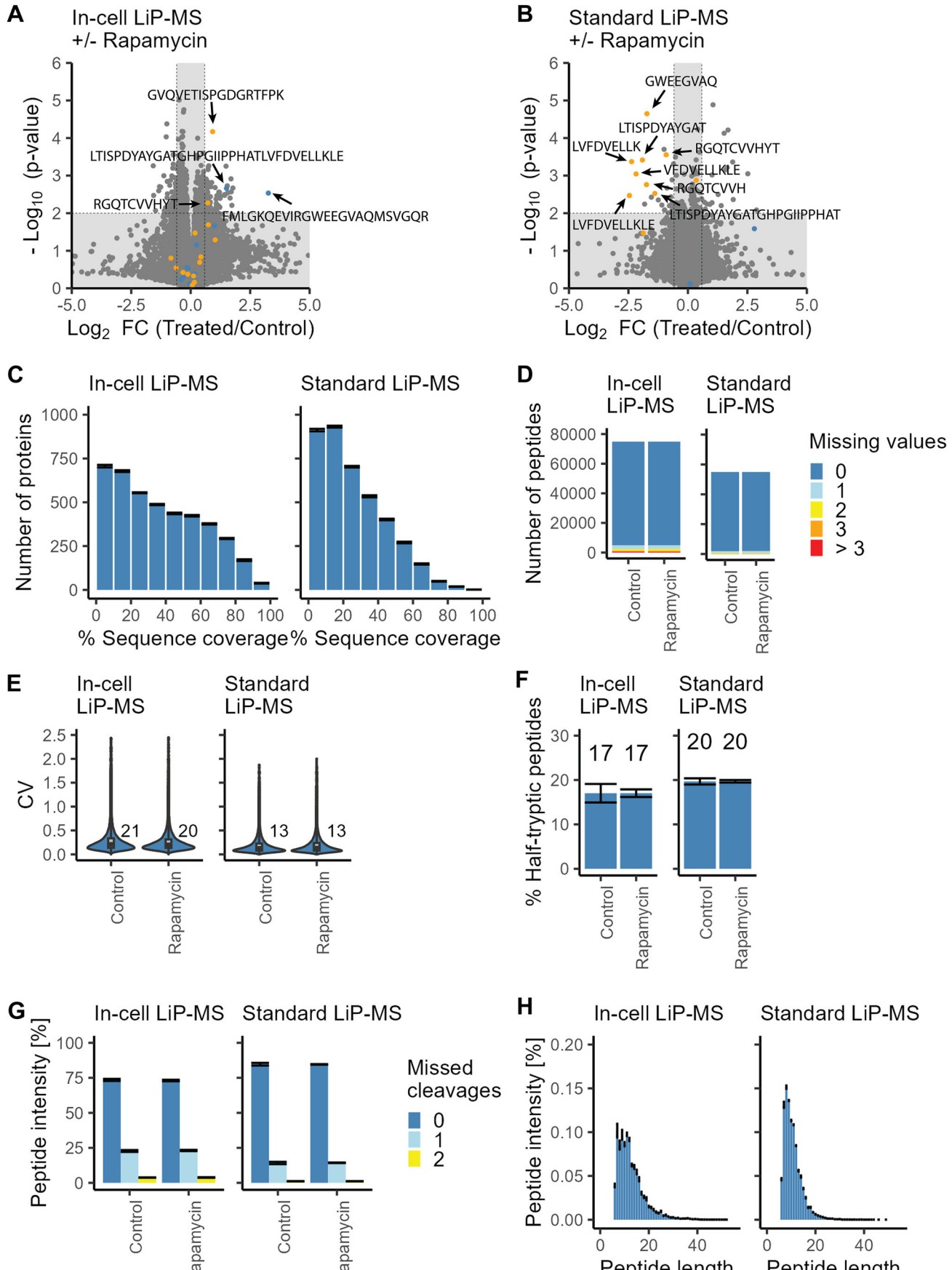

**Figure 3.  Target detection and reproducibility of in-cell LiP-MS.**

(A) Changes in in-cell LiP-MS peptide intensities in HEK293 cells treated with 20 µM rapamycin compared to DMSO control. Each data point represents a single peptide; half-tryptic peptides of FKBP1A are shown in orange, fully-tryptic peptides of FKBP1A are shown in blue. The lines marks significance levels (FC > 1.5, two-sample unpaired t-test, *p*-value < 0.01, *n* = 6 technical replicates). (B) Changes in standard LiP-MS peptide intensities in a native lysate of HEK293 cells treated with 10 nM rapamycin compared to DMSO control (6 technical replicates). Each data point represents a single peptide; half-tryptic peptides of FKBP1A are shown in orange, fully-tryptic peptides of FKBP1A are shown in blue. The shaded gray region marks significance levels (FC > 1.5, two-sample unpaired t-test, *p*-value < 0.01, *n* = 4 technical replicates). The four FKBP1A peptides with the lowest *p*-value are highlighted. (C) Overall sequence coverage of experiments in (A, B). The plot shows the indicated quantities across both conditions (in-cell LiP-MS *n* = 12 technical replicates; standard LiP-MS *n* = 8 technical replicates). Error bars are shown in black. (D) Number of peptides with missing values per treatment in (A, B). The plots report the number of replicates per condition, in which a specific peptide was not quantified. A missing value of 0 indicates that the peptide was quantified in all six replicates, 1 indicates that the peptide was not quantified in 1 out of 6 replicates, and so on. (E) Coefficient of variation (CV) of peptide intensities per treatment in (A, B). (F) Fraction of half-tryptic peptides relative to total peptide intensity for the experiments in (A, B). (G, H) Plots show the indicated quantities across both conditions (in-cell LiP-MS *n* = 12; standard LiP-MS *n* = 8). (G) Fraction of peptides with missed cleavages after tryptic digestion relative to total peptide intensity for the experiments in (A, B). (H) Fraction of peptides with indicated length relative to total peptide intensity in (A, B). Only peptides with up to 50 amino acids are shown. To generate technical replicates, cells were split into the indicated number of aliquots before treatment.

established in-lysate method. In a well-characterized rapamycin treatment experiment, the in-cell approach correctly identified the protein expected to undergo a change in proteolytic accessibility within the cell upon drug treatment, and detected the same changes within this target protein as the standard LiP-MS method.

## Subcellular coverage of in-cell LiP-MS

To test whether organellar coverage differed between in-cell and in-lysate LiP-MS, we made use of control (DMSO-treated) samples (*n* = 6 replicates, in-cell LiP; *n* = 4 replicates, in-lysate LiP; all measurements on the same mass spectrometer). Standard LiP-MS in our laboratory is generally conducted in four replicates; we increased this number to six replicates for in-cell LiP-MS to better account for variability of the in-cell setup. We first assessed the fraction of HT peptides in proteins annotated according to their cellular compartment, as a measure of the efficiency of PK cleavage in different cellular locations. For most organelles, the average fraction of half-tryptic peptides was similar or slightly lower for in-cell LiP-MS than for in-lysate LiP-MS, reflected also in the comparison for all peptides from all organelles (Appendix Fig. S3C). In both methods, cleavage was relatively low for proteins in the mitochondrial matrix, mitochondrial intramembrane space, Golgi lumen, and lysosomes. Thus, PK cleavage varies slightly between the two setups and for a few organelles.

Next, we asked whether we could use our data to detect proteins that undergo structural alterations upon cell lysis, reasoning that this set of proteins would benefit in particular from the in-cell setup. Of 76,592 peptides detected in in-cell LiP, 75,172 were detected in at least 3 of 6 replicates; 28,263 peptides were unique to in-cell LiP-MS compared to in-lysate LiP-MS, of which 27,041 were detected in at least 3 replicates (Fig. 4A). Notably, for most peptides the ratio of intensities in in-cell LiP-MS to lysate LiP-MS was close to 1 (Fig. 4B). Nevertheless, there were several proteins or protein regions with relatively poor correlation and we investigated these proteins further.

We tested whether proteins with low correlation between in-cell LiP-MS and classical LiP-MS were characterized by any particular feature, such as a subcellular localization or function, using a protein-level score of similarity between in-cell LiP-MS and classical LiP-MS that we calculated by averaging the peptide ratios (for proteins >3 peptides). Proteins that differed most between the methods (protein score <0.9 or >1.1) were located in mitochondria (EBI human GOA database, release 2018-02-20; Fig. 4C), whereas

proteins in the cytosol, nucleus, vesicles and the Golgi apparatus showed similar structural fingerprints in both methods (protein score >0.9 and <1.1; Fig. 4D). Correlation at the peptide level was also lower in mitochondria than in other organelles (ρ = 0.65; Fig. EV3A), and peptide ratios between in-cell LiP-MS and classical LiP-MS revealed two distinct peaks for mitochondrial proteins (Fig. 4E). In contrast, peptide ratios of proteins located in other organelles exhibit a single peak with an in-cell:lysate intensity ratio close to 1 (Fig. EV3B). This indicates two populations of mitochondrial proteins that are either differentially cleaved by PK in the two LiP-MS setups, or that are differentially covered by the two methods. The latter situation may occur because the non-denaturing lysis in standard LiP-MS may result in loss of mitochondria during the centrifugation step of sample preparation (mild mechanical lysis is commonly used for isolation of intact mitochondria (Zhou et al, 2023); this would be less likely in the in-cell context since denaturing conditions are used after the LiP step (Methods).

Next, we asked if LiP peptides reflective of low structural similarity between in-cell LiP and classical LiP were enriched in specific domains (Dörig et al, 2024). We excluded mitochondrial proteins from this analysis because we cannot distinguish functional differences from differences in localization. Domain-level gene ontology enrichment analysis for peptides with a higher intensity in in-cell LiP-MS (ratio >1.1 of $\log_{10}$-transformed peptide intensities) showed a significant enrichment for RNA and DNA binding domains (Fig. 4F). Interestingly, under physiological conditions (pH 7.0 and pH 7.4) as we would expect both within cells and in native lysates, PK exhibits both positively charged and negatively charged surface regions (Appendix Fig. S3D). Charged intracellular DNA/RNA molecules may recruit PK via its positive surface, leading to cleavage of DNA/RNA binding proteins and therefore to the enrichment of these domains in the in-cell data. Since DNA/RNA molecules are likely to be removed by centrifugation after cell lysis in standard LiP-MS prior to the PK cleavage step, this effect would not occur. While further experiments would be needed to test these hypotheses, our data indicate that in-cell LiP-MS provides information on RNA and DNA interactions with proteins that are not accessible with the classical method.

## In-cell LiP-MS identifies structural changes in arsenite stress

The promise of an in-cell method for detecting structural alterations is that it may be able to capture the dynamics of structures that are too

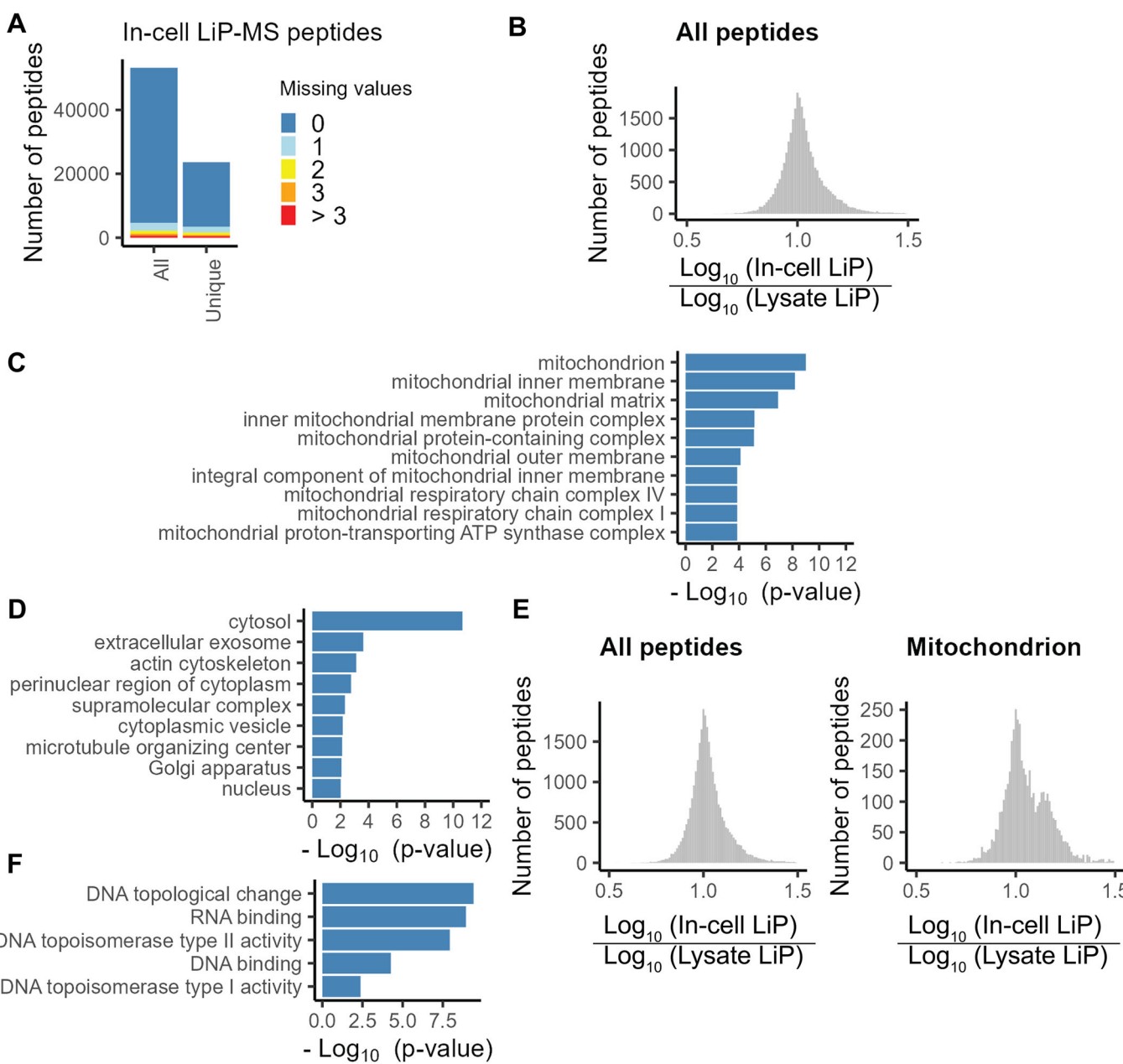

**Figure 4. Organelle and domain coverage of LiP-MS in cells and in lysate.**

(A) Number of peptides with missing values per treatment detected by in-cell LiP-MS. The plots report the number of replicates per condition, in which a specific peptide was not quantified. A missing value of 0 indicates that the peptide was quantified in all six replicates, 1 indicates that the peptide was not quantified in 1 out of 6 replicates. The bars show the total number of peptides detected by in-cell LiP-MS and peptides unique to in-cell LiP-MS in HEK293 cells between in-cell LiP-MS replicates after 5 min of DMSO treatment (in-cell LiP $n = 6$ technical replicates; lysate LiP $n = 4$ technical replicates). (B) Ratio of average peptide intensities across replicates in lysate and in-cell LiP-MS. (C, D) For proteins with >3 peptides detected in at least 3 replicates of both datasets, a protein score was calculated (protein score: mean ratio of $\log_{10}$-transformed peptide intensities between in-cell LiP and lysate LiP across peptides). Gene ontology enrichment of cellular compartments was performed for (C) proteins with protein scores <0.9 or >1.1, and (D) protein scores >0.9 and <1.1 ($p$-value < 0.001, Fisher's exact test using the elim-algorithm in *topGO*). (E) Ratio of $\log_{10}$-transformed peptide intensities between in-cell LiP and lysate LiP, for all peptides and for peptides mapping to proteins located in the mitochondrion based on gene ontology annotation. (F) Gene ontology enrichment of protein domains was performed for peptides with a ratio >1.1 of $\log_{10}$-transformed peptide intensities between in-cell LiP and lysate LiP (excluding mitochondrial peptides). Domain annotations are from Interpro database ($p$-value < 0.001, one-sided Fisher's exact test). To generate technical replicates, cells were split into the indicated number of aliquots before treatment.

complex to properly reconstitute in vitro and that may be partly lost upon cell lysis, such as biomolecular condensates (Jain et al, 2016). We therefore asked whether in-cell LiP-MS can detect structural changes of proteins in stress granules, cytosolic biomolecular condensates composed of proteins and RNA that form in response to various stresses. Their formation is at least partly driven by liquid-liquid phase separation via low-affinity interactions (Glauninger et al, 2022). We treated HEK293 cells with sodium arsenite, known to induce stress granules, for up to 90 min. At different time points (0, 10, 20, and 90 min after arsenite treatment), cells were detached in pre-warmed PBS, collected by centrifugation at $200 \times g$ for 30 s, resuspended in PBS, immediately electroporated with PK and then processed by in-cell LiP-MS. We confirmed that stress granules formed between 10 and 20 min of arsenite treatment using immunofluorescence staining for the stress granule marker G3BP1 (Fig. 5A), in line with previous observations (Jain et al, 2016).

After 10 min of treatment, before stress granules were observed by microscopy, we detected structural changes in only three proteins in arsenite-treated samples relative to untreated control; these were Creatine kinase B-type (CKB), Protein phosphatase 1 regulatory subunit 12A (PPP1R12A), and SERPINE1 mRNA-binding protein 1 (SERBP1) (Fig. 5B; FC > 1.5, q-value < 0.05); further, each change was reported by only a single peptide per protein. At 20 min of arsenite treatment, 2325 peptides on 1149 proteins had changed significantly, rising to 20,841 peptides on 3146 proteins at 90 min of treatment. With few exceptions (1 of 4204 proteins at 10 min, 13 of 4207 proteins at 90 min arsenite treatment), protein abundance levels did not change significantly in this time frame (Fig. EV4A; FC > 1.5, q-value < 0.05, $n = 6$ replicates), and we therefore did not normalize for protein abundance changes.

Gene ontology enrichment analysis showed that proteins that had changed structure after 20 min and 90 min of arsenite treatment were indeed enriched in components of stress granules (Fig. 5C). Moreover, proteins involved in RNA binding and RNA-related biological processes, again associated with stress granules, were also significantly enriched (Fig. EV4B,C). We observed that more cleavage events were detected at later time points as measured by the fraction of HT peptides (average percentage of HT peptides was 1.7-fold higher at 90 min arsenite treatment than in untreated samples; Appendix Fig. S4A), suggesting that the proteome of HEK cells may become slightly more susceptible to PK cleavage upon arsenite treatment. Other quality measures indicate slightly more variability than in previous experiments (Appendix Fig. S4B–F).

Next, we asked whether in-cell LiP identified structural changes in known stress granule-associated proteins. Interestingly, we captured changes consistent with the known mechanism of G3BP1 (Fig. 6A), a core component of stress granule assembly. Stress granule formation is triggered by a structural change in G3BP1 in response to free RNA. G3BP1 consists of a NTF2-like domain, which is required for dimerization (Fig. 6B), an RNA recognition motif, and three intrinsically disordered regions (IDRs). Both dimerization and RNA binding by G3BP1 are essential for stress granule assembly (Sanders et al, 2020). It has been suggested that G3BP1 adopts a closed conformation in the absence of stress, held together by electrostatic interactions between IDR1 and IDR3. When free RNA levels increase during arsenite stress, the interaction between the IDRs is disrupted and RNA binds to the RNA recognition motif and to IDR3. The RNA-bound open state of G3BP1 serves as the core of the stress granule and

leads to condensation (Yang et al, 2020). After 20 min of arsenite treatment, we detected only one altered LiP peptide on G3BP1: this peptide is located exactly on the RNA recognition motif of the protein (Fig. 6C), in line with the proposed model which should make the RNA binding domain accessible upon stress. At 90 min, structural changes occurred within all domains of G3BP1 (Fig. 6C). This suggests that structural changes associated with biomolecular condensation can be detected by in-cell LiP-MS.

More broadly, in-cell LiP-MS identified novel structural changes in 190 of 200 high confidence stress granule-associated proteins (based on the RNAgranuleDB v2.0 database) that are also MS-detectable (Data ref: Millar et al, 2023). We obtained peptide-level resolution of protein structure dynamics, as shown for 35 stress granule proteins selected manually for their special interest from a structural perspective, because they are known to be involved in stress granule formation (Fig. EV5). Although stress granules are not detected by microscopy until after 20 min of arsenite treatment, significantly changed LiP peptides were detected on three proteins, SERBP1, PPP1R12A, and CKB, after 10 min (Fig. 5B). PPP1R12A associates with stress granules (Yang et al, 2020). SERBP1 acts as a ribosome dormancy or preservation factor upon mTOR phosphorylation (Shetty et al, 2023) and, upon arsenite treatment, localizes to both stress granules and nucleoli (Lee et al, 2014). SERBP1 has also been implicated in clearance of stress granules after arsenite stress by modulating the degradation of G3BP1 (Wang et al, 2023). To our knowledge, CKB has not been previously associated with stress granules. Thus, we provide information on the structure dynamics of novel and known stress granule proteins during arsenite stress, with many changes mapping to known disordered and RNA recognition motifs (RRM) (Fig. 7).

We then further investigated SERBP1, for which the yeast homolog Stm1 has been found to act as a ribosome dormancy factor in other stresses (Shetty et al, 2023). Based on the structural change of SERBP1 in in-cell LiP-MS data and its known role to promote polysome disassembly in nitrogen starvation (Shetty et al, 2023), we predicted a stronger association of SERBP1 with monosomes (i.e., dormant ribosomes) already 10 min after sodium arsenite treatment. To test this, we performed polysome profiling at this time point and compared to the profiles of untreated control cells. We observed that 10 min of arsenite treatment led to a higher monosome peak (Fig. 8A,B), indicating polysome collapse, even though stress-granules are not yet visible under these conditions. We then extracted the proteins from selected fractions, and probed for SERBP1 and for the ribosomal protein Rpl7 on Western blot. Rpl7 levels confirmed polysome collapse upon arsenite treatment. In control conditions, SERBP1 was detected at higher levels in the fraction unbound to ribosomes, in fractions containing free 40S and 60S ribosome subunits, and to some extent in the fractions containing polysomes. Upon arsenite treatment, unbound SERBP1 and polysome-associated SERPB1 were almost undetectable. Instead, SERBP1 was detected in fractions containing 80S ribosomes and free 40S and 60S ribosomal subunits. This association with dormant ribosomes is in line with previous experiments in yeast and mammalian cells subject to nitrogen starvation (Shetty et al, 2023), and shows that also the mammalian SERBP1 may inhibit translation under arsenite stress prior to stress granule formation by causing ribosome dormancy.

Interestingly our gene ontology enrichment data suggested that nuclear speckles, another membraneless biomolecular condensate,

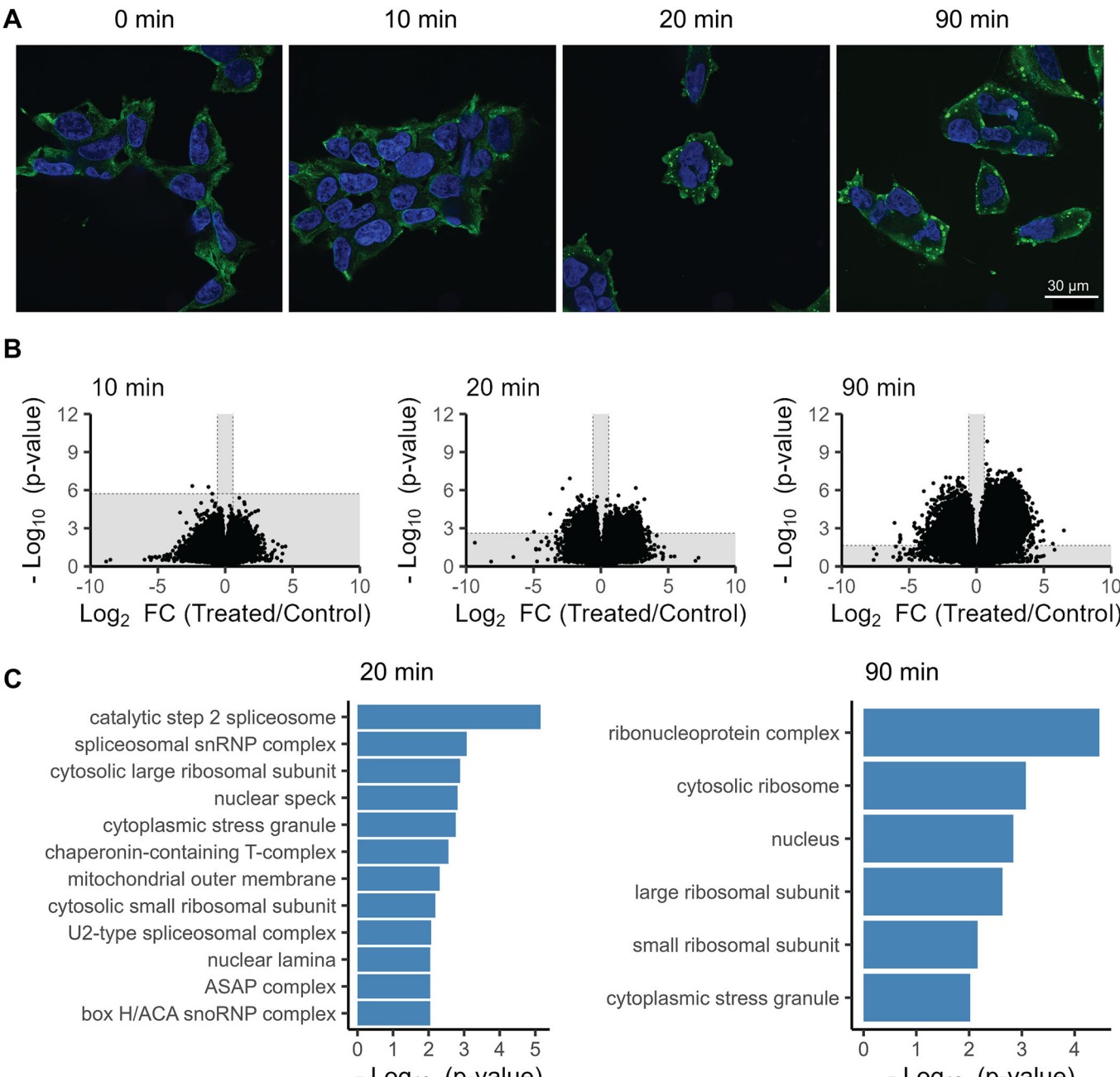

**Figure 5.    In-cell LiP-MS analysis of structural changes upon arsenite treatment of mammalian cells.**

(A) Representative micrographs of HEK293 cells stained with anti-G3BP1 antibody (green) and Hoechst (blue) at the indicated time points of sodium arsenite treatment. Scale bar, 30 µm. (B) Peptide intensities from HEK293 cells treated with sodium arsenite compared to untreated cells at 10, 20, and 90 min. Each data point represents a single peptide. The shaded gray region marks significance levels (FC > 1.5, two-sample unpaired t-test with Benjamini–Hochberg adjustment, q-value < 0.05, $n = 6$ biological replicates). (C) Gene ontology enrichment analysis (cellular component) of proteins showing structural changes upon arsenite treatment (p-value < 0.01, Fisher's exact test using the elim-algorithm in *topGO*). Each replicate was biologically independent, consisting of cells grown separately prior to treatment. Source data are available online for this figure.

and the spliceosome, which associates with nuclear speckles, are enriched (GO analysis) among proteins that show structural changes after 20 min of arsenite treatment. At this time point, we observed changes in 80 proteins associated with nuclear speckles, including 14 splicing factors, and the phosphatase PP1-gamma catalytic subunit, which has been reported to dynamically re-

localize between cytoplasm and nuclear speckles (Trinkle-Mulcahy et al, 2001). After 90 min, 161 nuclear speckle proteins showed evidence of structural changes, including the known marker protein SC35. Notably, fluorescence microscopy of arsenite-treated cells stained for the nuclear speckle marker SC35 showed that speckles became rounder upon sodium arsenite treatment in comparison to

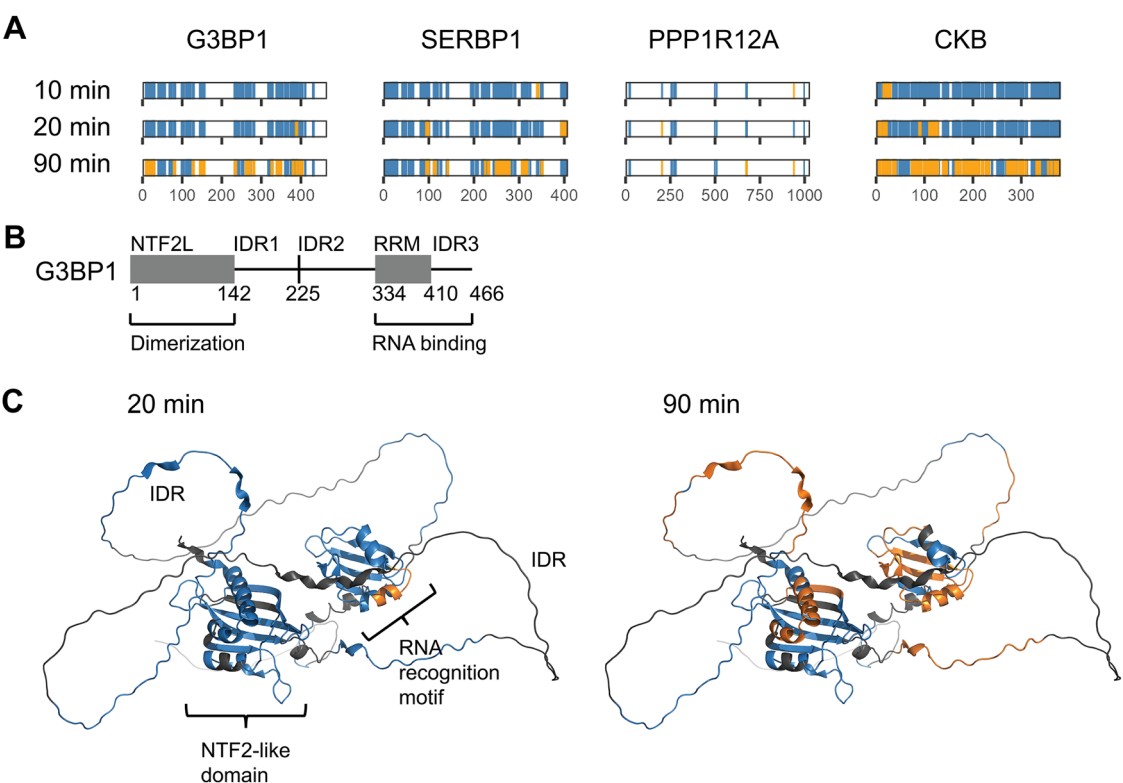

**Figure 6. Structural changes upon sodium arsenite treatment of HEK293 cells.**

(A) Barcode plots of protein structural changes upon arsenite treatment. The blue regions indicate peptides detected by mass spectrometry. Orange regions changed significantly at the indicated time (FC > 1.5, q-value < 0.05, $n = 6$ biological replicates). (B) Domains of G3BP1. NTF2L: NTF2-like domain; IDR: intrinsically disordered region; RRM: RNA recognition motif. (C) Alphafold structure of G3BP1. Blue regions were detected by mass spectrometry. Orange regions changed significantly at the indicated time. Each replicate was biologically independent, consisting of cells grown separately prior to treatment.

untreated controls (Fig. 8C,D). This effect is in line with changes previously reported under other stresses (Kim et al, 2019; Raina and Rao, 2022; Spector et al, 1991) and suggests alterations in the dynamics of this compartment. The structural alterations of nuclear speckle proteins we detected by in-cell LiP-MS may reflect these morphological changes in this phase-separated compartment.

Taken together, our data show that in-cell LiP-MS identified both known and novel alterations in proteins connected to biomolecular condensates within human cells. Further, we have used this approach to generate a compendium of proteins undergoing structural alterations upon arsenite stress, and to identify structurally altered regions in these proteins.

## Discussion

We present in-cell LiP-MS, a novel method to study proteome-scale structural changes in their native environment. Compared to LiP-MS in lysates, intracellular organization and molecular concentrations are preserved in this approach, enabling the study of changes in cellular physiology with minimal invasion. Since LiP-MS captures functional molecular events such as enzyme activity, protein-protein interaction, PTM and allosteric regulation, as we have previously shown (Cappelletti et al, 2021), the in-cell version of this technique should prove a powerful method for functional proteomics studies.

Although some existing structural proteomic methods can be implemented in intact cells, they face challenges. Most comparable to LiP-MS in principle are protein footprinting methods such as fast photochemical oxidation of proteins (FPOP) or covalent protein painting (CPP). FPOP reports on surface accessibility to hydroxyl radicals and can be applied intracellularly, with residue-level resolution and good proteome coverage (many hundreds or even thousands of proteins), but it requires a microfluidic setup and has so far not been reported to identify global protein structural changes between conditions (Espino et al, 2015; Kaur et al, 2020; Shortt et al, 2023). Covalent protein painting (CPP), which reports on lysine accessibility, also has good proteome coverage in cells (Bamberger et al, 2021, 2022), has recently been applied to identify accessibility changes in tissues of a mouse model of Alzheimer's disease (Son et al, 2024), and is a promising method for detecting protein structural changes in vivo or in situ. Crosslinking mass spectrometry applied to intact cells suffers from low proteome coverage. Thermal protein profiling (TPP) identifies proteins that change stability under conditions of interest but does not provide structural information about which region(s) of a protein may be altered (Savitski et al, 2014). Finally, in-cell LiP-MS has advantages over in-cell NMR, since LiP is applied on proteome-scale and does not require labeling but the two approaches are complementary since in-cell LiP-MS does not achieve the atomic resolution of NMR. Structurally altered proteins detected by in-cell LiP-MS in an unbiased manner could be followed up by in-cell NMR.

   

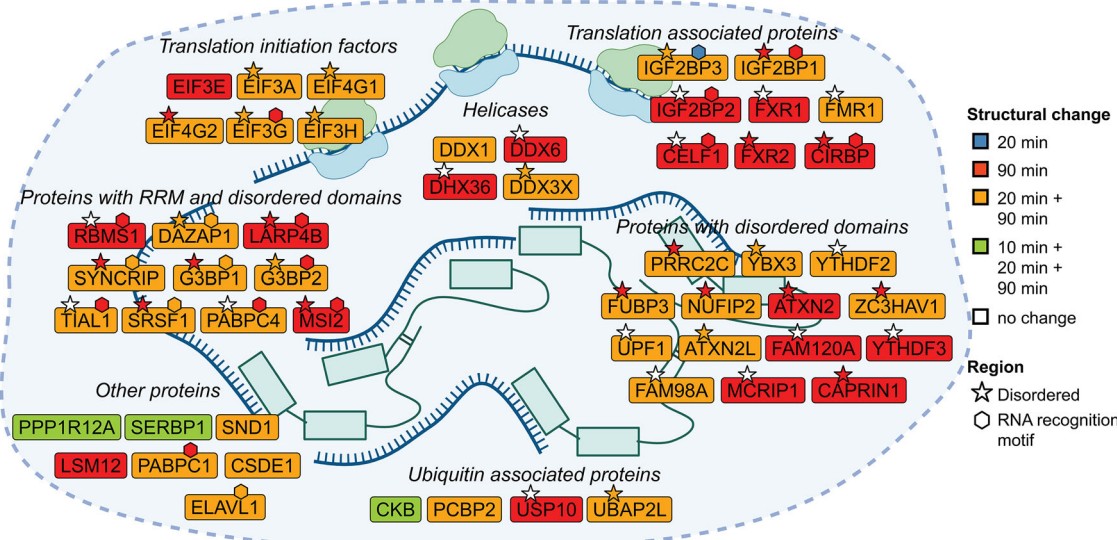

**Figure 7. Structural changes in stress granule proteins.**

Illustration of a stress granule at different times of sodium arsenite treatment. A star shape indicates at least one disordered region annotated in Uniprot, a hexagon at least one RNA recognition motif (RRM) annotated in Uniprot. Colored proteins change significantly in structure at the indicated time (FC > 1.5, q-value < 0.05, n = 6 biological replicates; blue: changes only at 20 min; red changes only at 90 min; green: changes at 10 min, 20 min, and 90 min; orange: changes at 20 min and 90 min). Colored stars and hexagons indicate that at least one significantly changing peptide is located in the indicated region. Depicted are proteins that are reported to be associated in RNAgranuleDB v2.0 with BioID or microscopy evidence on this website RNAgranuleDB v2.0 was further filtered for proteins that were detected in this study, and the fifty proteins with the highest score on RNAgranuleDB 2.0 were selected. Additionally, CKB, PPP1R12A, and SERBP1 are shown. Each replicate was biologically independent, consisting of cells grown separately prior to treatment.

We expect that our approach could be extended and improved in several ways in the future. In our current setup, we identified several thousands of proteins with mass spectrometric coverage equivalent to the classical method, but increasing this coverage could help to improve detection of structural changes. In applications where cells do not need to remain viable after electroporation, higher electroporation voltage and pulse number may lead to more uniform influx of PK and possibly to better sequence coverage. In-cell LiP-MS may also be adapted to other temperatures, pH, or different proteases. It may even be feasible to introduce a photocaged or otherwise inactivated protease by electroporation, such that the start of proteolysis can be more tightly controlled. Organelle-specific in-cell LiP-MS could be achieved by targeting the protease to different intracellular locations to detect organelle-specific structural states. Further, in-cell LiP-MS could be used not only in cell lines, but may be applicable also to tissues after dissociation and to unicellular organisms such as bacteria or yeast. To extend in-cell LiP-MS to different cell types or electroporation settings, we recommend first screening for conditions that yield good PK internalization using the FRET reporter, and then optimizing the conditions using the % half-tryptic peptides or the number of PK cleavages in the proteome as a readout. The reporter may also be used to test whether experimental conditions affect PK activity which, as in classical LiP-MS, can affect the interpretation of an experiment.

We have optimized in-cell LiP-MS conditions to preserve the native state of cells as far as possible, and note that our conditions are gentler than the 1400 V commonly used for electroporation in structural studies based on in-cell NMR. While our LiP-MS data show that most proteins remain in their native conformation under these conditions, we cannot exclude that electroporation may affect the structure of individual proteins. In addition, care should be taken that cells do not clump (i.e., form clusters) during electroporation in the presence of extracellular PK, which may be caused by partial proteolysis of surface proteins. Clusters occur if cells are brought in close proximity by pipetting or centrifugation. They can be avoided if the added PK is ice-cold, electroporation is performed within seconds after mixing PK, and cells are well resuspended prior to addition of PK. Good cell resuspension to obtain single cells and avoidance of cell clusters also results in higher PK uptake. Mixing after protease addition should be performed within seconds and further mixing should be avoided.

In classical LiP-MS on lysates, optimal coverage has been found for 1 µg PK added to 100 µg substrate protein. For in-cell LiP-MS, we obtained the highest coverage by electroporating 3 million HEK293 cells with 12 nmol PK, of which 36.6 pmol PK were estimated to be intracellular. If sufficient attention is paid to experimental details (i.e., avoiding cell clumping and including more replicates), reproducibility is comparable or only slightly higher in classical compared to in-cell LiP. Nevertheless, the system is necessarily more heterogenous in the in-cell context. In lysates, an almost homogeneous mixture of proteins, PK and any other added perturbant can be achieved, whereas in-cell LiP analyzes a mixture of cells which may be in different cellular states, and PK uptake varies. We compensated for this in part by using more replicates in the in-cell method. Since the variability in PK uptake is expected to reduce the LiP signal (i.e., fold changes between conditions of interest), more homogenous delivery methods should not only further reduce variability but also improve sensitivity of the in-cell method. Because exact timing matters in LiP, we have so

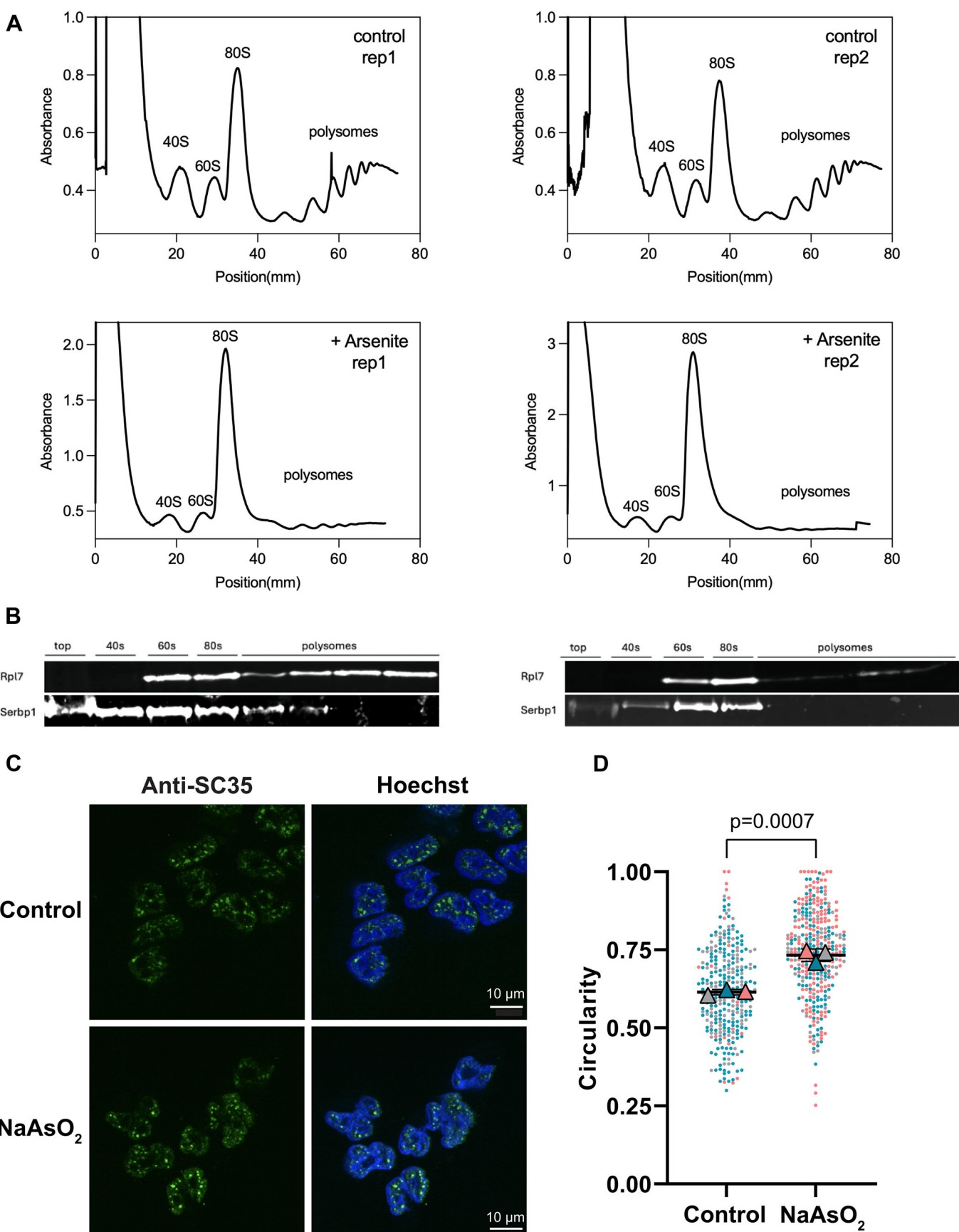

**Figure 8.  Validation of structural changes in arsenite stress.**

(A, B) SERBP1 is associated with polysome disassembly after 10 min of arsenite treatment. (A) Polysome profiling of HEK293 with or without 10 min arsenite treatment (2 biological replicates). (B) Western blots for SERBP1 and the ribosomal protein Rpl7 in selected fractions. Left blot, control; right blot, after 10 min of arsenite treatment. (C, D) Nuclear speckles become rounder upon sodium arsenite treatment. (C) Representative images of HEK293 cells stained with anti-SC35 antibody and Hoechst after 90 min treatment with sodium arsenite. Scale bars, 10 µm. (D) Circularity of nuclear speckles with and without sodium arsenite treatment measured for three biological replicates. Colors correspond to replicates. The mean of all replicates (horizontal lines) was compared (mean of control: 0.604; mean of sodium arsenite treatment: 0.741; error bars indicate the standard deviation; two-tailed t test, ***$p$-value = 0.0007). Each replicate was biologically independent, consisting of cells grown separately prior to treatment. Source data are available online for this figure.

far been able to treat and process up to eight lysates simultaneously. In our current setup, the electroporation equipment limits us to electroporating one sample at a time and thus we anticipate that increasing the throughput of this step would improve reproducibility even further. We show that in-cell LiP-MS captures the specific interaction between FKBP1A and rapamycin, using this interaction as a test case because we have previously used it to characterize classical LiP-MS (Malinovska et al, 2023). It should be noted, however, that classical LiP-MS is in general much better suited for target identification than in-cell LiP because cell lysis should minimize downstream effects of the drug, therefore introducing fewer confounding factors and facilitating interpretation. In-cell LiP-MS is more suited to capture complex biological processes like the stress response, which may depend on intact cellular organization. As such, in-cell LiP-MS should also have the ability to identify downstream effects of small molecules, which may be distinguished from target-engagement effects with time-course experiments. This opens the exciting possibility that the approach could be used to identify targeted pathways, downstream events, and off-target effects within intact cells under specific conditions of interest, e.g., hypoxia. Dose- and time-resolved drug profiling of post-translational modifications (PTMs) has been used to elucidate the mode of action of drugs in cells and to assign PTMs to pathways (Zecha et al, 2023). In-cell LiP-MS could also help to elucidate effects of mutations in pathways affected by drug binding, as well as the effect of two or more drugs added simultaneously or sequentially to cells.

Application of in-cell LiP-MS to arsenite-stressed cells provides, to our knowledge, the first in situ data on structural rearrangements of stress granule proteins. These dynamic structures have typically been studied by fluorescence microscopy, which lacks structural resolution, or by in vitro reconstitution, but until now there have been no methods to comprehensively study protein structural changes in large dynamic assemblies. Stress granule proteins, including the scaffolding protein G3BP1, were enriched in the set of structurally altered proteins upon arsenite stress, suggesting that the dataset may be used to generate hypotheses about so far unknown components or regulators of stress granules. Information on the specific protein regions involved, for instance the RNA-binding and intrinsically-disordered regions detected for G3BP1, may shed light on mechanisms of stress granule formation. Overall, these data provide an unprecedented, peptide-resolution resource on proteome-scale and time-resolved structural changes in human cells under arsenite stress.

Of special interest are the structural changes in CKB, PPP1R12A and SERBP1, detected even before stress granules are visible by fluorescence microscopy. The enzyme creatine kinase B-type (CKB) plays an important role in energy transfer and homeostasis; our data suggest that it might be involved in granule formation and this will

need to be validated in future work. The protein phosphatase 1 regulatory subunit 12A (PPP1R12A) is involved in various cellular processes including muscle contraction and cell cycle regulation, where it antagonizes the Polo-like kinase 1 (PLK1) (Yamashiro et al, 2008). PPP1R12A has been detected as part of stress granules (Yang et al, 2020) but its functional role is unknown; our data suggest that it may play a role in the early stages of stress granule formation and pinpoints the regions involved. The SERPINE1 mRNA-binding protein 1 (SERBP1), is known to interact with stress-granule proteins upon arsenite treatment in an RNA-dependent manner (Lee et al, 2014) and is essential for stress granule clearance after arsenite stress (Wang et al, 2023). Our data suggests that SERBP1 also plays a role early in the stress response, possibly by binding to mRNA-free ribosomes (Shetty et al, 2023).

Similarly, our application of in-cell LiP-MS has implicated nuclear speckles, dynamic assemblies of splicing factors, in the response to arsenite stress. In line with this, we observed morphological changes in these compartments with fluorescence microscopy, which have previously been seen upon heat shock (Spector et al, 1991), upon inhibition of transcription (Kim et al, 2023), and upon inhibition of the DEAD-box ATPase UAP56 (Hondele et al, 2019). Further analysis of the set of structurally altered proteins may provide insight into mechanisms leading to the altered morphology of nuclear speckles and the molecular players involved. Interestingly, a GO analysis restricted to proteins with only the most significantly changing peptides after 90 min of sodium arsenite treatment (FC > 1.5, q-value < 0.0005, Appendix Fig. S5A–C) revealed involvement of a broad range of cellular stress-related processes (e.g., response to toxic substance, plasma membrane repair), molecular functions (e.g., binding of unfolded proteins, de-ubiquitination, and the binding of chaperones), and cellular compartments (e.g., proteasome, spliceosome, mitochondrial outer membrane, secretory granules), which can be explored in future studies.

We envision a broad range of applications of in-cell LiP in the field of condensates. For example, our data may be used to identify LiP marker peptides for stress granule formation, potentially including markers that probe the involvement of the RNA binding and IDP regions for each protein. Similar markers could be identified for the formation of other physiological molecular condensates (such as Cajal bodies or P-bodies) or metabolons (e.g., the purinosome) and even for pathological condensates (e.g., from amyloidogenic proteins such as FUS or TDP43). These markers, potentially measured simultaneously with targeted MS, could be used to study the detailed dynamics of condensates within the cell, identify perturbations that influence them, and study potential cross-talk. One may envision using such markers to derive in cell phase diagrams of condensates. Beyond these assemblies, in-cell LiP-MS should prove useful in characterization of the global cellular response to any perturbation, for instance to changes in pH

as seen in the tumor microenvironment or to environmental factors such as toxins or irradiation.

In summary, we have shown that in-cell LiP-MS can detect proteome-wide structural changes upon a perturbation of interest, within the native environment of the intact cell and at peptide resolution. Since protein structural changes are likely to be ubiquitous under perturbation or stimulus of a biological system, in-cell LiP-MS has the potential to provide insight into numerous cellular processes including labile assemblies formed by phase separation that could so far not be addressed with established methods.

# Methods

### Reagents and tools table

| Reagent/Resource | Reference or Source | Identifier or Catalog Number |
|---|---|---|
| **Experimental models** | | |
| Flp-In T-Rex 293 cells | Thermo Fisher | R78007 |
| HEK293 cells | ATCC | CRL-1573 |
| **Recombinant DNA** | | |
| ECFP-TevS-YPET | Addgene | 100097 |
| **Antibodies** | | |
| Alexa Fluor 488, anti-Mouse | Life Technologies | A-11029 |
| Alexa Fluor 680, anti-Mouse | Thermo Fisher | A-21057 |
| Anti- Serbp1/pai-rbp1 | Santa Cruz Biotechnology | 376832 |
| Anti-Akt | CST | 2920 |
| Anti-G3BP1 | Santa Cruz Biotechnology | 365338 |
| Anti-phospho-Akt Ser473 | CST | 4060 |
| Anti-phospho-S6 Ribosomal Protein Ser240/244 | CST | 2215 |
| Anti-Rpl7 | Thermo Fisher | 14583-1-AP |
| Anti-S6 Ribosomal Protein | CST | 2317 |
| Anti-SC35 | Sigma-Aldrich | S4045 |
| IRDye® 800CW, anti-Rabbit | LI-COR Biosciences | 926-32211 |
| IRDye® 800CW, donkey anti-rabbit | LI-COR Biosciences | 926-32213 |
| **Peptides** | | |
| iRT peptides | Biognosys | Ki-3002-2 |
| NYSPASEPSV[C-carbamidomethyl]TVGASD(R) | Thermo Scientific | AQUA Ultimate Heavy |
| TQLFGV(K) | Thermo Scientific | AQUA Ultimate Heavy |
| TYYYSS(R) | Thermo Scientific | AQUA Ultimate Heavy |
| TYYYSS(R) | Thermo Scientific | AQUA Ultimate Heavy |
| **Chemicals, Enzymes and other reagents** | | |
| 2-Mercaptoethanol | Applichem | A1108 |
| Acetone | Sigma-Aldrich | 179973 |

| Reagent/Resource | Reference or Source | Identifier or Catalog Number |
|---|---|---|
| Acetonitrile (ACN), LC-MS grade | ROTISOLV | AE70.1 |
| Ammonium bicarbonate | Sigma-Aldrich | 9830 |
| Antimycin A | Sigma-Aldrich | A8674 |
| Bicinchoninic acid assay | Pierce | 23228 |
| Bradford assay | Biorad | 5000201 |
| Bromophenol blue | Acros Organics | 15134 |
| BSA | Lucerna-Chem | P06-1391050 |
| Cy5 | Tocris | 5436 |
| Cycloheximide | Sigma-Aldrich | C7698 |
| Dimethyl solfoxide (DMSO) | Sigma-Aldrich | 472301 |
| DMEM | Gibco Life Technologies | 41965039 |
| Ethylenediaminetetraacetic acid (EDTA) | Invitrogen, Applichem (Quantitative immunoblots) | AM9260G, A2937 |
| Fetal bovine serum | BioConcept | 2-01F16-I |
| Formic acid (FA) | Sigma-Aldrich/ Merck | 64-18-6 |
| Glycerol | Applichem | A2926 |
| Guanidinium hydrochloride | Sigma-Aldrich | G3272 |
| Halt Protease and Phosphatase Inhibitor Cocktail | Thermo Fisher | 78441 |
| HEPES (4-(2-hydroxyethyl)-1-piperazineethanesulfonic acid) | Sigma-Aldrich | H4034 |
| Hoechst 33258 | Molecular Probes | H-3569 |
| Iodoacetamide | Sigma-Aldrich | I1149 |
| JetPrime buffer and reagent | Polyplus | 101000046 |
| Lys-C | Wako | 121-05063 |
| Magnesium chloride hexahydrate | Fluka | 63072 |
| Milk powder | Rapilait | 101782 |
| NP-40 | Sigma-Aldrich | NP40S |
| NuPAGE™ LDS Sample Buffer | Invitrogen | NP0007 |
| Oligomycin | Sigma-Aldrich | O4876 |
| Paraformaldehyde (PFA) | Electron Microscopy Sciences | 15714 |
| PBS pH 7.4 | Gibco Life Technologies | 10010015 |
| Pefabloc | Roche | 11429868001 |
| Penicillin/Streptomycin | Gibco Life Technologies | 10378016 |
| Pitstop 2 | Sigma-Aldrich | SML1169 |
| Potassium chloride | Merck | K41042236-032 |
| Proteinase K | Sigma-Aldrich | P2308 |

| Reagent/Resource | Reference or Source | Identifier or Catalog Number |
|---|---|---|
| Rapamycin | Thermo Fisher | J62473 |
| Roche cOmplete EDTA free inhibitor cocktail | Roche | 11873580001 |
| Rotenone | Sigma-Aldrich | R8875 |
| Seahorse XF Real-Time ATP Rate Assay Kit | Agilent | Kit 103592-100 |
| Sodium arsenite | Pfaltz & Bauer | 7784-46-5 |
| Sodium bicarbonate | Sigma-Aldrich | S8875 |
| Sodium chloride | Acos Organics | 32730 |
| Sodium deoxycholate | Sigma-Aldrich | D6750 |
| Sodium dodecyl sulfate (SDS) | Roth | 8029.4 |
| Trans-Blot Turbo transfer buffer | Biorad | 10026938 |
| Trichloroacetic acid | Sigma-Aldrich | T6399 |
| Tris-HCl | Invitrogen, Sigma-Aldrich (Quantitative Immunoblots) | AM9855G, T5941 |
| Tris(2-carboxyethyl) phosphine hydrochloride) (TCEP) | Pierce | 20490 |
| Triton X-100 | Sigma-Aldrich | T8787 |
| Trypan blue solution and counting chamber slides | Invitrogen | C10312 |
| Trypsin | Promega | V5113 |
| Tween 20 | Sigma-Aldrich | P1379 |
| Urea | Sigma-Aldrich/Merck | 1084871000 |
| **Software** | | |
| Alphafold 2 | (Tunyasuvunakool et al, 2021) | v.4 |
| GraphPad Prism | | v.10 |
| imageJ | Fiji | v.2.9.0 |
| ImageStudio Lite | | v.5.2 |
| PyMOL | https://www.pymol.org/ | v.2.5.4 |
| R | https://www.r-project.org/ | v.4.2.2 |
| Spectronaut | Biognosys | v.17 |
| topGO R package | https://github.com/federicomarini/topGO | v.2.50.0 |
| **Other** | | |
| 10% Gels | Bio-Rad | 4561036 |
| 40 cm × 0.75 mm i.d. column | New Objective | PF360-75-10-N-5 |
| BioPureSPN MACRO | The Nest Group | HMM S18V |
| CLARIOstar Plus plate reader | BMG Labtech | |
| Countess 3 Cell Counter | Thermo Fisher | |

| Reagent/Resource | Reference or Source | Identifier or Catalog Number |
|---|---|---|
| EASY-nLC 1000 | Thermo Scientific | LC120 |
| EASY-nLC 1200 | Thermo Scientific | LC140 |
| iBlot 2 Gel Transfer Device | Thermo Fisher | |
| Kimble Pellet Pestle | DKW Life Sciences | 749540-0000, 749515-0000 |
| nanoAQUITY UPLC | Waters | 176016000 |
| Neon Transfection System | Thermo Fisher | 10431915 / Invitrogen MPK5000 |
| Nitrocellularose membranes | Thermo Fisher | IB23001 |
| NuPAGE gel | Invitrogen | |
| Odyssey Classic instrument | Li-Cor | |
| Orbitrap Eclipse Tribrid mass spectrometer | Thermo Scientific | FSN04-10000 |
| Orbitrap Exploris 480 mass spectrometer | Thermo Scientific | BRE725533 |
| Orbitrap Fusion Lumos Tribrid mass spectrometer | Thermo Scientific | FETD2-10001 |
| PD-10 desalting column | Cytiva | |
| Q Exactive Plus Hybrid Quadrupole-Orbitrap mass spectrometer | Thermo Scientific | IQLAAEGAAPFALGMBDK |
| ReproSil-Pur 120 C18-Aqua, 3 µm | Dr. Maisch GmbH | r13.aq.0003 |
| Seahorse XFe96 Analyzer | Agilent | |
| Spinning disk confocal microscope | Nikon | Eclipse Ti2 with Yokogawa Confocal Scanner Unit CSU-W1-T2 and Hamamtasu Orca Fusion BT camera |
| SW41rotor | Beckmann | |
| Trans-Blot | BIO-RAD | |
| Ultrasonic generator, Ultrasonic transducer, VialTweeter | Hielscher | UP200St-G, UP200St, S26d11x10 |
| VanquishNeo System | Thermo Scientific | VN-S10-A-01 |
| µ-slide manually coated with poly-L-lysine | Sigma-Aldrich | P8920 |
| µ-Slides with ibiTreat surface | ibidi | 80806 |

## Cell culture conditions

HEK293 cells (ATCC #CRL-1573) were cultured in Dulbecco's modified Eagle's medium (DMEM, Gibco Life Technologies #41965039) supplemented with 10% heat inactivated fetal bovine serum and 5% Penicillin/Streptomycin at 37 °C and 5% $CO_2$ in a humidified environment. HEK293 cells were passaged prior to confluency in phosphate-buffered saline (PBS) pH 7.4 (Gibco #10010015). Flp-In T-Rex 293 cells (Thermo Fisher R78007) were cultured under the same conditions. To measure cell viablility and concentration, cells were mixed 1:1 with trypan blue solution and analyzed in a Countess Cell Counter (Thermo Fisher Scientific).

The cell lines were recently tested for mycoplasma contamination. Cell lines were not authenticated.

## Cell transfection

HEK239 cells were transfected at ~70% confluency on 150 mm culture dishes. Medium was exchanged prior to transfection. Per plate, 1200 µl jetPrime buffer were mixed with 25 µg plasmid DNA (ECFP-TevS-YPET (Gray et al, 2010), a gift from Charlie Morgan and Jason Chin; Addgene #100097), and 50 µl jetPrime reagent (Polyplus #101000046). After incubation for 10 min, the mixture was added to the cells.

## Fluorescence-based PK activity assay

### For cell lysate

Flp-In T-Rex 293 cells were detached from culture dishes 2–3 days after transfection, washed with PBS and lysed using a tissue grinder (DKW Life Sciences Kimble Pellet Pestle) in ten cycles of 10 s of homogenization and 1 min pause on ice. The lysate was cleared by centrifugation at $10,000 \times g$ at 4 °C for 5 min. The supernatant was diluted to a protein concentration of 2 µg/µl, and 50 µl lysate per replicate were transferred to a 96-well plate. At the start of fluorescence measurement, 5 µl of a 100 µM PK solution (solubilized in water; in one condition electroporated with one 25 ms pulse at 1000 V), or water were added. Fluorescence of ECFP (430-10/480) and FRET to YPet (430-10/530-10) were measured using a CLARIOstar Plus plate reader (BMG Labtech).

### For intact cells

HEK293 cells were detached from culture dishes 2–3 days after transfection, washed with PBS, and resuspended in PBS at a concentration of $33.3 \times 10^6$ cells/ml. Prior to fluorescence analysis, 90 µl cell suspension (corresponding to $10^6$ cells) were mixed with 30 µl water or 30 µl of a 400 µM PK solution PK (solubilized in water). For electroporation, 100 µl of the solution were electroporated with one 25 ms pulse at 1000 V using a Neon Transfection System (Thermo Fisher Scientific), and transferred to a 96-well plate. Control cells without electroporation were directly transferred after mixing PK and cells. Fluorescence measurements were performed with the same settings as for cell lysates.

## Determining the variability of PK electroporation

400 µM PK (Sigma #P2308 dissolved in water) was incubated 1:2 with Cy5 (Tocris #5436) in sodium bicarbonate pH 8.25 for 2 h at room temperature. The reaction was stopped by adding Tris-HCl pH 7.5 at a final concentration of 100 mM. Unbound Cy5 was removed 3 times with a PD-10 desalting column (Cytiva). PK was inactivated by incubation overnight at room temperature in 5 mM Pefabloc (Roche #11429868001). Inactivated Cy5-PK was mixed with 3 million HEK293 cells in PBS (final PK concentration 100 µM), and electroporated with a Neon Transfection System (Thermo Fisher Scientific; 1 pulse of 25 ms at 1000 V). The electroporated cells were transferred on an ibidi µ-slide coated with poly-L-lysine and spun down at $2000 \times g$ for 2 min. The cells were washed 3 times with PBS and imaged with a Nikon spinning disk confocal microscope. For quantification the average signal intensity in the Cy5 channel was used from manually segmented cells.

## Rapamycin treatment of cells

For mass spectrometry analysis, HEK293 cells were detached from culture dishes, washed with PBS and resuspended in PBS at a concentration of $40 \times 10^6$ cells/ml. For treatment, 75 µl cell suspension per replicate (corresponding to $10^6$ cells) were mixed with 15 µl rapamycin solution to a final concentration of 20 µM rapamycin (rapamycin dissolved in DMSO and PBS; 1% DMSO concentration on cells in the 20 µM rapamycin experiment). After incubation for 10 min, cells were processed according to the in-cell LiP-MS workflow. For the analysis of organelle and domain coverage, cells were treated with 0.016% DMSO for 5 min prior to in-cell LiP-MS.

For quantitative immunoblots, HEK293T cells seeded in 12-well plates were treated with 100 nM or 250 nM Rapamycin (Thermo Fisher, J62473) at 70–80% confluency for indicated time points in normal growth medium.

## Sodium arsenite treatment of cells

HEK293 cells were grown in separate dishes for each replicate. Sodium arsenite dissolved in water was added to the medium in a final concentration of 500 µM. After 10 min, 20 min, and 90 min, the medium was removed, cells washed in PBS, and pelleted by centrifugation at $200 \times g$ for 1 min. Cells were resuspended in PBS. Part of the sample was snap-frozen in liquid nitrogen for analysis of protein abundances. For in-cell LiP, 90 µl containing ~$10^6$ cells were processed according to the in-cell LiP-MS workflow. No blinding was done.

## Quantitative immunoblots

Cell lysis was done by direct addition of RIPA buffer [50 mM Tris-HCl (pH 8.0), 150 mM NaCl, 2 mM EDTA, 1% NP-40, 0.5% sodium deoxycholate, 0.1% SDS with 1X Halt Protease and Phosphatase Inhibitor Cocktail (Thermo Fisher, 78441)] into the wells. 10 µg total protein from the cleared lysate was mixed with 6X loading buffer [375 mM Tris-HCl (pH 7.4), 50% Glycerol, 6% SDS, 30% 2-Mercaptoethanol, 0.04% Bromophenol blue], incubated at 95 °C for 5 min, resolved on 10% gels (Bio-Rad, 4561036) and transferred to Nitrocellulose membranes (Thermo Fisher, IB23001) using iBlot 2 Gel Transfer Device (Thermo Fisher). Blocking was done in 5% BSA/TBS-T and membranes were incubated overnight at 4 °C in primary antibody solutions against phospho-S6 Ribosomal Protein Ser240/244 (CST, #2215), S6 Ribosomal Protein (CST, #2317), phospho-Akt Ser473 (CST, #4060), and Akt (CST, #2920). Secondary antibody incubation was performed for 1 h at room temperature in dark using donkey anti-mouse IRDye® 680RD (Li-Cor, 926-68072) and donkey anti-rabbit IRDye® 800CW (Li-Cor, 926-32213). Membranes were visualized with Li-Cor Odyssey Classic instrument and signal quantification was performed using ImageStudio software. Statistical analyses were performed with GraphPad Prism v.10 software.

## In-cell LiP-MS workflow

For in-cell LiP-MS, 90 µl of treated or untreated cells (~$3 \times 10^6$ cells) were mixed with 30 µl ice-cold PK solution (100 µM final PK concentration except in optimization experiments, PK dissolved in

water). To determine the effect of PK on cells, 30 µl water were added instead. To determine the effect of electroporation independent of PK, PBS was added instead.

For electroporation, 100 µl of the cell suspension were electroporated using Neon Transfection System (Thermo Fisher Scientific). Electroporation was usually carried out with one 25 ms pulse at 1000 V, except for optimization experiments when 1–3 pulses and 800 V to 1400 V were tested. Exactly 15 s after electroporation, samples were heated to 37 °C, and incubated for 2 min (during optimization, 1 min to 5 min incubation were tested). For washing, 1 ml PBS pre-warmed to 37 °C was added, and cells were pelleted by centrifugation for 1 min at $4000 \times g$. The supernatant was removed, and cells were washed a second time with 1 ml PBS. For protease inactivation, 100 µl of 7 M guanidinium hydrochloride dissolved in water was added to the cells, and the samples were heated to 98 °C for at least 5 min. Samples were snap-frozen in liquid nitrogen before cell lysis. We used guanidinium hydrochloride for protease inactivation since we determined that sodium deoxycholate, which is used for this purpose in classical LiP-MS, does not sufficiently quench proteinase K activity within cells. However, since trypsin digestion is known to be less effective in buffers containing guanidine hydrochloride (Proc et al, 2010), this has the consequence that more missed cleavages are observed in in-cell versus classical LiP.

## Quantification of the cell supernatant

Approximately $3 \times 10^6$ HEK293 cells in 120 µl PBS were electroporated with one 25 ms pulse at 1000 V, transferred to 37 °C after 15 s, incubated 2 min, and pelleted by centrifugation at $4000 \times g$. The supernatant of non-electroporated cells served as control. Part of the supernatant was retained for a Bradford assay and the rest frozen at $-80$ °C. For mass spectrometry analysis, the frozen supernatant was diluted with an equal volume of 10% sodium deoxycholate, followed by tryptic digestion.

## Protease uptake with endocytosis inhibitor

Approximately $3 \times 10^6$ HEK293 cells in 75 µl PBS were incubated with 15 µl of the endocytosis inhibitor Pitstop 2 (Sigma #SML1169, dissolved in PBS and DMSO; final 0.1% DMSO on cells) for 15 min (final concentration 30 µM Pitstop 2). Control cells in PBS were treated with 0.1% DMSO. Cells were mixed with 30 µl PK solution (or water) to a final concentration of 100 µM PK. After 15 s, samples were transferred to 37 °C, incubated for 2 min, and washed twice with 1 ml PBS pre-warmed to 37 °C was by centrifugation for 1 min at $4000 \times g$. 100 µl of 7 M guanidinium hydrochloride dissolved in water was added to the cells. Samples were heated to 98 °C for 10 min, and snap-frozen in liquid nitrogen. Cells were further lysed and processed like in-cell LiP-MS samples.

## Effects of electroporation on protein expression

120 µl of HEK293 cells (~$3 \times 10^6$ cells) were resuspended in PBS, and 100 µl of the cell suspension were electroporated using Neon Transfection System (Thermo Fisher Scientific). Electroporation was carried out with one 25 ms pulse at 1000 V, and cells were transferred into 4 ml of Dulbecco's modified Eagle's medium (DMEM, Gibco Life Technologies #41965039) supplemented with 10% heat inactivated fetal bovine serum and 5% Penicillin/Streptomycin. Cells were cultured for 4 h at 37 °C and 5% $CO_2$ in a humidified environment. Cells were harvested by centrifugation ($160 \times g$, 2 min), washed in PBS, and snap-frozen in liquid nitrogen. Cell pellets were resuspended in lysis buffer (8 M Urea, 100 mM ammonium bicarbonate, 5 mM EDTA, pH 8), and sonicated two times 30 s using a Hielscher UP200St-G-Ultrasonic generator coupled to a UP200St with VialTweeter (60% cycle, 60% amplitude). The lysate was cleared by centrifugation at 4 °C (5 min at $20,000 \times g$). Protein concentration was measured using a bicinchoninic acid assay and evenly adjusted. Samples were then subjected to tryptic digestion.

## Seahorse assay

Cells were seeded in Seahorse XF96 microplates at a density of $5 \times 10^4$ cells per well. HEK293 cells (~$3 \times 10^5$ cells) were resuspended in 120 µl PBS, and 100 µl of the cell suspension were electroporated using a Neon Transfection System using one 25 ms pulse at 1000 V. Oxygen consumption rate (OCR) was measured using the Agilent Seahorse XF Real-Time ATP Rate Assay Kit 24 h post electroporation, following the manufacturer's protocol. Mitochondrial respiration was measured in real-time by sequentially adding 1.5 µM oligomycin (Sigma, O4876) and a mixture of 0.5 µM rotenone and antimycin A (Rot/AA) (Sigma R8875/A8674). The reagents and the Seahorse XFe96 Analyzer were kindly provided by the Flow Cytometry Core Facility of ETH Zurich.

## Polysome profiling and Western blotting

The cell lysate was loaded onto a 10–50% sucrose gradient in: 20 mM Tris-HCl pH 8.0, 140 mM KCl, 6 mM MgCl$_2$, 100 µg/ml cycloheximide, 0.1% Tween, and EDTA-free protease inhibitors cocktail, followed by centrifugation for 2.5 h at 35,000 rpm and 4 °C using a Beckmann SW41rotor, followed by fractionation. To precipitate proteins from each fraction, Triton X-100 was first added to final concentration of 1.6%, followed by addition of Trichloroacetic acid (TCA) to a final concentration of 25%, and centrifugation at $16,000 \times g$ for 30 s, and the upper phase was discarded. To the lower phase, 1 ml of ice-cold acetone was added, followed by vortexing and centrifugation to collect the protein pellet. The pellet was washed with 1 ml of acetone, followed by drying the pellet, resuspending in NuPAGE™ LDS Sample Buffer (Invitrogen), and boiled for 95 degrees for 5 min, before loading on NuPAGE gel (Invitrogen). Proteins were transferred to nitrocellulose membrane using BIO-RAD Trans-Blot™, in Trans-Blot Turbo transfer buffer. The membrane was blocked in 5% milk powder in PBST (0.1% Tween in PBS) for 1 h, followed by incubation for 1 h in 5% milk powder in PBST with the primary antibodies: Serbp1/pai-rbp1 antibody (Santa Cruz Biotechnology, Catalog # sc-376832) at 1:500 dilution, and Rpl7 antibody (Thermofisher, Catalog # 14583-1-AP) at 1:500 dilution. The membrane was washed three times in PBST, before incubation for 1 h in 5% milk powder in PBST with the secondary antibodies: anti-mouse IgG coupled to Alexa 680 (Thermofisher, Catalog # A-21057) at 1:10,000 dilution, and anti-rabbit IgG coupled to IRDye® 800CW (LI-COR Biosciences GmbH Catalog # P/N 926-32211) at 1:10,000 dilution, followed by washing three times in PBST, and imaging.

## Cell lysis

### In-cell LiP-MS

Samples were thawed on ice, and sonicated two times 30 s using a Hielscher UP200St-G-Ultrasonic generator coupled to a UP200St with VialTweeter (optimization experiments and 10 μM rapamycin experiment: 60% cycle, 100% amplitude; stress granule experiment, experiment for the analysis of organelle and domain coverage: 170 W power, 60% cycle). Lysates were cleared by centrifugation at $10,000 \times g$ for 5 min. Protein concentration was measured using a bicinchoninic acid assay and evenly adjusted. Samples were then subjected to tryptic digestion.

### Measuring protein abundances

To measure protein abundances after arsenite treatment, cell pellets were thawed on ice, and 8 M urea buffer supplemented with protease inhibitor cocktail (Roche) was added. Cells were sonicated like the in-cell LiP-MS samples of the stress granule experiment, protein concentration was measured using a bicinchoninic acid assay and evenly adjusted. Samples were then subjected to tryptic digestion.

### Classical LiP-MS

Cell pellets of HEK293 cells were resuspended in a buffer consisting of 1 mM $MgCl_2$, 150 mM KCl, 100 mM HEPES pH 7.4. Cells were lysed using a tissue grinder (DKW Life Sciences Kimble Pellet Pestle) in ten cycles of 10 s of homogenization and 1 min pause on ice. The lysate was cleared by centrifugation at $10,000 \times g$ at 4 °C for 15 min. The supernatant was diluted to a protein concentration of 2 μg/μl and treated with rapamycin.

## Classical LiP-MS workflow

For classical LiP-MS of rapamycin treatment, 50 μl native lysate of 2 μg/μl protein were incubated with 1 μl rapamycin (dissolved in DMSO), or 1 μl DMSO for 5 min at 25 °C. To start limited proteolysis, 5 μl PK (0.2 μg / μl in water) were added for 5 min. For protease inactivation, samples were heated to 99 °C for 5 min, then cooled to 4 °C for 5 min. Finally, an equal volume of a 10% sodium deoxycholate solution was added. Samples were subjected to tryptic digestion.

A similar ratio of protein substrate and PK was used in both standard LiP-MS and in-cell LiP-MS: i.e., 100 μg protein and 35 pmol PK (1 μg) in the lysate and 3 million cells (800–1000 μg protein) and 12 nmol (347 μg) PK for in-cell LiP.

## Tryptic digestion

Disulfide bonds were reduced by incubation with 5 mM tris(2-carboxyethyl)phosphine hydrochloride) (TCEP) shaking for 40 min at 37 °C. Subsequently, samples incubated 30 min in the dark at room temperature with 40 mM. Samples were then diluted using 0.1 M ammonium bicarbonate (Sigma-Aldrich) to achieve a final guanidinium chloride concentration of 0.5 M, or 1% sodium deoxycholate, or <2 M urea. Proteins were digested with Lys-C (FUJIFILM Wako Pure Chemical Corporation) and trypsin (Promega) at an enzyme substrate ratio of 1:100 shaking overnight at 37 °C. The tryptic digest was stopped by adding formic acid to a final pH of ~2.

## C18-Cleanup and MS sample preparation

A 96-well C18 Micro-Spin plate (The Nest Group, 40–400 μg) was washed with 200 μl methanol, followed by 100 μl buffer B (50% acetonitrile (ACN), 0.1% FA), and $2 \times 200$ μl buffer A (0.1% FA). Samples were loaded on the plate and washed with $3 \times 200$ μl buffer A by centrifugation at $1000 \times g$ for 1 min. Peptides were eluted with $3 \times 100$ μl buffer B and dried in a vacuum concentrator. Samples were resuspended in buffer A supplemented with iRT peptides (Biognosys) and analyzed by mass spectrometry.

## Absolute quantification of PK

Four synthetic peptides (AQUA Ultimate Heavy, lysines and arginines stable isotope-labeled, Thermo Scientific) that are proteotypic for PK were selected for absolute quantification (TQLFGV(K), TYYYSS(R), YIADTAN(K), NYSPASEPSV[C-carbamidomethyl]TVGASD(R)). Heavy-labeled peptides were added at a final concentration of 500 fmol/μl before mass spectrometry analysis to an in-cell LiP sample electroporated previously with 100 μM PK. Peptides were also added to a control sample to which PK was added but not electroporated. For calibration, heavy peptides were spiked into a lysate containing no PK (peptide concentrations 500 fmol/μl, 50 fmol/μl, 5 fmol/μl, 500 amol/μl). Samples were measured by PRM. Peptide intensity was determined by summing up the area of the five most intense fragment ions. A linear model was fitted separately for each peptide to $\log_{10}$-transformed peptide intensities and concentrations of the calibration samples. The linear model was used to quantify the corresponding light PK peptides in the samples of interest. Finally, PK quantity was calculated as the mean of the four determined peptide quantities.

## Liquid chromatography and mass spectrometry data acquisition

### Arsenite-treatment data, datasets for organelle and domain coverage

**DIA**: Samples were injected into a Orbitrap Fusion Lumos Tribrid mass spectrometer equipped with a Waters nanoAQUITY UPLC System using a linear gradient from 3 to 35% B (Eluent A: 0.1% formic acid, Eluent B: 99.9% acetonitrile) over 120 min at a flowrate of 300 nL/min. The MS1 scans ranged from 350 to 1400 $m/z$ with an Orbitrap resolution of 120,000 and an AGC target of 50% or 100 ms injection time. The DIA method consisted of 41 variable width windows with 1 $m/z$ overlap. Fragments were generated by high energy collision induced dissociation (HCD) using a fixed collision energy of 28%. MS2-DIA spectra were acquired over a scan range of 200 to 1800 $m/z$ with an Orbitrap resolution of 30,000 and 200% AGC target or 54 ms injection time.

### Optimization of EP settings, PK concentration and time

**DIA**: Samples were injected into a Orbitrap Eclipse Tribrid mass spectrometer equipped with a EASY-nLC 1200 system using a linear gradient from 3 to 30% B (Eluent A: 0.1% formic acid, Eluent B: 95% acetonitrile, 0.1% formic acid) over 120 min at a flowrate of 300 nL/min. The MS1 scans ranged from 350 to 1400 $m/z$ with an Orbitrap resolution of 120,000 and an AGC target of 800,000. The DIA method consisted of 41 variable width windows with 1 $m/z$ overlap. Fragments were generated by high energy collision induced

dissociation (HCD). MS2-DIA spectra were acquired over a scan range of 150 to 2000 *m/z* with an Orbitrap resolution of 30,000 and 50,000 AGC target.

### In-cell LiP-MS and standard LiP-MS data of rapamycin treatment, supernatant quantification, and test of endocytosis inhibitor

**DIA**: The in-cell LiP-MS samples of 10 μM rapamycin were injected into a Q Exactive Plus Hybrid Quadrupole-Orbitrap mass spectrometer equipped with an EASY-nLC 1000 system (Thermo Fisher Scientific). Separation of peptides was conducted on a 40 cm × 0.75 mm i.d. column (New Objective, PF360-75-10-N-5) which was in house packed with 3 μm beads (Dr. Maisch Reprosil-Pur 120). Separation followed a linear gradient of 3–30% buffer B over a total run time of 120 min, followed by 5 min with an isocratic constant concentration of 90% buffer B (Eluent A: 0.1% formic acid, Eluent B: 95% acetonitrile, 0.1% formic acid). The flow rate was kept at 300 nl/min. The MS1 scans ranged from 350 to 1500 *m/z* with an Orbitrap resolution of 70,000 and an AGC target of 3e6 or 120 ms injection time. The DIA method consisted of 41 variable width windows with 1 *m/z* overlap. Fragments were generated by high energy collision induced dissociation (HCD). MS2-DIA spectra were acquired over a scan range of 250 to 1150 *m/z* with an Orbitrap resolution of 35,000 and 3e6 AGC target.

### Effects of electroporation on protein expression

**DIA**: Samples were injected into a Orbitrap Exploris 480 mass spectrometer equipped with a VanquishNeo System using a linear gradient from 3 to 35% B (Eluent A: 0.1% formic acid, Eluent B: 80% acetonitrile, 0.1% formic acid) over 120 min at a flowrate of 300 nL/min. The MS1 scans ranged from 350 to 1150 *m/z* with an Orbitrap resolution of 120,000 and an AGC target of 200%. The DIA method consisted of 41 variable width windows with 1 m/z overlap. Fragments were generated by high energy collision induced dissociation (HCD). MS2-DIA spectra were acquired over a scan range of 150 to 2000 *m/z* with an Orbitrap resolution of 30,000 and 200% AGC target.

### Absolute PK quantification

**PRM**: Samples were injected into a Orbitrap Fusion Lumos Tribrid mass spectrometer equipped with a Waters nanoAQUITY UPLC System using a linear gradient from 3 to 35% B (Eluent A: 0.1% formic acid, Eluent B: 99.9% acetonitrile, 0.1% formic acid) over 120 min at a flowrate of 300 nL/min. The isolation window was 1.2 *m/z*. Fragments were generated by high energy collision induced dissociation (HCD) using a fixed collision energy of 28%. MS2 spectra were acquired over a scan range of 200 to 1800 *m/z* with an Orbitrap resolution of 60,000 and 1000% AGC target or 118 ms injection time.

## Search engine

The data was searched using Spectronaut 17 (Biognosys), with libraries built from the DIA files. The digestion type was set to semi-specific and peptide minimum length was adjusted to 6. The data was searched against the human reviewed UniProt database and the sequence of PK (https://www.uniprot.org, retrieved on April 16, 2023). Data analysis was performed with Spectronaut via a classical DIA analysis using a targeted library-based search with the following modifications: a single-hit was defined by modified

sequence, and single-hit proteins were excluded. Quant 2.0 was selected for label-free protein quantification, samples were normalized globally on the median, and PTM localization was selected.

## Statistical analysis

Only peptides with a quantity >1 were considered for statistical analysis using R version 4.2.2. A Welch's t-test was used for statistical analysis. *P*-values were adjusted to multiple testing using the Benjamini–Hochberg method. We used cutoffs of FC > 1.5 and of q-value < 0.05 for all proteomics experiments, except for the rapamycin datasets, where a *p*-value < 0.01 was used because no significant changes were detected after adjustment to multiple testing.

## Protein coverage calculation

In the rapamycin experiments, the distributions showing the average number of peptides and the sequence coverage were calculated prior to differential analysis (hence, the error bars) to facilitate comparison between two methods with different numbers of replicates. In other instances when we consider only in-cell LiP experiments, both quantities were calculated after differential analysis considering only peptides present in at least 3 replicates per condition (hence, no error bars).

## Gene ontology enrichment analysis

For gene ontology enrichment analysis, the topGO package (version 2.50.0) was used in R (https://github.com/federicomarini/topGO). Gene ontology annotations were obtained from Uniprot (https://www.uniprot.org, retrieved on May 17, 2022). Enrichment analysis was performed with the Fishers exact test using the elim algorithm in topGO. Significance was set at the *p*-value level <0.01.

## Peptide mapping on AlphaFold2 structures

Peptides were mapped onto structures predicted by Alphafold v4 using PyMOL (version 2.5.4).

## Immunostaining and circularity measurements

HEK293 cells were seeded on 8-well μ-Slides with ibiTreat surface (ibidi #80806) at 25,000 cells per well. After 24 h, 500 μM sodium arsenite (Pfaltz & Bauer #7784-46-5) was added to the cells. After 90 min cells were fixed for 10 min with 4% PFA (Electron Microscopy Sciences #15714) in PBS. After washing twice with PBS the cells were permeabilized for 10 min in PBS with 0.5% Triton X-100 (Sigma-Aldrich #T8787). Cells were washed twice with PBS and then incubated simultaneously with primary antibody, secondary antibody and 1:1000 Hoechst 33258 (Molecular Probes #H-3569) for 1 h in PBS with 0.1% Tween 20 (Sigma-Aldrich #P1379). Cells were washed afterward 2 times in PBS with 0.1% Tween 20 and kept in PBS for imaging.

To determine the circularity of nuclear speckles, a representative field of view of each replicate was taken and thresholded in imageJ to create a mask of the nuclear speckles that matches the observed speckles by eye. Circularity was then measured via the analyze particles function.

## Data availability

The datasets produced in this study are available in the following database: Mass spectrometry proteomics data: PRIDE - PRoteomics IDEntifications Database PXD069095 (https://www.ebi.ac.uk/pride/archive/projects/PXD069095).

The source data of this paper are collected in the following database record: biostudies:S-SCDT-10_1038-S44320-025-00182-6.

## Peer review information

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

## Acknowledgements

We thank Juan Gerez (ETH Zurich) for advice on electroporation settings, Roland Riek (ETH Zurich) for providing access to the electroporation equipment, Donald Hilvert (ETH Zurich) for insightful discussions, Walther Hänseler (University of Zurich) for performing electroporation tests, Charlie Morgan (Australian National University) and Jason Chin (MRC-LMB Cambridge) for sharing the ECFP-YPet plasmid for sensing protease activity, and Federico Uliana (ETH Zurich) for advice on the absolute quantification of proteinase K. This work was funded by the European Research Council (866004), and the EPIC-XS Consortium (823839), the last two under the EU Horizon 2020 program, and the National Center of Competence in Research AntiResist funded by the SNSF (grant numbers 51NF40_180541 and TMAG-3_209354).

## Author contributions

**Franziska Elsässer**: Conceptualization; Data curation; Software; Formal analysis; Validation; Investigation; Visualization; Methodology; Writing—original draft; Writing—review and editing. **Roberta Florea**: Formal analysis; Investigation; Methodology. **Felix Räsch**: Supervision; Validation; Investigation. **Mostafa Zedan**: Investigation. **Nesli-Ece Sen**: Investigation. **Tim Pflästerer**: Investigation. **Tatjana Kleele**: Supervision; Investigation. **Robbie Loewith**: Supervision; Investigation. **Karsten Weis**: Supervision; Investigation. **Natalie de Souza**: Supervision; Writing—original draft; Writing—review and editing. **Paola Picotti**: Conceptualization; Resources; Supervision; Funding acquisition; Methodology; Project administration; Writing—review and editing.

Source data underlying figure panels in this paper may have individual authorship assigned. Where available, figure panel/source data authorship is listed in the following database record: biostudies:S-SCDT-10_1038-S44320-025-00182-6.

## Funding

## Disclosure and competing interests statement

PP is a scientific advisor for the company Biognosys AG (Zurich, Switzerland) and an inventor of a patent licensed by Biognosys AG that covers the LiP-MS method used in this work. FE, RF, and PP have filed a patent application based on this work (European Patent Application No. 23197690). The remaining authors declare no competing interests.

# Expanded View Figures

**Figure EV1.  Effects of incubation time upon proteolytic cleavage.**

(**A**) The plots show peptide intensities after proteolytic cleavage for 1 min to 5 min, comparing samples with 100 µM PK with and without electroporation. Each data point represents a single peptide; half-tryptic peptides are shown in orange and fully tryptic peptides are shown in blue. The shaded gray region marks significance levels (FC > 1.5, q-value < 0.05, $n = 5$ technical replicates). (**B**) The plot shows the fraction of half-tryptic peptides under the indicated conditions after in-cell LiP-MS. (**C**) Changing peptides per protein after 90 min arsenite treatment compared to untreated cells (data in Figure **B**). Peptides due to PK cleavages upon PK electroporation (data in Fig. 1H) are in blue. (**D, E**) The number of detected peptides per protein (**D**) and overall sequence coverage (**E**) at optimized in-cell LiP-MS conditions (2 min +/− EP). (**F**) The plot shows peptide intensities after proteolytic cleavage for 2 min, comparing samples with 100 µM PK to untreated cells (both without electroporation). Each data point represents a single peptide; half-tryptic peptides are shown in orange and fully tryptic peptides are shown in blue. The shaded gray region marks significance levels (FC > 1.5, q-value < 0.05, $n = 5$ technical replicates). (**G**) Gene ontology enrichment analysis of proteins with PK cleavages in (**F**) (p-value < 0.001). (**H**) Position of PK cleavages along the sequence of substrate proteins. The cleavage position is defined either by the position of the first amino acid in a peptide with an n-terminal HT end, or by the position +1 of the last amino acid in a peptide with a c-terminal HT end. Cleavage positions on proteins with more than 2 cleavages are shown in blue, and those with up to 2 cleavages are shown in orange. To generate technical replicates, cells were split into the indicated number of aliquots before treatment.

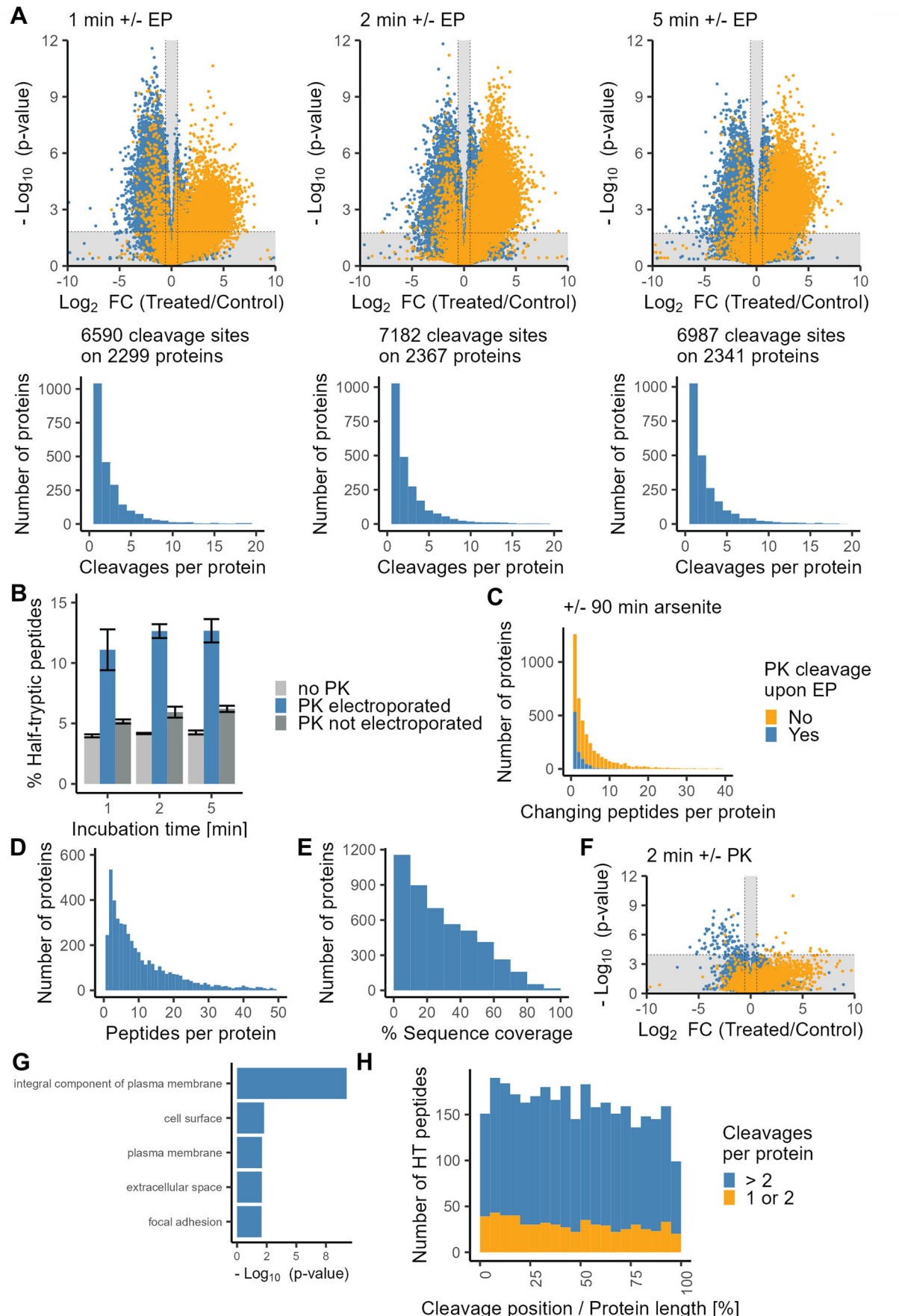

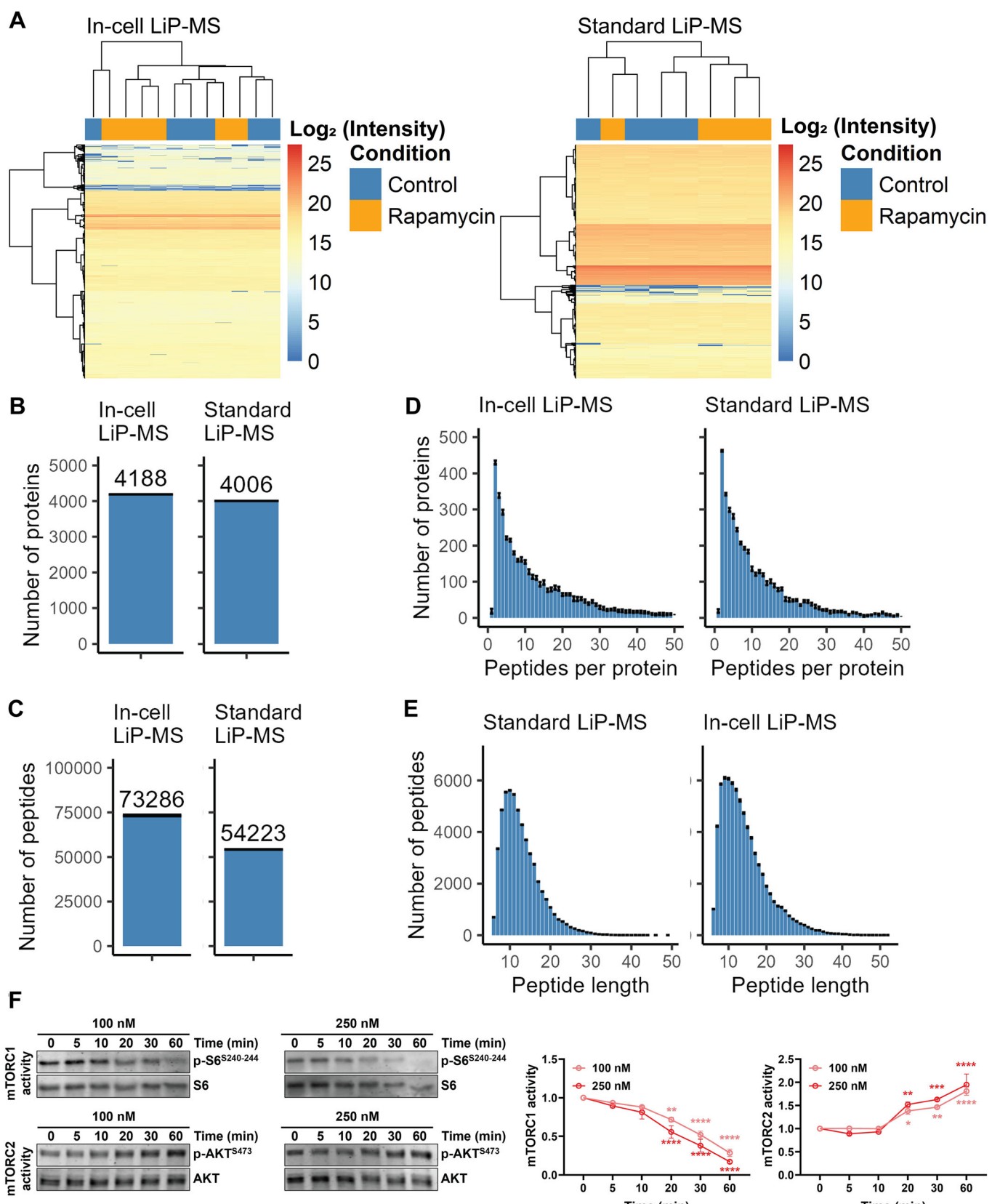

**Figure EV2.  Rapamycin treatment.**

(**A–E**) In-cell LiP-MS of HEK293 cells treated with 20 μM rapamycin compared to DMSO control; standard LiP-MS in a native lysate of HEK293 cells treated with 10 nM rapamycin compared to DMSO control. Plots (**B–E**) show indicated quantities across both conditions (in-cell LiP-MS $n = 12$; standard LiP-MS $n = 8$ technical replicates). Error bars are shown in black. (**A**) Heatmap of peptide intensities (stripped sequence level). Colors above the heatmap correspond to the indicated conditions. (**B**) Number of detected proteins. (**C**) Number of detected peptides. (**D**) Overall sequence coverage considering peptides in both rapamycin treated and control samples (in-cell LiP-MS $n = 12$ replicates; standard LiP-MS $n = 8$ technical replicates). (**E**) Number of peptides with indicated length relative to total peptide intensity. Only peptides with up to 50 amino acids are shown. To generate technical replicates, cells were split into the indicated number of aliquots before treatment. (**F**) HEK293T cells were treated with rapamycin under the indicated conditions and probed for mTORC1 or mTORC2 activity using Western blots against the indicated phospho-proteins. The plots (right) show quantification of the Western blots on the left. Statistics were performed with two-way ANOVA on three biological replicates (*$p \leq 0.05$; **$p \leq 0.01$; ***$p \leq 0.001$). Each replicate was biologically independent, consisting of cells grown separately prior to treatment.

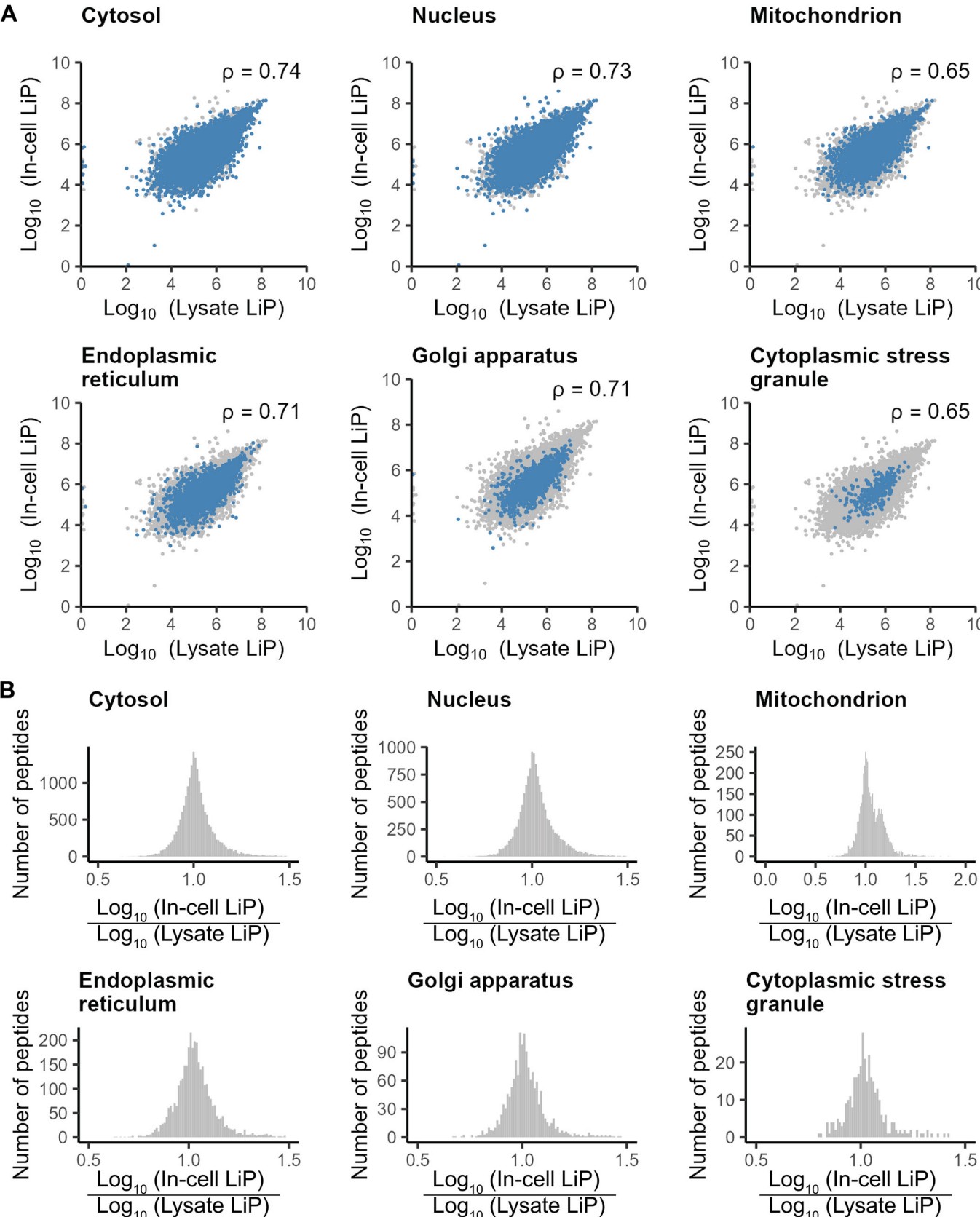

◀   **Figure EV3.   Organelle coverage in LiP-MS in cells.**

(**A**) Correlation of average peptide intensities in indicated organelles in HEK293 cells between LiP-MS in lysate and in cells after 5 min of DMSO treatment (in-cell LiP-MS $n = 6$ replicates; lysate LiP-MS $n = 4$ technical replicates); ρ indicates the Pearson correlation coefficient. Peptides in blue are mapping to proteins located at the indicated organelle based on gene ontology annotation in Spectronaut. Peptides from other organelles are colored gray. (**B**) Ratio of average peptide intensities in lysate and in-cell LiP-MS for indicated organelles. To generate technical replicates, cells were split into the indicated number of aliquots before treatment.

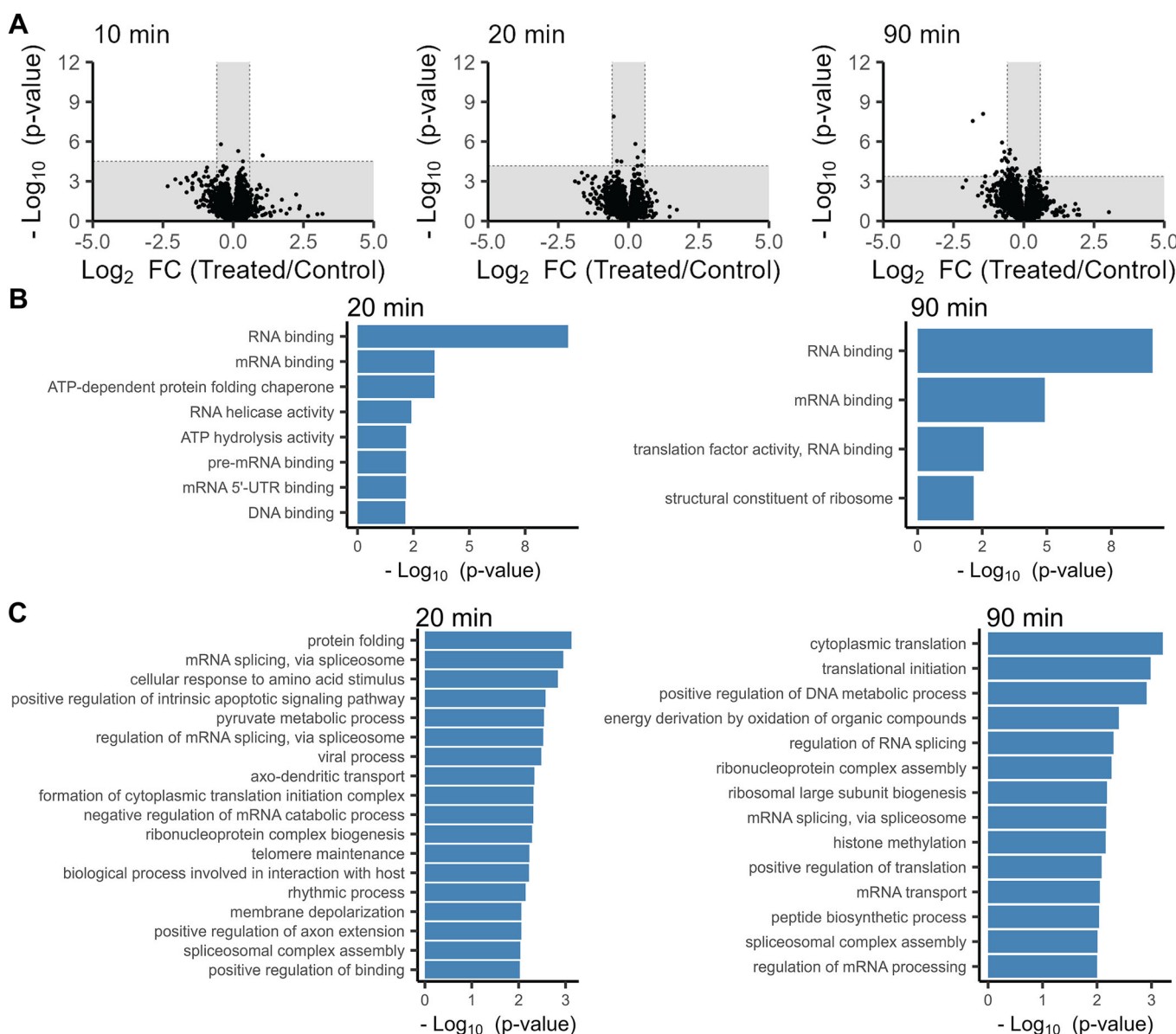

**Figure EV4. Stress granule formation upon arsenite treatment.**

(A) Comparison of HEK293 cells treated with sodium arsenite to untreated cells. Each data point represents a single protein. The shaded gray region marks significance levels (FC > 1.5, *p*-value < 0.05, *n* = 6 biological replicates). (B, C) Gene ontology enrichment analysis of proteins with peptide-level structural changes upon arsenite treatment (q-value < 0.01). (B) Molecular function. (C) Biological process. Each replicate was biologically independent, consisting of cells grown separately prior to treatment.

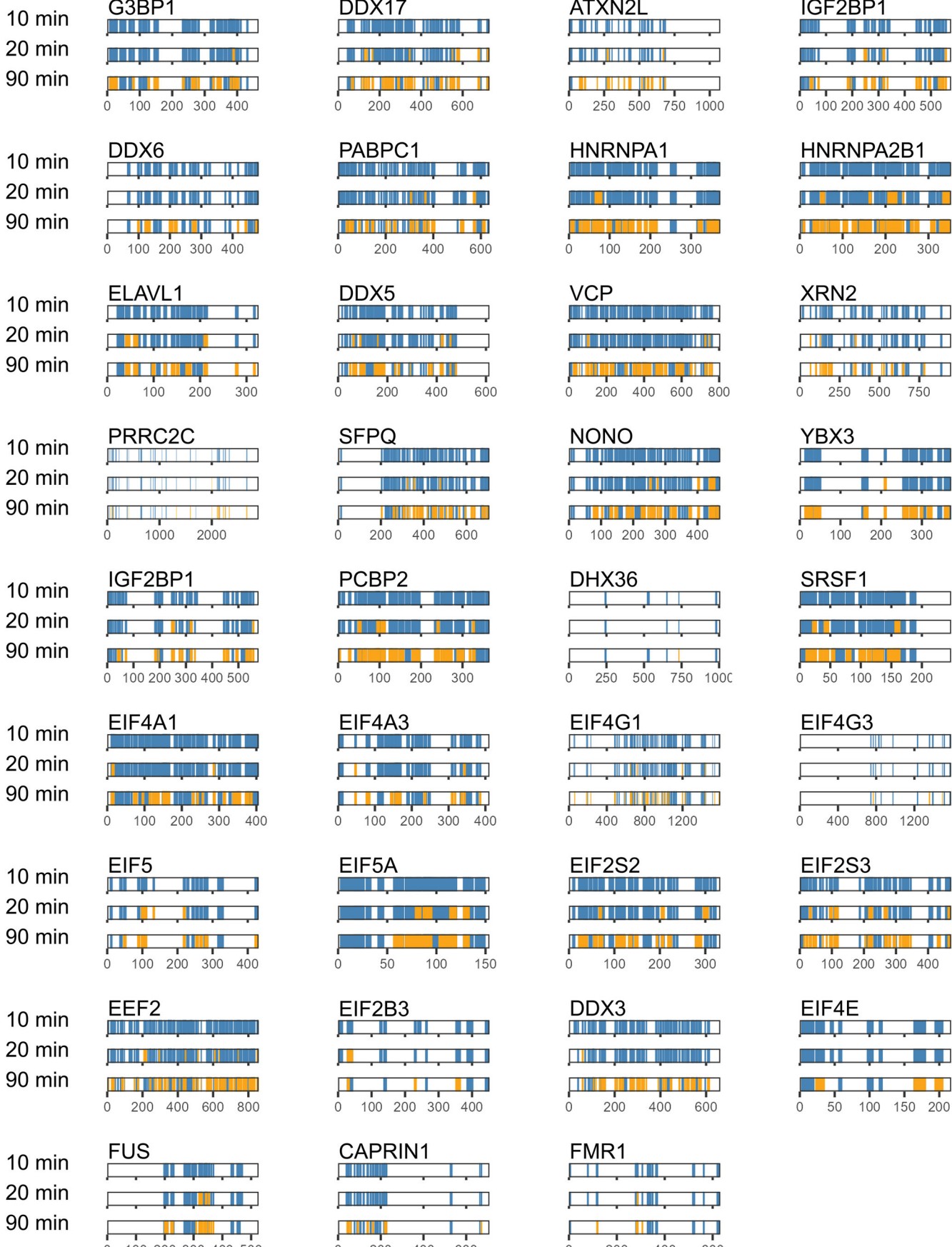

**Figure EV5.  Structural changes in stress granule proteins upon arsenite treatments.**

Proteins showing structural changes upon arsenite treatment. The barcode plots depict the protein sequence, blue regions are detected by mass spectrometry. Orange regions change significantly at the indicated time (FC > 1.5, q-value < 0.05, $n = 6$ biological replicates). Each replicate was biologically independent, consisting of cells grown separately prior to treatment.

