## [Peer Review File · Molecular Systems Biology]

In-cell LiP-MS captures protein structural alterations and biomolecular condensation in living cells

Franziska Elsässer, Roberta Florea, Felix Räscher, Mostafa Zedan, Nesli-Ece Sen, Tim Pflaesterer, Tatjana Kleele, Robbie Loewith, Karsten Weis, Natalie de Souza, and Paola Picotti

Corresponding author(s): Paola Picotti (picotti@imsb.biol.ethz.ch)

Review Timeline:

Submission Date:	7th Apr 25
Editorial Decision:	15th May 25
Revision Received:	7th Oct 25
Editorial Decision:	5th Nov 25
Revision Received:	24th Nov 25
Accepted:	11th Dec 25

Editor: Poonam Bheda

Transaction Report:

15th May 2025

Manuscript Number: MSB-2025-13023

Title: In-cell LiP-MS captures protein structural alterations and biomolecular condensation in living cells

Dear Dr. Picotti,

Thank you for the submission of your manuscript to Molecular Systems Biology. We have now received feedback from the three reviewers who agreed to evaluate your manuscript. As you will see from the reports below, the referees acknowledge the interest of the study and are overall supportive of your work; however they also comment on multiple aspects of the manuscript that should be strengthened in a revision.

Without repeating all the comments listed below, some of the more fundamental issues raised are the following:

- Different mass spectrometers were used for standard and in-cell LiP-MS, making the comparisons not equivalent (Reviewers 1 and 3)
- Novel biological insights are minimal and not fully validated, with advance over standard LiP-MS for generating such discoveries not fully demonstrated (Reviewer 3)

All other issues raised would need to be satisfactorily addressed. Please let me know in case you would like to discuss in further detail any of the any of the reviewer comments or your proposed revisions, I would be happy to schedule a call.

We require:

1) A .docx formatted version of the manuscript text (including legends for main figures, EV figures and tables). Please make sure that the changes are highlighted to be clearly visible. Alternatively you may choose to submit your manuscript as a LaTeX file.

4) A .docx formatted letter INCLUDING the reviewers' reports and your detailed point-by-point responses to their comments. As part of the EMBO Press transparent editorial process, the point-by-point response is part of the Peer Review File (PRF), which will be published alongside your paper.

5) A complete author checklist, which you can download from our author guidelines (<https://www.embopress.org/page/journal/17574684/authorguide#submissionofrevisions>). Please insert information in the checklist that is also reflected in the manuscript. The completed author checklist will also be part of the PRF.

6) Please note that all corresponding authors are required to supply an ORCID ID for their name upon submission of a revised manuscript.

7) It is mandatory to include a 'Data Availability' section after the Materials and Methods. Before submitting your revision, primary datasets produced in this study need to be deposited in an appropriate public database, and the accession numbers and database listed under 'Data Availability'. Please remember to provide a reviewer password if the datasets are not yet public (see <https://www.embopress.org/page/journal/17574684/authorguide#dataavailability>).

In case you have no data that requires deposition in a public database, please state so in this section as follows: "This study includes no data deposited in external repositories". Note that the Data Availability Section is restricted to new primary data that are part of this study.

8) All Materials and Methods need to be described in the main text using our 'Structured Methods' format, which is required for all research articles. According to this format, the Methods section includes a Reagents and Tools Table (listing key reagents, experimental models, software and relevant equipment and including their sources and relevant identifiers) followed by a Methods and Protocols section describing the methods using a step-by-step protocol format. The aim is to facilitate adoption of the methodologies across labs. Please upload the Reagents and Tools table as a separate document when submitting your revised manuscript. More information on how to adhere to this format as well as a downloadable template (.docx) for the

Reagents and Tools Table can be found in our author guidelines:
<https://www.embopress.org/page/journal/17444292/authorguide#structuredmethods>

An example of a Method paper with Structured Methods can be found here:
<https://www.embopress.org/doi/10.15252/msb.20178071>.

9) For data quantification: please specify the name of the statistical test used to generate error bars and p-values, the number (n) of independent experiments (specify technical or biological replicates) underlying each data point and the test used to calculate p-values in each figure legend. The figure legends should contain a basic description of n, p-values and the test applied. Graphs must include a description of the bars and the error bars (s.d., s.e.m.). Please provide exact p-values (in either the figure or figure legend).

10) Our journal encourages inclusion of *data citations in the reference list* to directly cite datasets that were re-used and obtained from public databases. Data citations in the article text are distinct from normal bibliographical citations and should directly link to the database records from which the data can be accessed. In the main text, data citations are formatted as follows: "Data ref: Smith et al, 2001" or "Data ref: NCBI Sequence Read Archive PRJNA342805, 2017". In the Reference list, data citations must be labeled with "[DATASET]". A data reference must provide the database name, accession number/identifiers and a resolvable link to the landing page from which the data can be accessed at the end of the reference. Further instructions are available at .

11) We replaced Supplementary Information with Expanded View (EV) Figures and Tables that are collapsible/expandable online. EV Figures should be cited as 'Figure EV1, Figure EV2' etc... in the text and their respective legends should be included in the main text after the legends of regular figures.

- Additional Tables/Datasets should be labeled and referred to as Table EV1, Dataset EV1, etc. Legends should be provided in a separate tab in case of .xls files. Alternatively, the legend can be supplied as a separate text file (README) and zipped together with the Table/Dataset file.

<https://www.embopress.org/page/journal/17574684/authorguide#expandedview>

12) Author contributions: CRediT has replaced the traditional author contributions section because it offers a systematic machine-readable author contributions format that allows for more effective research assessment. Please remove the Authors Contributions from the manuscript and use the free text boxes beneath each contributing author's name in our system to add specific details on the author's contribution. More information is available in our guide to authors.

13) Disclosure statement and competing interests: We updated our journal's competing interests policy in January 2022 and request authors to consider both actual and perceived competing interests. Please review the policy
<https://www.embopress.org/competing-interests> and update your competing interests if necessary.

14) Every published paper now includes a 'Synopsis' to further enhance discoverability. Synopses are displayed on the journal webpage and are freely accessible to all readers. They include a short stand first (maximum of 300 characters, including space) as well as 2-5 one-sentences bullet points that summarizes the paper. Please write the bullet points to summarize the key NEW findings. They should be designed to be complementary to the abstract - i.e. not repeat the same text. We encourage inclusion of key acronyms and quantitative information (maximum of 30 words / bullet point). Please use the passive voice. Please attach these in a separate file or send them by email, we will incorporate them accordingly.

Please note that these would be the final versions and changes during proofing are usually not allowed.

15) As part of the EMBO Publications transparent editorial process initiative (see our policy here:
https://www.embopress.org/transparent-process#Review_Process), Molecular Systems Biology will publish online a Peer Review File (PRF) to accompany accepted manuscripts.

In the event of acceptance, this file will be published in conjunction with your paper and will include the anonymous referee reports, your point-by-point response and all pertinent correspondence relating to the manuscript. Let us know whether you agree with the publication of the PRF and as here, if you want to remove or not any figures from it prior to publication. Please note that the Author checklist will be published at the end of the PRF.

Molecular Systems Biology has a "scooping protection" policy, whereby similar findings that are published by others during review or revision are not a criterion for rejection. Should you decide to submit a revised version, I do ask that you get in touch after three months if you have not completed it, to update us on the status.

Yours sincerely,

Poonam Bheda, PhD
Scientific Editor
Molecular Systems Biology

Reviewer #1:

This manuscript presents a significant advancement in the field of proteome-wide structural analysis through the development of in-cell Limited Proteolysis-coupled Mass Spectrometry (LiP-MS). The approach successfully addresses the limitations of traditional LiP-MS by enabling the study of protein structures directly within cells, preserving native conditions and capturing dynamic changes. The data presented is compelling, demonstrating the feasibility and utility of the method. However, several points require clarification and further discussion to strengthen the conclusions and enhance the rigor of the study.

1. Lines 130-131: The authors observe no difference in semi-digested peptides between cells treated with PK without electroporation and the control. While acknowledging PK activity on the cell surface, the lack of detectable difference is surprising. Could the observed cleavage be below the limit of detection, or is there a mechanism preventing these extracellularly cleaved peptides from being identified? Further investigation or discussion of this point is warranted.
2. Lines 281-283: The observation that in-cell LiP-MS generates longer peptides with missed cleavages compared to standard LiP-MS is interesting. Could this be attributed to the potential interference from protein interactions within the cell. Could this be due to reduced PK access or activity within the cellular environment? Is this phenomenon consistent across other proteins analyzed, or is it specific to FKBP1A? A more detailed exploration of this effect and its potential impact on data interpretation is needed.
3. Lines 284-286: The use of different mass spectrometers for in-cell and lysate LiP-MS raises concerns about comparability. While acknowledging potential technical variability, utilizing the same instrument would significantly improve the reliability of the comparative analysis. Specifically, it would allow for a more robust assessment of differences in peptide identification numbers, protein identification numbers, and peptide length distributions.
4. The clarity of the peptide labeling in Fig 3a and 3b needs improvement. Utilizing different colors to distinguish between HT and FT peptides would enhance readability. HT and FT peptides should present opposite trends in change of intensity. The data presented in Fig 3a shows some peptides with significant increase in intensity, but no peptides with significant decrease in intensity were observed. Moreover, error bars in Fig 3c, 3d, and 3f are missed.
5. The difference in replicate numbers between in-cell and in-lysate LiP-MS is concerning. The rationale behind this discrepancy should be clearly explained in the section "Subcellular coverage of in-cell LiP-MS".
6. The finding of a significant enrichment for RNA and DNA binding domains in in-cell LiP-MS compared to classical LiP is interesting. Together with the difference in mitochondria proteins, could this result be due to the properties of PK itself, such as its surface charge state? Does the crowded cellular environment enhance the electrostatic interactions between PK and RNA, DNA, and related proteins, leading to this difference?

Reviewer #2:

In this article the authors describe a novel method to study protein structure and protein interactions inside living cells. The method is based on limited proteolysis followed by mass spectrometric analysis (LiP-MS), a methodology pioneered by the same group. Extension of LiP-MS into living cells is a remarkable achievement. The manuscript provides strong evidence that the describe method worked as intended. It is well written and merits publication in Molecular System Biology.

A few minor comments are described below.

1. Figure 3-a and b. In the in-cell experiment (a), FKBP1A peptides have higher intensities in the treated sample, while in the standard experiment (b), FKBP1A peptides can be higher or lower in the treated sample. Can you explain this discrepancy? Additionally, it is challenging to ascertain which sequence corresponds to each dot. It is recommended to add arrows to indicate these relationships.
2. Page 13. "Figure c" should be "Figure 4-c".
3. Figure 3-c and 4-a. It will be beneficial to explain what a "missing value" means.

Reviewer #3:

In the manuscript 'Limited proteolysis-coupled mass spectrometry captures proteome-wide protein structural alterations and biomolecular condensation in living cells' Picotti and co-workers demonstrate that their previously described Limited Proteolysis approach to capture structural alterations on a proteome-wide range can be extended to capture intracellular proteolysis events. It is demonstrated that by introduction of proteinase K in human cells using electroporation intracellular cleavage occurs. The new in-cell LiP-MS method captured the well-characterized and high-affinity interaction of rapamycin with FKBP1A. Further, the study provides a resource of in vivo proteome-wide protein structural changes during arsenite stress, identifies proteins that undergo structural alterations upon cell lysis, and pinpoints structurally altered regions in proteins involved in biomolecular condensates.

The modifications to the LiP-MS approach introduced in the presented study demonstrate a significant step towards intracellular proteolytic profiling to study structural changes in response to cellular perturbations. The electroporation approach used in this study is not without flaws but the authors demonstrate that intracellular proteolysis by transfected proteinase K is plausible and enables detection of protein undergoing structural alterations in response to treatments e.g. by arsenite. However, the study focusses on the technical innovation but offers only two application examples of which the rapamycin study is weak - known target did not achieve statistical significance (when applying multiple testing correction) and the work on arsenite induced stress granules is mainly descriptive and lacks validation.

As such the manuscript in its present state does not provide sufficient evidence that the new method enables the discovery of novel biology or could replace the established lysate based approach for drug target identification.

This reviewer recommends addressing this gap and the detailed points listed below in a major revision.

Major points:

P5, l 150: No abundance changes in stress markers as indication of no stress in 5-6 min.

Abundance changes are not really expected after this short period of time. However, their absence is not a proof that there is no stress that affects protein structures. It might be useful to assess oxidative stress after electroporation. (e.g. seahorse assay) and measure HSP activation rather than expression, if you want to measure HSP level changes then it would need to be later as a response to the stress induced during electroporation

Figure 1h: 7k cleavages on 2.4k proteins, 12 % of human proteome. For how many of those especially the approx 1k for which only one or two cleavages were found, was this on the floppy N-ter/C-ter?

It is argued that with arsenite stress many more cleavage sites are observed.

Hence does the approach lend itself to detect events that lead to destabilization of structures rather than stabilization, so a good way to detect cell stress and less so for e.g. identification of direct drug targets? The comparably weak data for the rapamycin experiment also point in this direction (despite the positive example given). If you want to position this approach for target identification, more examples are needed, e.g. a pan kinase inhibitor could provide insights in this respect.

Comparison standard LiP and in cell lip for rapamycin: with few notable exceptions less accessibility in presence of the drug is found for peptides derived from the known target FKBP1A. This was not the case in the in-cell method. Further, for the in cell method, most prominently regulated peptides of the known target had more missed cleavage sites, which suggests that trypsinolysis of the samples was less effective. Could this be an experimental glitch rather than the result of the in-cell approach? It would be good to compare missed cleavage rates between in cell and in lysate experiments globally.

It is puzzling to see that the crucial comparison of standard vs in cell LiP-MS was analyzed using different mass spectrometers which makes several of the plots regarding peptide coverage etc. rather obsolete.

Please make clear in the results section that p-values are given for stating significance for the rapamycin experiment as no significant changes were detected after adjustment to multiple testing. I have only noticed this when studying the methods section in detail.

How do you expect the heterogeneity of protease delivery with the applied electroporation conditions affect the robustness of the method? Figure 2d suggests that a larger fraction of the cells did not show much or any PK signal.

Did you compare to other methods for PK delivery?

The arsenite study is interesting and understandably data analysis is biased by the focus on confirming known biological effects. There is no elucidation/validation of the potential roles of the three proteins already found at 10 min. As these findings are interpreted as demonstration of the approaches ability to find novelty, some sort of validation would be advisable. A deeper insight into the mechanisms coming into play at 90 min would be desirable as well as the stress granule GO terms are no longer the most prominently enriched ones.

Reviewer #1:

This manuscript presents a significant advancement in the field of proteome-wide structural analysis through the development of in-cell Limited Proteolysis-coupled Mass Spectrometry (LiP-MS). The approach successfully addresses the limitations of traditional LiP-MS by enabling the study of protein structures directly within cells, preserving native conditions and capturing dynamic changes. The data presented is compelling, demonstrating the feasibility and utility of the method. However, several points require clarification and further discussion to strengthen the conclusions and enhance the rigor of the study.

1. Lines 130-131: The authors observe no difference in semi-digested peptides between cells treated with PK without electroporation and the control. While acknowledging PK activity on the cell surface, the lack of detectable difference is surprising. Could the observed cleavage be below the limit of detection, or is there a mechanism preventing these extracellularly cleaved peptides from being identified? Further investigation or discussion of this point is warranted.

The only few significant peptide changes observed between cells treated with PK without electroporation and the no-PK control (**Figure R1**) is due to the fact that cells were washed twice with PBS prior to PK inactivation, since we intended to remove extracellular peptides; we have now clarified this in the manuscript (lines 131-133).

Note that, even with this washing step, we do see evidence for low levels of PK cleavage: in samples with different amounts of added PK, we see a higher fold-change of half-tryptic peptides with larger amounts of PK (**Figure R1A,B**; orange dots shifted to the right in **B**). In addition, we see a slight increase in the total intensity of half-tryptic peptides at 50-200 μM PK compared to the no-PK control (**Figure R1C**, grey bars).

Figure R1 (revised figures 1d, S3f, 1g): (A, B) Peptide intensities after proteolytic cleavage with +/- 100 μM PK (A) or +/- 200 μM PK (B), comparing samples treated with PK to untreated cells (both without electroporation). Each data point represents a single peptide; half-tryptic peptides are shown in orange, and fully tryptic peptides are shown in

blue. The shaded gray region marks significance levels ($FC > 1.5$, $q\text{-value} < 0.05$, $n = 5$ replicates). (C) Fraction of half-tryptic peptides without (control; grey) and with electroporation (blue) at the indicated PK concentration.

2. Lines 281-283: The observation that in-cell LiP-MS generates longer peptides with missed cleavages compared to standard LiP-MS is interesting. Could this be attributed to the potential interference from protein interactions within the cell. Could this be due to reduced PK access or activity within the cellular environment? Is this phenomenon consistent across other proteins analyzed, or is it specific to FKBP1A? A more detailed exploration of this effect and its potential impact on data interpretation is needed.

We now compared missed cleavages for peptides from all detected proteins (**Figure R2**). In standard LiP-MS, peptides with missed cleavages account for 15% of total peptide intensity, compared to 27% in in-cell LiP-MS. Thus, we indeed observed more missed cleavages across the detected proteome for the in-cell method. Note, however, that the optimized protocols for standard and in-cell LiP-MS are not identical and therefore not directly comparable. Trypsin digest of in-cell LiP-MS samples is performed in a buffer containing guanidinium hydrochloride. In standard LiP, we used a buffer containing sodium deoxycholate. These differences are because sodium deoxycholate did not sufficiently quench proteinase K activity from cells, while guanidinium hydrochloride does, based on our data. Trypsin digestion is known to be less effective in buffers containing guanidine hydrochloride¹, leading to more missed cleavages in the in-cell LiP-MS workflow. We have made this clearer in the revised manuscript (lines 317-325) and in the Methods section (lines 1032-1038).

Figure R2 (revised figure 3g): Fraction of peptides with missed cleavages relative to total peptide intensity in in-cell LiP-MS compared to standard LiP-MS.

3. Lines 284-286: The use of different mass spectrometers for in-cell and lysate LiP-MS raises concerns about comparability. While acknowledging potential technical variability, utilizing the same instrument would significantly improve the reliability of the comparative

analysis. Specifically, it would allow for a more robust assessment of differences in peptide identification numbers, protein identification numbers, and peptide length distributions.

We thank the reviewer for raising this point. We have now re-measured the lysate LiP-MS sample on the same mass spectrometer on which the in-cell LiP-MS sample were previously measured (**Figure R3**). The average number of detected proteins was similar in both approaches (in-cell LiP: 4188; standard LiP: 4006), and the average number of peptides was higher for in-cell LiP (73286) than standard LiP (54223), corresponding to higher protein sequence coverage in the in-cell method (**Figure R3C**). Most peptides were quantified consistently across replicates (**Figure R3D**), and while coefficients of variation were slightly higher for in-cell LiP-MS (**Figure R3E**), they were still within the range of CVs reported for standard LiP experiments². There was similar fraction of HT peptides between the in-cell and in-lysate methods (**Figure R3F**). As already discussed in the previous point, we observed more missed cleavages and longer peptides in in-cell compared to standard LiP-MS (**Figure R3G, H**), at least in part due to differences in trypsin efficiency under the respective experimental conditions. These data are now shown in **Revised Figure 3 and S6**.

Figure R3 (Revised Figure 3). Target detection and reproducibility of in-cell LiP-MS. (a) Changes in in-cell LiP-MS peptide intensities in HEK293 cells treated with 20 μ M

rapamycin compared to DMSO control. Each data point represents a single peptide; half-tryptic peptides of FKBP1A are shown in orange, fully-tryptic peptides of FKBP1A are shown in blue. The lines marks significance levels ($FC > 1.5$, p -value < 0.01 , $n = 6$ technical replicates). (b) Changes in standard LiP-MS peptide intensities in a native lysate of HEK293 cells treated with 10 nM rapamycin compared to DMSO control (6 technical replicates). Each data point represents a single peptide; half-tryptic peptides of FKBP1A are shown in orange, fully-tryptic peptides of FKBP1A are shown in blue. The shaded gray region marks significance levels ($FC > 1.5$, p -value < 0.01 , $n = 4$ technical replicates). The four FKBP1A peptides with the lowest p -value are highlighted. (c) Overall sequence coverage of experiments in (a, b). The plot shows the indicated quantities across both conditions (in-cell LiP-MS $n=12$ technical replicates; standard LiP-MS $n=8$ technical replicates). Error bars are shown in black. (d) Number of peptides with missing values per treatment in (a, b). The plots report the number of replicates per condition, in which a specific peptide was not quantified. A missing value of 0 indicates that the peptide was quantified in all six replicates, 1 indicates that the peptide was not quantified in 1 out of 6 replicates, and so on. (e) Coefficient of variation (CV) of peptide intensities per treatment in (a,b). (f) Fraction of half-tryptic peptides relative to total peptide intensity for the experiments in (a, b). (g-h) Plots show the indicated quantities across both conditions (in-cell LiP-MS $n=12$; standard LiP-MS $n=8$). (g) Fraction of peptides with missed cleavages after tryptic digestion relative to total peptide intensity for the experiments in (a, b). (h) Fraction of peptides with indicated length relative to total peptide intensity in (a, b). Only peptides with up to 50 amino acids are shown. To generate replicates, cells were split into the indicated number of aliquots before treatment.

4. The clarity of the peptide labeling in Fig 3a and 3b needs improvement. Utilizing different colors to distinguish between HT and FT peptides would enhance readability. HT and FT peptides should present opposite trends in change of intensity. The data presented in Fig 3a shows some peptides with significant increase in intensity, but no peptides with significant decrease in intensity were observed. Moreover, error bars in Fig 3c, 3d, and 3f are missed.

As suggested, we now highlight HT peptides of FKBP1A in orange and FT peptides in blue (**Figure R3A-B, see above**). Upon drug binding, one would expect protection of the binding site, and therefore that HT peptides decrease in intensity and the corresponding FT peptides increase in intensity; more generally, one would indeed expect opposite FT and HT trends as suggested by the reviewer. However, we have already observed in control experiments with the standard LiP-MS protocol that this is sometimes not the case. This is due to two factors: secondary cleavages by PK, and missed cleavages by trypsin, the occurrence of which can be influenced by the presence and extent of a prior PK cleavage event. We now mention this in the revised manuscript (lines 317-328). Although these events can sometimes result in unpredictable directions of change for HT and FT peptides, we have shown in several previous studies that

comparing peptide intensities between conditions still correctly identifies regions undergoing structural change. because both secondary cleavages and missed cleavages are themselves influenced by the local structural environment. Since these effects occur reproducibly across treated and untreated samples, they contribute to a structurally informative proteolytic fingerprint rather than introducing random noise.

Regarding the referee's point about error bars, the original **Figure 3c** reported the number of missing values across all replicates per condition, i.e., a missing value of 1 indicates that a peptide was quantified in 5 out of 6 replicates. There is only one exact value for each peptide and condition displayed, so that it is not possible to calculate an error (**Figure R3D, see above**). We now explain the plot more clearly in the legend, see above.

The original **Figure 3d** reported the median CV across all peptides. To address the reviewer's request for showing the variability, we now show violin and box plots of peptide CVs (**Figure R3E**). As already reported in our initially submitted version, CVs are slightly higher for in-cell LiP-MS owing to the variability of PK uptake and of cell compared to the homogeneous lysates probed by standard LiP-MS, but these values are still within the range of CVs reported for standard LiP experiments ².

The original **Figure 3f** reported the overall sequence coverage in the experiment comparing rapamycin to control samples. To include error bars as requested, we now calculated sequence coverage for each replicate (**Figure R3C, see above**), as well as the number of peptides per protein (**Figure R4**), in both cases considering all detected peptides. The small error bars in the two distributions indicate that variability of sequence coverage across replicates is very low for both LiP-MS approaches.

Figure R4 (revised figure S6d): Number of peptides per protein considering peptides in both rapamycin treated and control samples. Only proteins with up to 50 peptides are shown. Error bars are shown in black.

5. The difference in replicate numbers between in-cell and in-lysate LiP-MS is concerning. The rationale behind this discrepancy should be clearly explained in the section "Subcellular coverage of in-cell LiP-MS".

Standard LiP-MS is generally conducted with four replicates². We increased the number of replicates for in cell LiP as part of the optimization of the method to account for its slightly higher variability, in order to obtain robust results. We had already mentioned this in the previous version of the manuscript (revised manuscript lines 199-200), and we have now added further explanation as requested (lines 364-366).

6. The finding of a significant enrichment for RNA and DNA binding domains in in-cell LiP-MS compared to classical Lip is interesting. Together with the difference in mitochondria proteins, could this result be due to the properties of PK itself, such as its surface charge state? Does the crowded cellular environment enhance the electrostatic interactions between PK and RNA, DNA, and related proteins, leading to this difference?

This is an interesting suggestion. We assessed the electrostatic potential of proteinase K (PDB: 2prk) under physiological conditions (pH 7.0 and pH 7.4) using the PDB2PQR and APBS tools (<https://server.poissonboltzmann.org/>) with default settings. Note that the pH of the lysate in standard LiP-MS is also maintained at 7.4, so that PK is likely to be in the same electrostatic state in the cell and in the native lysate. Visualized in Pymol, PK exhibits both positively charged and negatively charged surface regions (**Figure R5A, B**). These charged surface regions may indeed enable interactions with charged DNA, RNA and protein molecules in the crowded in-cell environment. For instance, charged intracellular DNA/RNA molecules may recruit PK via its positive surface, leading to cleavage of DNA/RNA binding proteins and therefore to the enrichment of these domains that we see in the in-cell data. Since DNA/RNA molecules are likely to be removed by centrifugation in standard LiP-MS prior to the PK cleavage step, this effect would not occur. We have now commented on this in the revised manuscript while making clear that this remains speculative (lines 410-419), and we show the data below in revised **Figure S4**.

Figure R5 (revised figure S4d): Electrostatic potential of proteinase K surface (PDB 2prk) at the indicated pH, units ranging from -3 kT/e (red) to 3 kT/e (blue).

Reviewer #2:

In this article the authors describe a novel method to study protein structure and protein interactions inside living cells. The method is based on limited proteolysis followed by mass spectrometric analysis (LiP-MS), a methodology pioneered by the same group. Extension of LiP-MS into living cells is a remarkable achievement. The manuscript provides strong evidence that the describe method worked as intended. It is well written and merits publication in Molecular System Biology.

A few minor comments are described below.

1. Figure 3-a and b. In the in-cell experiment (a), FKBP1A peptides have higher intensities in the treated sample, while in the standard experiment (b), FKBP1A peptides can be higher or lower in the treated sample. Can you explain this discrepancy? Additionally, it is challenging to ascertain which sequence corresponds to each dot. It is recommended to add arrows to indicate these relationships.

As requested, we have now added arrows to the volcano plots (**Figure R6**), and we now also use color to indicate half tryptic (HT, orange) and fully tryptic (FT, blue) peptides for added clarity.

Figure R6 (revised figure 3a, b): (a) Changes in in-cell LiP-MS peptide intensities in HEK293 cells treated with 20 μ M rapamycin compared to DMSO control. Each data point represents a single peptide; half-tryptic peptides of FKBP1A are shown in orange, fully-tryptic peptides of FKBP1A are shown in blue. The lines marks significance levels ($FC > 1.5$, p -value < 0.01 , $n = 6$ technical replicates). (b) Changes in standard LiP-MS peptide intensities in a native lysate of HEK293 cells treated with 10 nM rapamycin compared to DMSO control (6 technical replicates). Each data point represents a single peptide; half-tryptic peptides of FKBP1A are shown in orange, fully-tryptic peptides of FKBP1A are shown in blue. The shaded gray region marks significance levels ($FC > 1.5$, p -value < 0.01 , $n = 4$ technical replicates). The four FKBP1A peptides with the lowest p -value are highlighted.

Note that, to improve comparability between the standard and in-cell methods as requested by referee 1, we have now re-measured the standard LiP-MS samples on the mass spectrometer previously used for the in-cell LiP-MS samples; **Figure R6** shows this new comparison (please also see **Figure R3** above). The precise pattern of FKBP1A peptides changed in this new measurement. Specifically, all six HT peptides that were significantly changing in the previous measurement are still significantly changing, and we find two additional HT peptides (GWEEGVAQ, VEDVELLKLE) to be significantly changing upon rapamycin treatment. However, the two FT peptides that previously changed are not detected in the new measurement. We note that effects like this are expected when comparing measurements made months apart and on different instruments. The loss of the FT peptides may be the result of different sensitivities of the instruments or peptide loss during storage. In general, the peptide sequences that are found altered in a given LiP experiment are also known to depend on the exact enzyme-to-substrate ratio and incubation times used. Especially the E/S ratio may vary slightly between the in cell and in lysate versions of LiP. However, we find that altered peptides found in different experiments typically map to the same regions even when their sequences vary slightly.

In terms of the direction of peptide intensity changes, the theoretical expectation is that, upon drug binding, there is protection of the binding site such that HT peptides decrease in intensity and the corresponding FT peptides increase in intensity. Indeed, the expectation in general is of opposite FT and HT trends in LiP. However, we have already observed in control experiments with the standard LiP-MS protocol that this is sometimes not the case. This is due to two factors: secondary cleavages by PK, and missed cleavages by trypsin, the occurrence of which can be influenced by the presence and extent of a prior PK cleavage event. We now mention this in the revised manuscript (lines 317-328). We note however that, although these events can sometimes result in unpredictable directions of change for HT and FT peptides, we have shown in several previous studies that comparing peptide intensities between conditions still correctly identifies regions undergoing a protease accessibility change, because both secondary cleavages and missed cleavages are themselves influenced by the local structural environment. Since these effects occur reproducibly across treated and untreated samples, they contribute to a structurally informative proteolytic fingerprint rather than introducing random noise.

Missed cleavages in particular occur more frequently in the in-cell method (see also our response to referee 1, point 2) because of the experimental conditions. In brief, guanidinium hydrochloride is used to quench PK in the in-cell method, rather than sodium deoxycholate used in the standard method, because sodium deoxycholate did not fully quench PK activity from cells. Since trypsin digestion is less effective in buffers containing guanidine hydrochloride¹, this causes more missed cleavages.

2. Page 13. "Figure c" should be "Figure 4-c".

We corrected this in the manuscript.

3. Figure 3-c and 4-a. It will be beneficial to explain what a "missing value" means.

The various "missing value" plots (**Figure R7**) report the number of replicates per condition in which any given peptide was not quantified. A missing value of 0 indicates that the peptide was quantified in all six replicates, 1 indicates that the peptide was not quantified in 1 out of 6 replicates, and so on. We clarified this in the relevant legends of **revised figures 3c, 4a, and S8b (Figure R7)**. In each case we have included the following text "*The plots report the number of replicates per condition, in which a specific peptide was not quantified. A missing value of 0 indicates that the peptide was quantified in all six replicates, 1 indicates that the peptide was not quantified in 1 out of 6 replicates, and so on.*"

We realized that Figure 4a previously did not include the missing value of 0. We corrected this in the revised manuscript. This slightly increased the number of peptides

detected by in-cell LiP in at least 3 replicates (75,172 instead of 73,912; among unique peptides 27,041 instead of 26,070). We corrected this in the text (lines 377-379).

Figure R7 (revised figure 3d, S8b, 4a): Number of peptides with missing values per condition in different experiments. The plots report the number of replicates per condition in which any given peptide was not quantified. A missing value of 0 indicates that the peptide was quantified in all six replicates, 1 indicates that the peptide was not quantified in 1 out of 6 replicates, and so on. Peptides with 0-6 missing values are shown by colour. (A) Rapamycin treatment experiment comparing missing values between in-cell LiP-MS and standard LiP-MS. (B) Missing values after different time points of arsenite treatment and measurement with in-cell LiP-MS. (C) Comparison of missing values in untreated HEK293 cells analyzed with in-cell LiP-MS and standard LiP-MS. The bars show the total number of peptides detected by in-cell LiP-MS (all) and peptides unique to in-cell LiP-MS (Unique) after 5 min of DMSO treatment (in-cell LiP n = 6 replicates; lysate LiP n = 4 replicates).

Reviewer #3:

In the manuscript 'Limited proteolysis-coupled mass spectrometry captures proteome-wide protein structural alterations and biomolecular condensation in living cells' Picotti and co-workers demonstrate that their previously described Limited Proteolysis approach to capture structural alterations on a proteome-wide range can be extended to capture intracellular proteolysis events. It is demonstrated that by introduction of proteinase K in human cells using electroporation intracellular cleavage occurs. The new in-cell LiP-MS method captured the well-characterized and high-affinity interaction of rapamycin with FKBP1A. Further, the study provides a resource of in vivo proteome-wide protein structural changes during arsenite stress, identifies proteins that undergo structural alterations upon cell lysis, and pinpoints structurally altered regions in proteins involved in biomolecular condensates.

The modifications to the LiP-MS approach introduced in the presented study demonstrate a significant step towards intracellular proteolytic profiling to study structural changes in response to cellular perturbations. The electroporation approach used in this study is not without flaws but the authors demonstrate that intracellular proteolysis by transfected proteinase K is plausible and enables detection of protein undergoing structural alterations in response to treatments e.g. by arsenite. However, the study focusses on the technical innovation but offers only two application examples of which the rapamycin study is weak - known target did not achieve statistical significance (when applying multiple testing correction) and the work on arsenite induced stress granules is mainly descriptive and lacks validation.

As such the manuscript in its present state does not provide sufficient evidence that the new method enables the discovery of novel biology or could replace the established lysate based approach for drug target identification.

This reviewer recommends addressing this gap and the detailed points listed below in a major revision.

Major points:

P5, l 150: No abundance changes in stress markers as indication of no stress in 5-6 min. Abundance changes are not really expected after this short period of time. However, their absence is not a proof that there is no stress that affects protein structures. It might be useful to assess oxidative stress after electroporation. (e.g. seahorse assay) and measure HSP activation rather than expression, if you want to measure HSP level changes then lit would need to be later as a response to the stress induced during electroporation

As suggested, we assessed HSP levels at a later time point after electroporation. We electroporated cells using our in-cell LiP settings (1000 V, 1 pulse of 25 ms), cultured them for 4 h, and then harvested and lysed all cells (including dead cells) and prepared the lysates for mass spectrometry analysis. We did not observe any significant changes

in protein abundances either for HSP or other proteins, even after this longer time (Figure R8A). We chose 4h as the time point for our analysis because translation of proteins is expected to be active in response to stress at this time point.

We also performed a Seahorse ATP rate measurement as suggested. We observed a slight but statistically insignificant drop in oxygen consumption rate (OCR), and therefore in mitochondrial respiration, in cells 24 hours after electroporation (Figure R8B). This indicates, if anything a small reduction in oxidative stress and reactive oxygen species relative to control cells. These data are now shown in Revised Figure S1g,h.

Figure R8 (revised figure S1g, h): (A) The plot shows protein intensities comparing electroporated to non-electroporated cells 4 h after electroporation (1000 V, 1 pulse of 25 ms). Each data point represents a single protein. The shaded gray region marks significance levels (FC > 1.5, q-value < 0.01, n = 5 technical replicates) (B) Oxygen consumption rate (OCR) of control cells (green) and cells 24h after electroporation (orange) (left) and basal mitochondrial respiration (right) calculated as basal OCR minus non-mitochondrial OCR (post Rot/AA). Data are shown as mean \pm SD (n=10 wells per condition). ns, not significant (Mann–Whitney U test).

Figure 1h: 7k cleavages on 2.4k proteins, 12 % of human proteome. For how many of those especially the approx 1k for which only one or two cleavages were found, was this on the floppy N-ter/C-ter?

To address this, we quantified cleavages across all proteins, using only HT peptides for this analysis because they allow exact localization of the cleavage site. Of 3233 HT peptides, 30 peptides were not bonafide HT peptides but represented the very N-terminus of the protein; these were excluded from the analysis. For the remaining 3203 HT peptides, cleavages were distributed almost equally along protein sequences, without a marked preference for protein termini (Figure R9). This was also the case among proteins for which only one or two cleavages were found. We now report this analysis in revised Figure S3H.

Figure R9 (revised figure S3h): Position of PK cleavages along the sequence of all substrate proteins. The cleavage position is defined either by the position of the first amino acid in a peptide with an n-terminal HT end, or by the position +1 of the last amino acid in a peptide with a c-terminal HT end. Cleavage positions on proteins with more than 2 cleavages are shown in blue, and those with up to 2 cleavages are shown in orange.

It is argued that with arsenide stress many more cleavage sites are observed. Hence does the approach lend itself to detect events that lead to destabilization of structures rather than stabilization, so a good way to detect cell stress and less so for e.g. identification of direct drug targets? The comparably weak data for the rapamycin experiment also point in this direction (despite the positive example given). If you want to position this approach for target identification, more examples are needed, e.g. a pan kinase inhibitor could provide insights in this respect.

We used identification of the target of rapamycin as a test case simply because it is an experiment that we have previously used to characterize classical LiP-MS² and that others have used to characterize limited proteolysis with trypsin (i.e. the PELSA approach³). In general, classical LiP-MS is much better suited for finding specific interactions like drug targets than in-cell LiP because cell lysis should minimize downstream effects of the drug, therefore introducing fewer confounding factors and facilitating interpretation. In-cell LiP-MS on the other hand, is more suited to capture complex biological processes like the stress response which may depend on intact cellular organization. We have made this clearer in the revised manuscript (lines 684-693).

With regard to the point the referee raises about detecting stabilization versus destabilization, we do not find that LiP (in either format) is particularly suited to one versus the other. In previous studies we have shown that changes in the LiP pattern can reflect both increased accessibility/destabilization (e.g., in the case of local or global unfolding^{4, 5} as well as increased resistance/stabilization (e.g., in the case of protein aggregation or protein complex formation^{4,6-8}). There is no reason to believe that in-cell LiP behaves differently from in-lysate LiP in this regard as it is based on the same

biochemical principle. Indeed, in our in-cell LiP data we detect both cases of increased and decreased protease accessibility. An increase or decrease in the intensity of fully tryptic peptides in a LiP experiment typically indicates that the corresponding region is becoming less or more accessible to proteolysis, respectively (with the exceptions described in our next response). After 90 min of arsenite treatment, 10% and 16% of FT peptides increase and decrease their abundance respectively, suggesting that proteins become more as well as less accessible to proteolysis. Similar observations apply to other treatments we have performed and analyzed via both the in-cell and in-lysate versions of LiP, suggesting that both approaches can capture both destabilization and stabilization events.

Comparison standard LiP and in cell lip for rapamycin: with few notable exceptions less accessibility in presence of the drug is found for peptides derived from the known target FKBP1A. This was not the case in the in-cell method. Further, for the in cell method, most prominently regulated peptides of the known target had more missed cleavage sites, which suggests that trypsinolysis of the samples was less effective. Could this be an experimental glitch rather than the result of the in-cell approach? It would be good to compare missed cleavage rates between in cell and in lysate experiments globally.

Yes, the difference in missed cleavage sites between in-cell LiP-MS and classical LiP-MS is indeed the result of experimental differences, and not of structure-based cleavage. During development of in-cell LiP-MS, we tested different buffers for quenching of PK. In standard LiP-MS, a buffer containing sodium deoxycholate is used, but this did not sufficiently quench PK in the in-cell LiP setup. We therefore used guanidinium hydrochloride for this purpose in in-cell LiP. Since trypsin digestion is less effective in buffers containing guanidine hydrochloride ¹, this leads to more missed cleavages in the optimized in-cell LiP-MS protocol. This is now clearer in the revised manuscript (lines 317-325) and in the Methods section (lines 1032-1038).

We have also now quantified the extent of this issue at the proteome level by comparing missed cleavages for peptides from all detected proteins, as also requested by referee 1 (**Figure R2**, reproduced below for convenience). In standard LiP-MS, peptides without missed cleavages account for 85% of total peptide intensity, compared to 73% in in-cell LiP-MS. These data are shown in **revised Figure 3f**.

A	B
---	---

Figure R2 (revised figure 3g): Fraction of peptides with missed cleavages relative to total peptide intensity in in-cell LiP-MS compared to standard LiP-MS.

Regarding patterns of protease accessibility for the rapamycin target FKBP1A, the theoretical expectation is that, upon drug binding, there is protection of the binding site such that HT peptides decrease in intensity and the corresponding FT peptides increase in intensity. Indeed opposite FT and HT trends is the general expectation in LiP. However, we have already observed in control experiments with the standard LiP-MS protocol that this is sometimes not the case. This is due to two factors: secondary cleavages by PK, and missed cleavages by trypsin (as discussed also above), the occurrence of which can be influenced by the presence and extent of a prior PK cleavage event. We now mention this in the revised manuscript (lines 317-328). We note however that, although these events can sometimes result in unpredictable directions of change for HT and FT peptides, we have shown in several previous studies that comparing peptide intensities between conditions still correctly identifies regions undergoing a protease accessibility change, because both secondary cleavages and missed cleavages are themselves influenced by the local structural environment. Since these effects occur reproducibly across treated and untreated samples, they contribute to a structurally informative proteolytic fingerprint rather than introducing random noise. Please also see our response to the next point.

It is puzzling to see that the crucial comparison of standard vs in cell LiP-MS was analyzed using different mass spectrometers which makes several of the plots regarding peptide coverage etc. rather obsolete.

We thank the reviewer for raising this point. We have now re-measured the standard LiP samples on the mass spectrometer previously used for the in-cell LiP samples and now show this more meaningful comparison (**Figure R6A, B**, reproduced also below for convenience).

Figure R6 (revised figure 3a, b): (a) Changes in in-cell LiP-MS peptide intensities in HEK293 cells treated with 20 μ M rapamycin compared to DMSO control. Each data point represents a single peptide; half-tryptic peptides of FKBP1A are shown in orange, fully-tryptic peptides of FKBP1A are shown in blue. The lines marks significance levels ($FC > 1.5$, p -value < 0.01 , $n = 6$ technical replicates). (b) Changes in standard LiP-MS peptide intensities in a native lysate of HEK293 cells treated with 10 nM rapamycin compared to DMSO control (6 technical replicates). Each data point represents a single peptide; half-tryptic peptides of FKBP1A are shown in orange, fully-tryptic peptides of FKBP1A are shown in blue. The shaded gray region marks significance levels ($FC > 1.5$, p -value < 0.01 , $n = 4$ technical replicates). The four FKBP1A peptides with the lowest p -value are highlighted.

We still see higher sequence coverage for in-cell than for standard LiP-MS, as before (average number of peptides for in-cell LiP = 73286, for standard LiP = 54223), and the target of rapamycin FKBP1A is still detected. Specifically, all six HT peptides that were significantly changing in the previous measurement are still significantly changing, and we find two additional HT peptides (GWEEGVAQ, VEDVLLKLE) to be significantly changing upon rapamycin treatment. However, the two FT peptides that previously changed are not changing in the new measurement. One fell below the significance threshold, although it still shows abundance changes, and the second one is not detectable in 3 out of 4 control replicates. We note that effects like these are expected when comparing measurements made months apart and on different instruments. The loss of the FT peptides may be the result of different sensitivities of the instruments or peptide loss during storage. Please also see our response to referee 1 points 3 and 4. These new comparative data are now shown in revised **Figure 3**.

Please make clear in the results section that p -values are given for stating significance for the rapamycin experiment as no significant changes were detected after adjustment to multiple testing. I have only noticed this when studying the methods section in detail. We have now made this clear in the main text (Lines 303-308).

How do you expect the heterogeneity of protease delivery with the applied electroporation conditions affect the robustness of the method? Figure 2d suggests that a larger fraction of the cells did not show much or any PK signal.

The reviewer correctly points out that many cells (55.5%) do not contain PK after electroporation. This certainly dilutes our signal as it will reduce fold-changes in peptide intensity between conditions by adding unchanging background. This heterogeneity of PK delivery does not seem to affect robustness (i.e., variability) because fluorescence microscopy of cells electroporated with Cy5-labeled PK showed very stable average fluorescence intensity across replicates (**Figure R10**). Indeed, in mass spectrometry too we measure average intensities of cleavage products across cells. In keeping with our expectations, our mass spectrometry measurements show that coefficients of variation of LiP peptides for in-cell LiP, while slightly higher than in our standard LiP-MS experiment (**Figure R3E**, see above), are still below the 25% that would be considered high in a standard LiP experiment². We have now mentioned these points in both the results (lines 266-268) and the discussion sections (lines 677-680) of the revised manuscript.

Figure R10 (revised figure 2c): PK-Cy5 signal after background subtraction in cells with and without electroporation from three independent replicates. *p*-value between mean of electroporated and control replicates < 0.001, *t*-test.

Did you compare to other methods for PK delivery?

Yes, before settling on electroporation, we had tested several other delivery methods. We tested expression of low-specificity proteases in cells as well as photo-caging of PK using unnatural amino acids. However, the proteases were either not soluble or aggregated in human cells (proteinase K), or did not result in active proteases due to auto-inhibition under physiological conditions (papain, subtilisin). We could not find the right conditions for activating these proteases within minutes under physiological conditions. We also tested delivery of active proteases by extracellular vesicles, but this yielded very low uptake rates. Previous efforts in the lab also investigated membrane permeabilization by detergents (digitonin, saponin) and pore-forming toxins (streptolysin-

o) for proteinase K delivery into cells. The detergents strongly impacted cell integrity, and streptolysin-o did not sufficiently permeabilize cells.

The arsenite study is interesting and understandably data analysis is biased by the focus on confirming known biological effects. There is no elucidation/validation of the potential roles of the three proteins already found at 10 min. As these findings are interpreted as demonstration of the approaches ability to find novelty, some sort of validation would be advisable.

In response to the reviewer's comment, we took a closer look at SERBP1, one of the three proteins changing after 10 min arsenite treatment. SERBP1 and its yeast homolog Stm1 have previously been found to act as ribosome dormancy factors in other stresses (rapamycin treatment or nitrogen starvation⁹; however, to our knowledge this protein has not been previously associated with the early stages of stress granule formation. Based on the structural change of SERBP1 in in-cell LiP-MS data and the known interaction of the protein, we hypothesize that SERBP1 is recruited to ribosomes and triggers the dissociation of polysomes into dormant monosomes already after 10 min of arsenite treatment. To test this, we performed polysome profiling at this time point and compared to the profiles of untreated control cells. We observed that 10 min of arsenite treatment led to a higher monosome peak (**Figure R11**), indicating polysome collapse, even though stress-granules are not yet visible under these conditions.

We then extracted the proteins from selected fractions and probed for SERBP1 and for the ribosomal protein Rpl7 on Western blot. Rpl7 levels confirmed polysome collapse upon arsenite treatment. In control conditions, SERBP1 was detected at higher levels in the fraction unbound to ribosomes, in fractions containing free 40S and 60S ribosome subunits, and to some extent in the fractions containing polysomes. Upon arsenite treatment, unbound SERBP1 and polysome-associated SERBP1 were almost undetectable. Instead, SERBP1 was detected in fractions containing 80S ribosomes and free 40S and 60S ribosomal subunits. This association with dormant ribosomes is in line with observations from yeast grown in the presence of other stresses with the homolog Stm1⁹, and added a novel insight into the early stress response upon arsenite treatment. We now show these data in revised Figure **8a, b**.

Figure R11 (revised figure 8A, B): (a, b) SERBP1 is associated with polysome disassembly after 10 min of arsenite treatment. (a) *Polysome profiling of HEK293 with or without 10 min arsenite treatment (2 replicates).* (b) *Western blots for SERBP1 and the ribosomal protein Rpl7 in selected fractions. Left blot, control; right blot, 10 minutes of arsenite treatment.*

A deeper insight into the mechanisms coming into play at 90 min would be desirable as well as the stress granule GO terms are no longer the most prominently enriched ones.

The GO terms after 90 min of arsenite treatment are more general than after 20 min of arsenite treatment but in both cases include several terms related to RNA binding and processing and ribosomal structures, in addition to stress granules (**Figure R12**). We interpret this larger number of general GO terms as most probably being due to the larger number of structural changes at the later time point, which therefore encompass more processes (at 20 minutes of arsenite treatment, 2325 peptides on 1149 proteins had changed significantly, rising to 20,841 peptides on 3146 proteins at 90 minutes of treatment). Importantly, the association with RNA binding/processing/ribosomes is to be expected, since RNA molecules are a central constituent of stress granules. Note that the 'cytoplasmic stress granule' term, which is still enriched at 90 min of treatment,

points to changes in 190 of the 200 high confidence stress granule-associated proteins, as we had already mentioned (lines 514-516 in the revised manuscript).

Figure R12 (revised figures 5c; S7b,c): Gene ontology enrichment analysis of structural changes upon 20 min and 90 min arsenite treatment of mammalian cells (q -value < 0.01). (top) Cellular component. (middle) Molecular function. (bottom) Biological process.

Still, the reviewer is correct that the broad nature of these terms does not allow us to draw specific conclusions on underlying mechanisms, as is sometimes the case also for classical protein expression profiling experiments. In an exploratory analysis, we therefore limited the GO analysis to proteins containing only the most significantly changing peptides (q-value < 0.0005, **Figure R13**), which yielded more specific enriched terms/processes. Notably, RNA and ribosome associated terms were still the most strongly enriched when considering these top hits. In addition, we now see enrichment at the 90 min time point in terms related to proteasomal processing, mitochondrial outer membrane processes, and secretory granules. Also enriched is the term “response to toxic substance”, and “plasma membrane repair”, suggesting that cell integrity may be affected by the long-term arsenite stress, as previously described¹⁰. Hence, our dataset captures a broad range of processes triggered by arsenite stress. We have mentioned this analysis in the discussion (lines 739-746) as a starting point for future studies and show the functional analyses in revised figure S10.

Figure R13 (revised Figure S10): Gene ontology enrichment analysis of structural changes upon 90 min arsenite treatment of mammalian cells ($p\text{-value} < 0.01$), considering only the most strongly changing peptides ($FC > 1.5$, $q\text{-value} < 0.05$, $n = 6$ replicates). (top) Cellular component. (middle) Molecular function. (bottom) Biological process.

To study mechanisms triggered by arsenite treatment, a separate study is currently ongoing in the lab, which investigates global complex formation and dissociation at 90 min of treatment using size-exclusion chromatography (SEC) in combination with chemical crosslinking. Ongoing SEC analyses corroborate the finding that Gem-associated protein 5 (GEMIN5) (**Figure R14**), which is involved in spliceosome formation, the DNA repair protein DNA ligase 3 (LIG3), the E3 ubiquitin-protein ligase CHIP (STUB1), which mediates proteasomal degradation, and tRNA (guanine(6)-N(2))-methyltransferase THUMP3 (THUMPD3), which methylates tRNAs, all undergo structural rearrangements. Specifically, these proteins likely undergo changes in their assembly states, based on the SEC profiles. LiP patterns of all of these proteins are also significantly changing in our in-cell LiP-MS dataset after 90 min of treatment. These SEC data are shown here for the referees, but we have not added them to the revised manuscript as they are the subject of a separate study. Besides these few highlighted examples, we also expect to see large-scale overlaps in proteins with structural changes captured by in-cell LiP-MS and complex formation/dissociation captured by SEC. Global analyses of SEC data are ongoing as part of our separate study.

Figure for referee with unpublished data and its description has been removed upon request by the authors.

1. Proc, J. L. *et al.* A quantitative study of the effects of chaotropic agents, surfactants, and solvents on the digestion efficiency of human plasma proteins by trypsin. *J Proteome Res* 9, 5422–37 (2010).
2. Malinovska, L. *et al.* *Proteome-Wide Structural Changes Measured with Limited Proteolysis-Mass Spectrometry: An Advanced Protocol for High-Throughput Applications.* (2022).
3. Li, K. *et al.* A peptide-centric local stability assay enables proteome-scale identification of the protein targets and binding regions of diverse ligands. *Nat Methods* 22, 278–282 (2025).
4. Feng, Y. *et al.* Global analysis of protein structural changes in complex proteomes. *Nat Biotechnol* 32, 1036–1044 (2014).
5. Leuenberger, P. *et al.* Cell-wide analysis of protein thermal unfolding reveals determinants of thermostability. *Science (1979)* 355, (2017).
6. Dörig, C. *et al.* Global profiling of protein complex dynamics with an experimental library of protein interaction markers. *Nat Biotechnol* (2024) doi:10.1038/s41587-024-02432-8.
7. Cappelletti, V. *et al.* Dynamic 3D proteomes reveal protein functional alterations at high resolution in situ. *Cell* 184, 545-559.e22 (2021).
8. Holfeld, A. *et al.* Systematic identification of structure-specific protein-protein interactions. *Mol Syst Biol* 20, 651–675 (2024).
9. Shetty, S., Hofstetter, J., Battaglioni, S., Ritz, D. & Hall, M. N. TORC1 phosphorylates and inhibits the ribosome preservation factor Stm1 to activate dormant ribosomes. *EMBO J* 42, e112344 (2023).
10. Li, M.-K. *et al.* Long-Term Real-Time Tracking of Morphology and Migration of Neuronal Cells under Oxidative Stress. *Chemical & Biomedical Imaging* 3, 191–198 (2025).

5th Nov 2025

Manuscript Number: MSB-2025-13023R

Title: In-cell LiP-MS captures protein structural alterations and biomolecular condensation in living cells

Dear Dr. Picotti,

Thank you for the submission of your revised manuscript to Molecular Systems Biology. We have now received the enclosed reports from the referees that were asked to re-assess it. As you will see the reviewers are now globally supportive and I am pleased to inform you that we will be able to accept your manuscript pending the following final amendments:

- 1) In the main manuscript file, please label the corresponding author in the author list and provide an email address on the title page.
- 2) Please include keywords to max. 5.
- 3) Please format the Data availability section according to the example below:
"The datasets and computer code produced in this study are available in the following databases:
- Chip-Seq data: Gene Expression Omnibus GSE46748 (<https://www.ncbi.nlm.nih.gov/geo/query/acc.cgi?acc=GSE46748>)
- Modeling computer scripts: GitHub (<https://github.com/SysBioChalmers/GECKO/releases/tag/v1.0>)
- [data type]: [full name of the resource] [accession number/identifier] ([doi or URL or identifiers.org/DATABASE:ACCESSION])"
- 4) The proteomics dataset in PRIDE should now be publicly released, and the direct link to the dataset should be provided in the Data Availability statement (as requested above).
- 5) Please rename "Competing Interest statement" to "Disclosure and competing interests statement". We updated our journal's competing interests policy in January 2022 and request authors to consider both actual and perceived competing interests. Please review the policy <https://www.embopress.org/competing-interests> and update your competing interests if necessary.
- 6) Author contributions: Please remove it from the manuscript and specify author contributions in our submission system. CRediT has replaced the traditional author contributions section because it offers a systematic machine-readable author contributions format that allows for more effective research assessment. You are encouraged to use the free text boxes beneath each contributing author's name to add specific details on the author's contribution. More information is available in our guide to authors:
<https://www.embopress.org/page/journal/17574684/authorguide#authorshipguidelines>
- 7) References: Please correct the reference citation in the reference list such that when there are more than 10 authors on a paper, only the first 10 should be listed, followed by "et al." and that DOIs are removed. Please check "Author Guidelines" for more information.
<https://www.embopress.org/page/journal/17574684/authorguide#referencesformat>
- 8) Our journal encourages inclusion of *data citations in the reference list* to directly cite datasets that were re-used and obtained from public databases. Data citations in the article text are distinct from normal bibliographical citations and should directly link to the database records from which the data can be accessed. In the main text, data citations are formatted as follows: "Data ref: Smith et al, 2001" or "Data ref: NCBI Sequence Read Archive PRJNA342805, 2017". In the Reference list, data citations must be labeled with "[DATASET]". A data reference must provide the database name, accession number/identifiers and a resolvable link to the landing page from which the data can be accessed at the end of the reference. Further instructions are available at .
- 9) In the Methods, please take care of the following:
 - The Materials and Methods section should be renamed to "Methods".
 - Please also be sure to include a sentence in the Methods as to whether or not the cell lines were recently authenticated and tested for mycoplasma contamination. Please also be sure to update the Author Checklist with this information and where it can be found in the manuscript.
 - Please move the antibody information given in the table in the Methods into the in Reagents and Tools table (as described in the next point)
- 10) All Materials and Methods need to be described in the main text using our 'Structured Methods' format. According to this format, the Methods section includes a Reagents and Tools Table (listing key reagents, experimental models, software and relevant equipment and including their sources and relevant identifiers) followed by a Methods and Protocols section describing the methods, ideally using a step-by-step protocol format. The aim is to facilitate adoption of the methodologies across labs. Please download and fill our Reagents and Tools Table template (.docx), which you can find in our author guidelines:
<https://www.embopress.org/page/journal/14693178/authorguide#structuredmethods>.
When submitting your revised manuscript, please do not include the Reagents and Tools Table in the Methods section of the manuscript but upload it as a separate file choosing the file type "Reagent Table".
An example of a Method paper with Structured Methods can be found here:
<https://www.embopress.org/doi/10.15252/msb.20178071> . "
- 11) Please place individual sections of the manuscript in the following order: Title page - Abstract & Keywords - Introduction - Results - Discussion - Methods - Data Availability - Acknowledgements - Disclosure and Competing Interests Statement - References - Figure Legends - Expanded View Figure Legends.
- 12) For the figures and figure legends, please take care of the following:

- Please note that the figure 8D is mislabeled as figure 8B in the manuscript. This needs to be rectified.
- Please note that the exact p values are not provided in the legend of figure 2C
- Please indicate the statistical test used for data analysis in the legends of figures 3A, B; 4C, D, F; 5B, C
- Please note that information related to n is missing in the legend of figure 1G
- Please note that the error bars are not defined in the legends of figures 1G, 2A, B
- Please note that the measure of center for the error bars needs to be defined in the legend of figure 8D

13) Supplementary Figures 1-10 should be compiled into an Appendix, and this needs to be uploaded in PDF format. The title page should contain "Appendix for + manuscript title" and a Table of Contents with the page numbers for the listed items. The nomenclature should be Appendix Figure Sx and Appendix Table Sx throughout the manuscript and Appendix PDF.

14) Synopsis:

- Synopsis image: Please provide a graphic that summarises the main findings of the manuscript on a glance and upload it as a high-resolution jpeg file 550 pixels wide x (300-600) pixels high.

- Synopsis text: Please provide a separate word document including a short standfirst (maximum of 300 characters, including spaces) and up to 5 bullet points to summarise the key NEW findings. They should be designed to be complementary to the abstract - i.e. not repeat the same text. We encourage inclusion of key acronyms and quantitative information (maximum of 30 words / bullet point). Please use the passive voice.

15) Source Data: Please ensure that a completed Source Data checklist is uploaded as a Related Manuscript File. Source Data should be uploaded to your submission and organized as a single source data file (zipped) per figure for main figures (all EV and/or Appendix figure Source Data can be included in a single folder), with the panels clearly visible in the folder structure instead of a single excel file for all Source Data. e.g. all the Source data files for figure 1 need to be saved in a single folder and this needs to be zipped and then uploaded as "SD figure 1.zip" file.

16) As part of the EMBO Publications transparent editorial process initiative (see our policy here:

https://www.embopress.org/transparent-process#Review_Process), Molecular Systems Biology will publish online a Peer Review File (PRF) to accompany accepted manuscripts. This file will be published in conjunction with your paper and will include the anonymous referee reports, your point-by-point response and all pertinent correspondence relating to the manuscript. Let us know whether you agree with the publication of the PRF and as here, if you want to remove or not any figures from it prior to publication. Please note that the Authors checklist will be published at the end of the PRF.

17) After your paper is published, we may promote it on social media. If you have any handles or hashtags for Bluesky you would like included, please let us know.

18) Please provide a point-by-point letter INCLUDING my comments and your detailed responses (as Word file).

I look forward to reading a new revised version of your manuscript as soon as possible.

Yours sincerely,

Poonam Bheda, PhD
Scientific Editor
Molecular Systems Biology

Reviewer #1:

The authors have addressed all the concerns I had in the revision. I recommend acceptance of this manuscript.

Reviewer #2:

The authors have properly addressed the reviewer's concerns and the manuscript is ready for publication.

Thank you for the submission of your revised manuscript to Molecular Systems Biology. We have now received the enclosed reports from the referees that were asked to re-assess it. As you will see the reviewers are now globally supportive and I am pleased to inform you that we will be able to accept your manuscript pending the following final amendments:

1) In the main manuscript file, please label the corresponding author in the author list and provide an email address on the title page.

Paola Picotti has been labeled as the corresponding author and her email address has been added (picotti@imsb.biol.ethz.ch).

2) Please include keywords to max. 5.

The keywords "LiP-MS, structural proteomics, stress granules, nuclear speckles, biomolecular condensation" have been added before the abstract.

3) Please format the Data availability section according to the example below:

"The datasets and computer code produced in this study are available in the following databases:

- Chip-Seq data: Gene Expression Omnibus GSE46748

(<https://www.ncbi.nlm.nih.gov/geo/query/acc.cgi?acc=GSE46748>)

- Modeling computer scripts: GitHub

(<https://github.com/SysBioChalmers/GECKO/releases/tag/v1.0>)

- [data type]: [full name of the resource] [accession number/identifier] ([doi or URL or identifiers.org/DATABASE:ACCESSION])"

The section has been formatted accordingly, as follows.

The datasets produced in this study are available in the following database:

- mass spectrometry proteomics data: PXD069095

(<https://www.ebi.ac.uk/pride/archive/projects/PXD069095>)

PRIDE - PRoteomics IDentifications Database

4) The proteomics dataset in PRIDE should now be publicly released, and the direct link to the dataset should be provided in the Data Availability statement (as requested above).

The PRIDE dataset is now publicly released and the link is provided.

5) Please rename "Competing Interest statement" to "Disclosure and competing interests statement". We updated our journal's competing interests policy in January 2022 and request authors to consider both actual and perceived competing interests. Please review the policy

<https://www.embopress.org/competing-interests> and update your competing interests if necessary.

The section has been renamed.

6) Author contributions: Please remove it from the manuscript and specify author contributions in our submission system. CRediT has replaced the traditional author contributions section because it offers a systematic machine-readable author contributions format that allows for more effective research assessment. You are encouraged to use the free text boxes beneath each contributing author's name to add specific details on the author's contribution. More information is available in our guide to authors:

<https://www.embopress.org/page/journal/17574684/authorguide#authorshipguidelines>

We have removed the author contributions from the manuscript and will use the CRediT system instead.

7) References: Please correct the reference citation in the reference list such that when there are more than 10 authors on a paper, only the first 10 should be listed, followed by "et al." and that DOIs are removed. Please check "Author Guidelines" for more information.

<https://www.embopress.org/page/journal/17574684/authorguide#referencesformat>

We have formatted references as requested.

8) Our journal encourages inclusion of *data citations in the reference list* to directly cite datasets that were re-used and obtained from public databases. Data citations in the article text are distinct from normal bibliographical citations and should directly link to the database records from which the data can be accessed. In the main text, data citations are formatted as follows: "Data ref: Smith et al, 2001" or "Data ref: NCBI Sequence Read Archive PRJNA342805, 2017". In the Reference list, data citations must be labeled with "[DATASET]". A data reference must provide the database name, accession number/identifiers and a resolvable link to the landing page from which the data can be accessed at the end of the reference. Further instructions are available at

<https://www.embopress.org/page/journal/17574684/authorguide#referencesformat>.

The RNAgranuleDB v2.0 is now mentioned in the text as (Data ref: Millar et al., 2023).

We have cited it in the reference list as follows:

[Dataset] Millar, S. R., Huang, J. Q., Schreiber, K. J., Tsai, Y.-C., Won, J., Zhang, J., Moses, A. M., & Youn, J.-Y. (2023). A New Phase of Networking: The Molecular Composition and Regulatory Dynamics of Mammalian Stress Granules. *Chemical Reviews*, 123(14), 9036–9064.

<https://rnagranuledb.lunenfeld.ca>

9) In the Methods, please take care of the following:

- The Materials and Methods section should be renamed to "Methods".

The section has been renamed.

- Please also be sure to include a sentence in the Methods as to whether or not the cell lines were recently authenticated and tested for mycoplasma contamination. Please also be sure to update the Author Checklist with this information and where it can be found in the manuscript.

We have added the following sentence to the Methods section on cell culture conditions, "The cell lines were recently tested for mycoplasma contamination. Cell lines were not authenticated". The author checklist has been updated.

- Please move the antibody information given in the table in the Methods into the in Reagents and Tools table (as described in the next point)

The antibody information has been moved into the Reagents and Tools table.

10) All Materials and Methods need to be described in the main text using our 'Structured Methods' format. According to this format, the Methods section includes a Reagents and Tools Table (listing key reagents, experimental models, software and relevant equipment and including their sources and relevant identifiers) followed by a Methods and Protocols section describing the methods, ideally using a step-by-step protocol format. The aim is to facilitate adoption of the methodologies

across labs.

Please download and fill our Reagents and Tools Table template (.docx), which you can find in our author guidelines:

<https://www.embopress.org/doi/10.15252/msb.20178071>. "

We have included the requested Reagents and Tools table.

11) Please place individual sections of the manuscript in the following order: Title page - Abstract & Keywords - Introduction - Results - Discussion - Methods - Data Availability - Acknowledgements - Disclosure and Competing Interests Statement - References - Figure Legends - Expanded View Figure Legends.

We have ordered the manuscript as requested.

12) For the figures and figure legends, please take care of the following:

- Please note that the figure 8D is mislabeled as figure 8B in the manuscript. This needs to be rectified.

We have corrected the error.

- Please note that the exact p values are not provided in the legend of figure 2C

We have added the requested information as follows: “(e) PK-Cy5 signal after background subtraction in cells with and without electroporation from three independent technical replicates. p-value between mean of electroporated and control replicates < 0.001, two-tailed t-test (replicate 1: p-value = 2.045E-07; replicate 2: p-value = 0.0006261; replicate 3: p-value = 2.032E-07).”

- Please indicate the statistical test used for data analysis in the legends of figures 3A, B; 4C, D, F; 5B, C

We have added the requested information to the figure legends. In brief, the tests were:

A two-sample unpaired t-test (Fig3A, B).

A Fisher’s exact test using the elim-algorithm in *topGO* (Fig4C, D, and 5C).

A two-sample unpaired t-test with Benjamini-Hochberg adjustment (Fig 5B).

A one-sided Fisher’s exact test (Fig. 4F).

- Please note that information related to n is missing in the legend of figure 1G

The information (n = 5) was added to the figure legend.

- Please note that the error bars are not defined in the legends of figures 1G, 2A, B

Error bars indicate the standard deviation in figures 1G, 2A, B. This information was added to the figure legends.

- Please note that the measure of center for the error bars needs to be defined in the legend of figure 8D

We now indicate that the measure of center is the mean, and have provided the values (mean of control: 0.604; mean of sodium arsenite treatment: 0.741).

13) Supplementary Figures 1-10 should be compiled into an Appendix, and this needs to be uploaded in PDF format. The title page should contain "Appendix for + manuscript title" and a Table of Contents with the page numbers for the listed items. The nomenclature should be Appendix Figure Sx and Appendix Table Sx throughout the manuscript and Appendix PDF.

We have selected 5 figures as Expanded View figures. The remaining Supplementary figures have been compiled into an Appendix as requested.

14) Synopsis:

- Synopsis image: Please provide a graphic that summarises the main findings of the manuscript on a glance and upload it as a high-resolution jpeg file 550 pixels wide x (300-600) pixels high.

We have provided a synopsis figure.

- Synopsis text: Please provide a separate word document including a short standfirst (maximum of 300 characters, including spaces) and up to 5 bullet points to summarise the key NEW findings. They should be designed to be complementary to the abstract - i.e. not repeat the same text. We encourage inclusion of key acronyms and quantitative information (maximum of 30 words / bullet point). Please use the passive voice.

We have added the requested document.

We have checked the synopsis text and image.

15) Source Data: Please ensure that a completed Source Data checklist is uploaded as a Related Manuscript File. Source Data should be uploaded to your submission and organized as a single source data file (zipped) per figure for main figures (all EV and/or Appendix figure Source Data can be included in a single folder), with the panels clearly visible in the folder structure instead of a single excel file for all Source Data. e.g. all the Source data files for figure 1 need to be saved in a single folder and this needs to be zipped and then uploaded as "SD figure 1.zip" file.

We have added the requested source data and the source data checklist.

16) As part of the EMBO Publications transparent editorial process initiative (see our policy here: https://www.embopress.org/transparent-process#Review_Process), Molecular Systems Biology will publish online a Peer Review File (PRF) to accompany accepted manuscripts. This file will be published in conjunction with your paper and will include the anonymous referee reports, your point-by-point response and all pertinent correspondence relating to the manuscript. Let us know whether you agree with the publication of the PRF and as here, if you want to remove or not any figures from it prior to publication. Please note that the Authors checklist will be published at the end of the PRF.

We ask that you remove Figure R14 and the paragraph of text describing it from the response to reviewers document provided with our last version.

17) After your paper is published, we may promote it on social media. If you have any handles or hashtags for Bluesky you would like included, please let us know.

Our Bluesky handle is [@picottilab.bsky.social](https://bsky.app/profile/picottilab.bsky.social)

18) Please provide a point-by-point letter INCLUDING my comments and your detailed responses (as Word file).

We have provided this as requested.

11th Dec 2025

Manuscript number: MSB-2025-13023RR

Title: In-cell LiP-MS captures protein structural alterations and biomolecular condensation in living cells

Dear Dr. Picotti,

Congratulations on an excellent manuscript, I am pleased to inform you that your manuscript has been accepted for publication in Molecular Systems Biology. Thank you for your comprehensive response to referee concerns. It has been a pleasure to work with you to get this to the acceptance stage.

You may qualify for financial assistance for your publication charges - either via a Springer Nature fully open access agreement or an EMBO initiative. Check your eligibility: <https://link.springer.com/journal/44320/how-to-publish-with-us>

Yours sincerely,

Poonam Bheda, PhD
Scientific Editor
Molecular Systems Biology

>>> Please note that it is Molecular Systems Biology policy for the transcript of the editorial process (containing referee reports and your response letter) to be published as an online supplement to each paper. If you do NOT want this, you will need to inform the Editorial Office via email immediately. More information is available here: <https://link.springer.com/partners/embo-press/editorial-policies#Peer%20review>